# Small Leaves, Big Diversity: Citizen Science and Taxonomic Revision Triples Species Number in the Carnivorous *Drosera microphylla* Complex (*D.* Section *Ergaleium*, Droseraceae)

**DOI:** 10.3390/biology12010141

**Published:** 2023-01-16

**Authors:** Thilo Krueger, Alastair Robinson, Greg Bourke, Andreas Fleischmann

**Affiliations:** 1School of Molecular and Life Sciences, Curtin University, GPO Box U1987, Bentley, WA 6102, Australia; 2National Herbarium of Victoria, Royal Botanic Gardens Victoria, Melbourne, VIC 3004, Australia; 3P.O. Box 3001, Bilpin, NSW 2758, Australia; 4Botanische Staatssammlung München, Menzinger Strasse 67, 80638 Munich, Germany; 5GeoBio-Center LMU, Ludwig Maximilians University, 80638 Munich, Germany

**Keywords:** Australia, carnivorous plants, non-core *Caryophyllales*, Nepenthales, sundews, taxonomy, typification

## Abstract

**Simple Summary:**

A novel taxonomic treatment is provided for the *Drosera microphylla* complex, which is a group of closely related carnivorous plants endemic to southwest Western Australia. The species that comprise this group are generally rare, micro-endemic, and are potentially threatened by habitat destruction and illegal collection. Resolving the taxonomy and systematics of this complex has been critical to the accurate assessment of its component species under conservation legislation. Following two decades of fieldwork in Western Australia, studies of preserved plant collections, and crucial contributions by citizen scientists and social media, we establish here that the *Drosera microphylla* complex comprises nine distinct species, three times the number previously recognised. Four species are here described and illustrated as new to science. Two previously described varieties are here re-circumscribed as distinct species in light of their rediscoveries via social media posts, allowing them to be studied for the first time since they were described more than 100 years ago. We provide examples from the genus *Drosera* for the impact of social media and citizen science on taxonomic work and biological conservation. This work demonstrates the great potential that citizen science has in supporting rapid advances in taxonomic knowledge in the face of extinction crises worldwide.

**Abstract:**

The carnivorous *Drosera microphylla* complex from southwest Western Australia comprises a group of rare, narrowly endemic species that are potentially threatened by habitat destruction and illegal collection, thus highlighting a need for accurate taxonomic classification to facilitate conservation efforts. Following extensive fieldwork over two decades, detailed studies of both Australian and European herbaria and consideration of both crucial contributions by citizen scientists and social media observations, nine species of the *D. microphylla* complex are here described and illustrated, including four new species: *D. atrata*, *D. hortiorum*, *D. koikyennuruff*, and *D. reflexa*. The identities of the previously described infraspecific taxa *D. calycina* var. *minor* and *D. microphylla* var. *macropetala* are clarified. Both are here lectotypified, reinstated, and elevated to species rank. A replacement name, *D. rubricalyx*, is provided for the former taxon. Key morphological characters distinguishing the species of this complex include the presence or absence of axillary leaves, lamina shape, petal colour, filament shape, and style length. A detailed identification key, comparison figures, and a distribution map are provided. Six of the nine species are recommended for inclusion on the Priority Flora List under the Conservation Codes for Western Australian Flora and Fauna.

## 1. Introduction

*Drosera* L. (Droseraceae Salisbury; commonly known as sundews) is a cosmopolitan genus of carnivorous plants comprising ca. 260 herbaceous species, of which ca. 110 are endemic to the Southwest Australian Floristic Region (SWAFR) [1,2]. The SWAFR is recognised as the global centre of diversity for *Drosera* and for carnivorous plants in general [2,3]; this high alpha-diversity is enabled by the region’s abundance of nutrient-deficient soils, broad diversity and close geographic proximity of different habitats, and the seasonal Mediterranean climate with long-term climatic stability [4,5].

Within *Drosera*, the geophytes of *D.* section *Ergaleium* (DC.) Planch. (“tuberous sundews”; infrageneric classification following Fleischmann et al. [2]) form the largest and morphologically most diverse evolutionary lineage, comprising 71 currently accepted species [1,2,6]. The members of this monophyletic clade [2,7] can have rosetted, erect, or even climbing habits, various stem branching patterns and leaf arrangements, and many different style shapes [1,8]. However, they all share a swollen, stem-derived subterranean tuber that acts as a storage organ and allows the plants to perennate underground during the very dry summer conditions of the SWAFR [2,8,9,10,11,12,13,14,15] and to survive bushfires [12,15,16].

The complex of morphologically similar (and putatively closely related) species that includes *Drosera microphylla* Endl. consists of freestanding erect, tuberous sundew species that produce long-petiolate, peltate, alternating leaves with long internodes along the stem and terminal inflorescences [1,8,15]. They are further characterised by relatively large sepals that equal or exceed the petals in length and by deeply concave (“cupped”) petals [1,8,12,17] and non-ephemeral flowers that open for several days but close every night [1,12,15,18,19]. This combination of features is not paralleled in any other species of *Drosera*.

The *D. microphylla* complex has a complicated taxonomic history, which is summarised in Table 1. The first species of the complex, *Drosera microphylla*, was described by Austrian botanist Stephan Ladislaus Endlicher in 1837 based on a single plant collected by his contemporary, Austrian naturalist Karl (“Charles”) von Hügel, from near Albany (“King Georges [George] Sound”; *Hügel s.n.*, W 0009732!) on the south coast of Western Australia [1,20]. Eleven years later, French botanist and *Drosera* monographer Jules Émile Planchon separated *D. calycina* Planch. based on its crescent-like lamina shape [17], in contrast with those of *D. microphylla* that both he and Endlicher [20] described as “orbicular” (circular). In 1864, *D. calycina* var. *minor* Benth. became the first infraspecific taxon of the *D. microphylla* complex, distinguished by smaller leaves and flowers as compared to *D. calycina* [21]. Bentham also synonymised *D. microphylla* with *D. filicaulis* Endl. (a taxon based on *Hügel s.n.* (W 0046809!), which today is considered conspecific with *D. menziesii* R.Br. ex DC., a species that is not part of the *D. microphylla* complex [1], and incorrectly stated that the short diagnosis provided by Endlicher [20] would not allow it to be distinguished from *D. filicaulis* and other species [21].

In his monographic treatment of Droseraceae, Ludwig Diels resurrected *D. microphylla* and synonymised both *D. calycina* and *D. calycina* var. *minor* under the former name [8], resulting in a broad treatment of *D. microphylla* that was generally followed for more than a century (e.g., [12,22,23]). However, Diels did recognise another infraspecific taxon, *D. microphylla* var. *macropetala* Diels, based on its larger petals that are white in dried specimens (“siccata pallida”) [8].

Marchant [24] and Marchant et al. [23] subsequently synonymised *D. microphylla* var. *macropetala* with *D. microphylla* and also made the first attempt to lectotypify these names. In common with many plant species described prior to the type method becoming mandatory worldwide in 1935 [25], the names (except for *D. microphylla*) were described without designating a holotype from amongst the numerous specimens (syntypes) comprising the respective type collections. However, Marchant et al. [23] used the term “isotypes” rather than “lectotypes” for *D. calycina*, *D. calycina* var*. minor*, and *D. microphylla* var. *macropetala*, which accordingly did not comprise a valid lectotypification (being the designation of one gathering among the syntypes as the representative, name-bearing type of a species) per Arts. 7.11 and 9.17 of the International Code of Nomenclature for algae, fungi, and plants (ICN; [26]).

In 1987, Australian naturalist Allen Lowrie noted that flower colour can be used to distinguish at least three taxa in the *D. microphylla* complex, suggesting that further studies might separate plants from near Esperance (ca. 400 km east of Albany) based on their white petals [12]. In contrast, plants from near Albany (the area of the type collection of *D. microphylla*) have orange petals, while populations around Perth were described as having dark red petals [12]. This polymorphism was further investigated by Robert Gibson in 2006 [14], who recognised a number of morphological characters (leaf shape, petal colour, sepal and petal length, and plant colour) to distinguish five potentially separate taxa in the complex. He subsequently concluded that “further taxonomic study into this complex appears warranted, and would likely be most rewarding” [14].

The broad circumscription of *D. microphylla* finally changed in 2014, when Lowrie reinstated Planchon’s *D. calycina* and described the white-flowered taxon from near Esperance as *D. esperensis* Lowrie [1]. He used some of the same differential characters mentioned earlier by Planchon [17] and Gibson [14] (plant colour, leaf shape, and petal colour) but also noted the stamen and style colour as important characters to distinguish these three taxa [1]. All three species were further described as being geographically well-separated, with *D. microphylla* occurring around Albany, *D. calycina* occurring around and to the north of Perth, and *D. esperensis* occurring east of Esperance [1]. While Lowrie [1] listed “holotypes” for *D. calycina*, *D. calycina* var. *minor*, and *D. microphylla* var. *macropetala*, this did not constitute a valid lectotypification as ICN Art. 7.11 requires the phrase “designated here” for typifications made after 1 January 2001 [26].

Field observations by the authors of the present work from 2002 to 2022, as well as observations on social media and the citizen science website iNaturalist (www.inaturalist.org (accessed on 13 January 2023)), indicated that several distinctive additional taxa in the *D. microphylla* complex should be recognised. The photographic records and increased geographic coverage obtained through observations made by citizen scientists were instrumental in bringing about the formal, scientific documentation of two of the species described in the present work and, crucially, revealed the true identities of *D. calycina* var. *minor* and *D. microphylla* var. *macropetala.* The latter taxon, with reference to its floral display, was described as “The most beautiful of the genus *Drosera*” by its first collector James Drummond [18,19], yet no photographs of it existed until a few were posted on Facebook almost 150 years later. This echoes the discovery of *D. magnifica* Rivadavia & Gonella, a large and spectacular South American sundew species that was recognised as new from images posted on Facebook [27].

Four new species and two new combinations at species rank are published and illustrated here based on the examination of both herbarium material and living plants in situ, along with the careful re-evaluation of the morphological characters that reliably distinguish the taxa within this complex. We designate lectotypes and provide clarification for the identity of the previously described infraspecific taxa *D. calycina* var. *minor* and *D. microphylla* var. *macropetala*, both of which are reinstated and elevated to species rank. A new name (replacement name) has had to be provided for *D. calycina* var. *minor* at species rank. This raises the total number of species in the *D. microphylla* complex to nine, six of which are rare and potentially threatened. A detailed identification key, comparative figures, and distribution maps are provided.

## 2. Materials and Methods

Populations of all the taxa from the *Drosera microphylla* complex were studied in situ in Southwest Western Australia from 2002 to 2022 and herbarium specimens (including types) were examined by the authors at B, BM, G, K, M, MEL, PERTH, and W (herbarium acronyms following Index Herbariorum; Thiers, B. M. [updated continuously] https://sweetgum.nybg.org/science/ih/ (accessed on 6 December 2022)). Additional digitised specimen images were obtained and examined from E, FI, KFTA, L, LE, LD, LINN, MPU, NSW, OXF, P, and RSA. New field collections were made for *D. macropetala* (collected by TK under Western Australian flora taking licence FT61000860) and type material was collected for the taxa here described as new to science as *D. reflexa* (collected by GB under scientific licence SW019597), *D. atrata*, and *D. hortiorum* (collected in collaboration with Fred and Jean Hort under Western Australian flora taking licence FT61000255). Scanning electron microscopy (SEM) was carried out to measure seed of *D. hortiorum*, *D. microphylla*, and *D. reflexa* using a TM4000Plus II low-vacuum SEM (Hitachi Co. Ltd., Tokyo, Japan) without sputter coating, using an accelerating voltage of 15 kV, a secondary electron detector, and a 30 Pa vacuum. Macro photographs of seed for all species except *D. koikyennuruff* were taken in situ using 1 mm grid paper for scale. Cultivated material of *D. hortiorum* (originating from the late Allen Lowrie) and *D. reflexa* (originating from the late Phill Mann) was also examined. Measurements and morphological characters were recorded from plants in situ, herbarium material, and cultivated plants. The distribution map was prepared using Google Earth Pro and the DBCA-011 and DBCA-012 datasets published by the Department of Biodiversity, Conservation and Attractions (DBCA).

## 3. Results

### 3.1. **Drosera atrata**
*T.Krueger, A.Fleischm. & G.Bourke*
**sp. nov.** (*Figure 1, Figure 2, Figure 3, Figure 4 and Figure 5*)

**Type:** AUSTRALIA. Western Australia: Badgingarra [precise locality withheld for conservation purposes], upper hillslope, sandy clay with laterite gravel, 26 June 2022, *F. Hort, J. Hort & T. Krueger FH 4506* (holotype PERTH!).

**Figure 1 biology-12-00141-f001:**
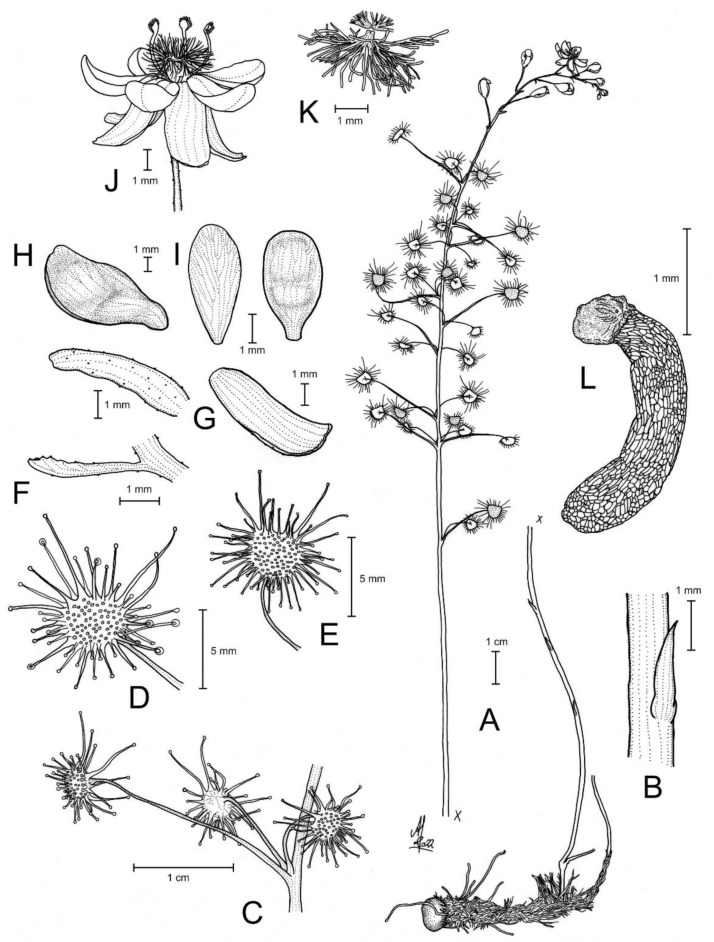
*Drosera atrata* T.Krueger, A.Fleischm. & G.Bourke. (**A**) habit; (**B**) cataphyll from stem base; (**C**) group of leaves from an upper stem internode, comprising one cauline leaf and two axillary leaves; (**D**,**E**) lamina, abaxial surface, (**D**) from cauline leaf, (**E**) from axillary leaf; (**F**) bract; (**G**) sepals, abaxial view (left), adaxial view (right); (**H**,**I**) petals, (**H**) semi-lateral view, (**I**) adaxial view (left example spread, right as in living state); (**J**) flower, lateral view (two stamens removed to reveal the ovary); (**K**) styles, two styles only partially shown; (**L**) seed. (**A**,**B**,**E**,**G**,**I** (left)) from the type (*F. Hort* et al. *FH 4506*), (**C**,**D**,**F**,**H**,**I** (right),**J**–**L**) from in situ photographs. Drawing: A. Fleischmann.

**Figure 2 biology-12-00141-f002:**
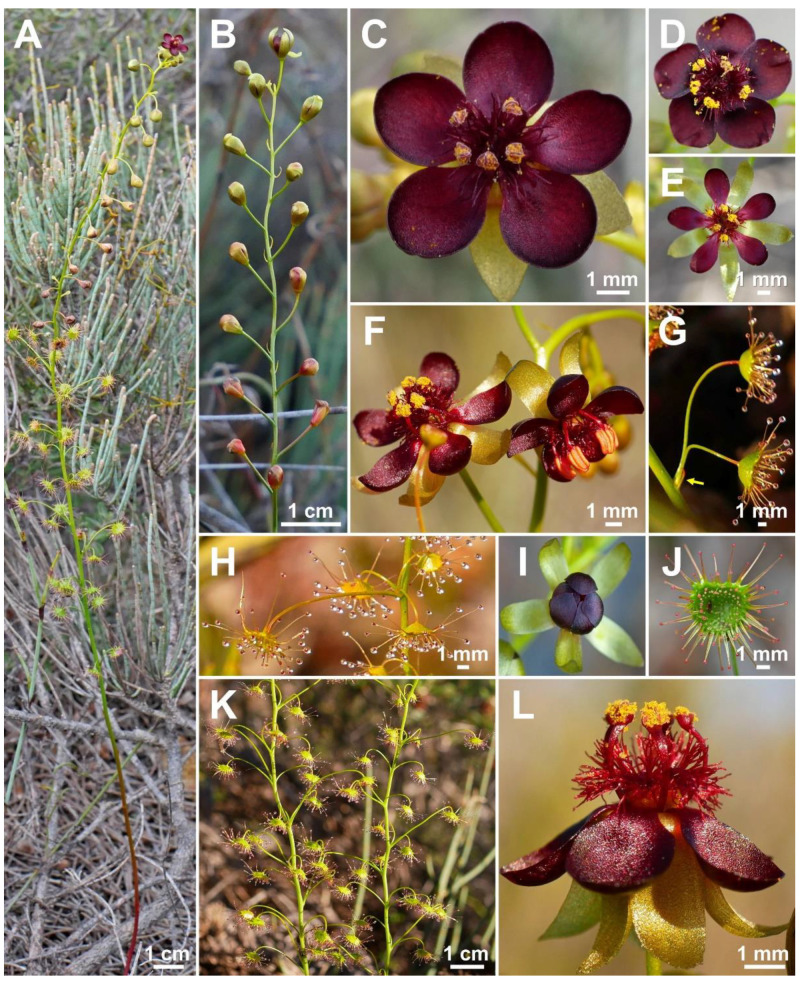
*Drosera atrata* T.Krueger, A.Fleischm. & G.Bourke. (**A**) habit; (**B**) inflorescence; (**C**–**E**) flowers in diffuse light; (**F**) flowers in bright sunlight, note that thecae of anthers are open to present the yellow pollen on the left flower, while in the younger flower at right they are still closed and orange; (**G**) cataphyll (yellow arrow) with two carnivorous axillary leaves; (**H**) cauline leaf with three smaller axillary leaves; (**I**) flower with closed petals in the late afternoon; (**J**) lamina; (**K**) two stems with cauline leaves, note groups of axillary leaves present throughout on all nodes and the downward-facing laminae; (**L**) flower in bright sunlight, lateral view. (**A**,**C**–**E**,**I**,**J**) from Coomallo Nature Reserve, Western Australia, 21 July 2019; (**B**) from near Warradarge, Western Australia, 25 June 2021; (**F**,**K**) from east of Warradarge, Western Australia, 22 May 2022; (**G**,**H**,**L**) from Badgingarra, Western Australia, 25 June 2022. Images: T. Krueger.

**Figure 3 biology-12-00141-f003:**
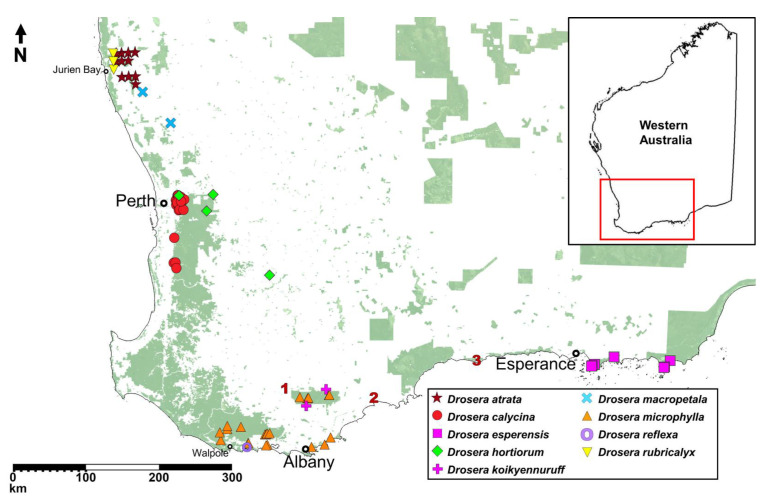
Map showing the known localities of all nine species of the *Drosera microphylla* complex based on herbarium records and field observations by the authors. Locality coordinates of species recommended for inclusion on the Priority Flora List under the Conservation Codes for Western Australian Flora and Fauna have been generalised to the nearest 0.1 degrees. Numbers 1–3 indicate localities of potential undescribed taxa discussed in the Taxonomic notes sections of *D. koikyennuruff* and *D. microphylla*. Background map illustrates protected conservation lands managed by the Department of Biodiversity, Conservation and Attractions (DBCA).

**Figure 4 biology-12-00141-f004:**
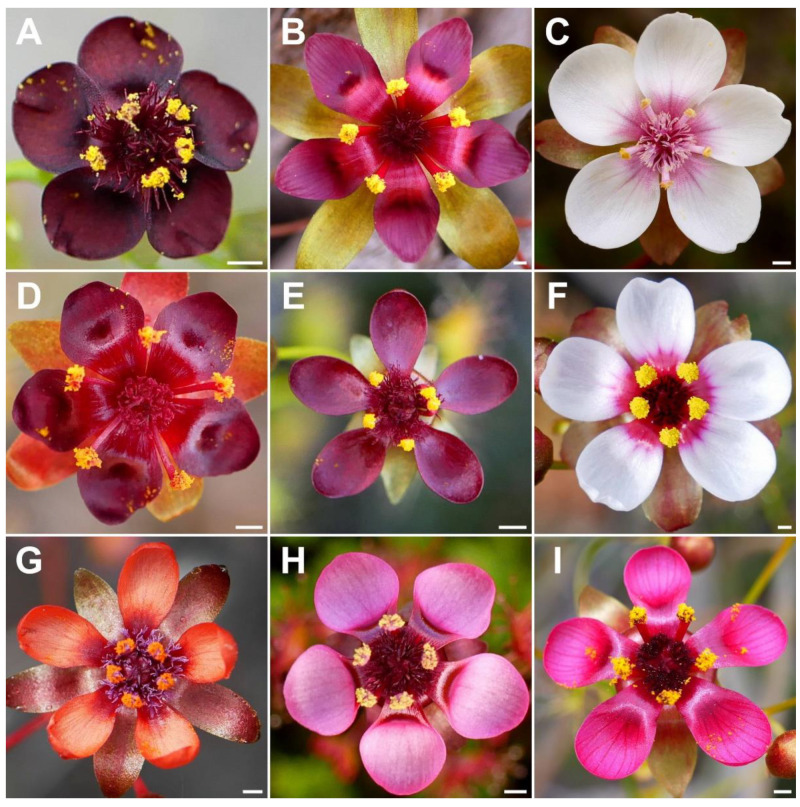
Comparison of the flowers of all nine species of the *Drosera microphylla* complex. (**A**) *D. atrata*; (**B**) *D. calycina*; (**C**) *D. esperensis*; (**D**) *D. hortiorum*; (**E**) *D. koikyennuruff*; (**F**) *D. macropetala*; (**G**) *D. microphylla*; (**H**) *D. reflexa*; and (**I**) *D. rubricalyx*. Scale bars = 1 mm. Images: T. Krueger.

**Figure 5 biology-12-00141-f005:**
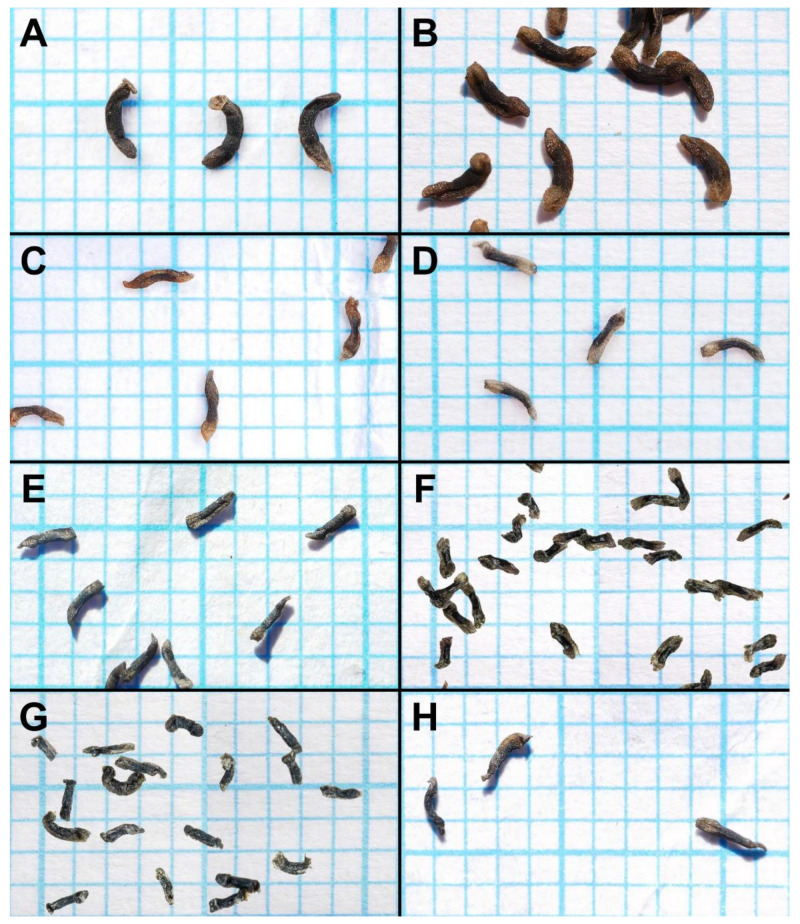
Seed comparison of eight species of the *Drosera microphylla* complex. Seed is placed on 1 mm grid paper. (**A**) *D. atrata*; (**B**) *D. calycina*; (**C**) *D. esperensis*; (**D**) *D. hortiorum*; (**E**) *D. macropetala*; (**F**) *D. microphylla*; (**G**) *D. reflexa*; and (**H**) *D. rubricalyx*. (**A**) from Coomallo Nature Reserve, Western Australia, 28 August 2021; (**B**) from Roleystone, Western Australia, 7 November 2022; (**C**) from Cape Le Grand, Western Australia, 25 November 2021; (**D**) from near York, Western Australia, 14 October 2022; (**E**) from near Dandaragan, Western Australia; (**F**) from near Walpole, Western Australia; (**G**) from near Kentdale, Western Australia; and (**H**) from near Jurien Bay, Western Australia. (**A**–**E**,**H**) by T. Krueger; (**F**,**G**) by G. Bourke, seed digitally superimposed onto grid paper.

**Diagnosis:** *Drosera atrata* differs from all other species of the *D. microphylla* Endl. complex by (contrasting characters in parentheses) (1) its leaf arrangement, with all leaves in groups of 2–5 per node due to the presence of 1–4 slightly shorter axillary leaves in the axils of all cauline leaves (cauline leaves solitary or axillary leaves only found in the upper 1–9 nodes of the stem with the lower 4–15 cauline leaves being solitary); (2) its many-flowered inflorescences typically producing 5–23 flowers per scape (1–8 flowers per scape); and (3) its styles, which mainly branch close to their base into entire or sparsely branched style segments (styles branching near their base and style segments additionally strongly divided). It is further distinguished by its falcate to allantoid seeds (very narrowly obconic, narrowly clavate to acerose seeds, except in *D. calycina* Planch., which has similar but less falcate seed) and its very dark red to blackish-red petals (a similar, but slightly brighter petal colour is also found in *D. hortiorum* T.Krueger & G.Bourke and in *D. koikyennuruff* T.Krueger & A.S.Rob., with the remainder of the species having very different petal colours).

**Description:** Tuberous perennial herb, (14–)17–44 cm tall above ground including inflorescence. **Tuber** subglobose, ca. 10 mm in diameter, enclosed in black papery sheaths from previous seasons’ growth. **Stem** (subterranean part) ca. 5 cm long, 3–6 mm in diameter, enclosed in brown, fibrous tunic formed from previous seasons’ stems and roots. **Roots** few, fibrous, emerging laterally from along subterranean part of stem, mostly immediately above tuber. **Stem** (epigeous part) erect, self-supporting, simple, terete, straight, or rarely slightly fractiflex (zig-zag-shaped), glabrous, (10–)12–29 cm tall, 1.2–2.0 mm in diameter near soil surface, 0.7–1.2 mm in diameter at internodes, yellowish green but always reddish orange to red near soil level; sometimes 2–4 stems emerging from the same tuber. **Cataphylls** (often erroneously termed “prophylls” in tuberous *Drosera*) 5–11 on lower part of stem, subulate, 1.7–5.0(–9.0) mm long, ca. 0.5 mm wide, red to orangey yellow or yellowish green with red apex, uppermost 1–4 cataphylls often supporting (1–)2(–3) carnivorous axillary leaves. **Leaves** in groups of 2–5 per node, due to (1–)2(–4) slightly shorter axillary leaves emerging from the axils of all cauline leaves (only rarely lowermost or uppermost 1–2 cauline leaves solitary); internodes (1–)5–20(–24) mm and (1–)3–12 nodes bearing leaves (foliose nodes) present in flowering individuals. **Petioles** terete, semi-erect or horizontal, arcuated abaxially (downwards) with arching usually increasing gradually towards tip, sometimes only arcuated near tip, glabrous, (10–)12–33 mm long, 0.5–1.0 mm wide at base, tapering to 0.2–0.3 mm towards lamina, yellowish green, tip often tinged yellowish pink to orangey yellow. **Lamina** peltate, orbiculate with flattened adaxial lateral margin or reniform, shallowly concave, adaxial surface mostly facing downwards, 2.1–4.0 mm long, 2.3–5.0 mm wide; lamina adaxial surface covered with stalked, carnivorous, secretive capitate glands (tentacles); tentacles 2–6 mm long at lamina margin, decreasing in size towards centre of lamina, with red to greenish yellow stalk; lamina abaxial surface minutely sparsely punctate. **Petioles of axillary leaves** terete, semi-erect, arcuated downwards with arching usually increasing gradually towards tip, glabrous, 5–17(–25) mm long, 0.3–0.5 mm wide at base, tapering to 0.1–0.3 mm towards lamina, yellowish green. **Lamina of axillary leaves** of same shape as the lamina described above, 2.0–3.5 mm long, 2.0–3.8 mm wide. **Inflorescence** a (1–)5–23-flowered scorpioid cyme, terminal, simple or rarely branched, sometimes 1–2 lateral scapes emerging from axils of uppermost leaves, single-sided, (4.5–)5.0–11.0(–15.0) cm long. **Peduncle** terete, 0.4–3.2 cm long, (0.4–)0.6–1.0 mm in diameter, microscopically glandular (appearing glabrous), yellowish green, rarely red. **Pedicels** terete, semi-erect or horizontal in fruit, (3–)4–11(–14) mm long in fruit, 0.3–0.6 mm in diameter, spaced by 2–10 mm along rhachis, microscopically glandular (appearing glabrous), yellowish green. **Bracts** spathulate, narrowly spathulate, narrowly obovate or subulate, often arcuated adaxially (upwards) but not concave, apex entire or irregularly crenulate, 1.5–4.0(–5.0) mm long, 0.3–0.9(–1.7) mm wide, abaxial surface microscopically glandular. **Sepals** 5, narrowly elliptic to narrowly obovate, arcuated adaxially (upwards), slightly concave, often reflexed during anthesis, lateral and/or apical margins sometimes shallowly involute, apex entire, truncate, emarginate or crenulate, 4–6 mm long, 1.5–2.5 mm wide, abaxial surface microscopically glandular, yellowish brown to yellowish green, minute black spots often apparent. **Corolla** 6–9 mm in diameter. **Petals** 5, very dark red to blackish red, minutely punctate with black spots, obovate, broadly obovate, spathulate or broadly spathulate, deeply concave and slightly arcuated adaxially (upwards), apex rounded and entire, 3.0–4.5 mm long, 1.9–2.9 mm wide. **Stamens** 5, 2.0–3.0 mm long. **Filaments** ± linear or only very slightly dilated towards apex, straight or slightly falcate, 0.2–0.3 mm wide, deep red. **Anthers** bithecate, retrorse, 0.5–0.9 mm wide, thecae orange. **Pollen** yellow. **Ovary** obovoid, 3-carpellate, fused, 0.8–1.4 mm in diameter, yellowish green to yellowish brown. **Styles** 3, divided into many filiform segments just above the base, style segments entire or sparsely branched, terete, filiform, extending laterally beyond filaments, 1.4–2.4 mm long, red to dark red. **Stigmas** simple, at tips of style segments, surface appearing smooth, red. **Seeds** falcate to allantoid, flattened, apices obtuse, with funicular base usually present as disc-like appendage, 2.3–2.6 mm long, 0.5–0.6 mm wide, testa black-brown with funicular disc pale brown (sometimes chalazal end also pale brown); testa more or less longitudinally reticulate, with anticlines thin and only shallowly raised.

**Etymology:** The specific epithet is derived from the Latin *atratus* (=blackened) and refers to the very dark red to blackish red flower colour of this species, which is the darkest petal colour known in the genus *Drosera* (under some lighting conditions appearing almost black).

**Taxonomic notes:** *Drosera atrata* is arguably the most morphologically distinct species within the *D. microphylla* complex, exhibiting a unique leaf and inflorescence morphology not found in any other species in this group. While the presence of axillary leaves in the axils of all cauline leaves (thus having groups (sometimes incorrectly called “whorls”) of 2–5 leaves at each node) is paralleled in other species of *D.* section *Ergaleium* (e.g., in *D. macrantha* Endl. and *D. menziesii* R.Br. ex DC.), its occurrence in *D. atrata* is unique within the *D. microphylla* complex. *Drosera hortiorum*, *D. macropetala*, and *D. rubricalyx* feature similar (but relatively shorter-petioled) axillary leaves in their uppermost nodes, but, in these species, they are never present in all nodes. In *D. atrata*, carnivorous axillary leaves are frequently found even in association with the uppermost cataphylls (Figure 1G).

The number of flowers (5–23) per inflorescence in *D. atrata* is 2–3 times greater than in any other species of the *D. microphylla* complex. The inflorescence sometimes equals or exceeds the foliose part of the stem in length (Figure 1A). In plants with compound inflorescences (lateral flower scapes emerging from the axils of the uppermost leaves), the total number of flowers per plant may exceed 40. Despite its relatively large height of up to 44 cm, *D. atrata* produces the smallest flowers of the *D. microphylla* complex, the corolla measuring just 6–9 mm in diameter. Similarly small flowers are occasionally found in very small individuals of *D. hortiorum*, *D. koikyennuruff*, *D. microphylla*, and *D. reflexa*. In contrast with all other members of the *D. microphylla* complex, the styles of *D. atrata* mainly branch shortly above the base; the style segments themselves are entire or only very sparsely branched.

The petal colour of *Drosera atrata* is the darkest within the genus with only the tropical rainforest species of north-eastern Australia (*D.* section *Prolifera* C.T.White) producing similarly dark red or dark-pink flowers, in particular certain forms of *D. adelae* F.Muell. While some Australian pygmy sundews (*D.* section *Bryastrum* Planch.) and several members of the South African *D.* section *Ptycnostigma* Planch. produce almost completely black petal bases, these are always paired with a relatively bright colour that comprises the largest part of the petal [28]. In contrast, the dark colour of *D. atrata* is relatively uniform across the entire petal. However, this colour often appears even darker (almost black) when viewed at certain angles. Given the deeply concave petal shape, this often results in the appearance of especially dark areas near the petal margins (Figure 1C,E,F,L) or bases (Figure 1D; here, the strongly reflexed and deeply concave petals result in the bases being viewed at an angle while the margins are viewed ± perpendicular).

The seeds of *D. atrata* differ from all other species of this affinity except *D. calycina* by their falcate to allantoid shape (narrowly obovate, narrowly clavate, or narrowly obtrullate with truncate upper end (=nail- or pin-shaped) and more or less terete in the remainder of species from the *D. microphylla* complex), often with the funiculus still attached to the funicular seed end as a pale brown discus (Figure 5). Only seeds of *D. calycina* are somewhat more similar to those of *D. atrata* in being slightly falcate and flattened. Seed morphology has already been shown to be a reliable taxonomic tool for species delimitation in some other species complexes of tuberous *Drosera* [1].

**Distribution and habitat:** *Drosera atrata* is known from eleven locations between Warradarge in the north and Badgingarra in the south (Figure 3). It occurs in low kwongan heath on the upper slopes of lateritic hills in poorly drained sandy clay with laterite.

**Phenology:** Flowering has been recorded from May to August.

**Conservation status:** Recommended for listing as Priority Three (poorly known species) under Conservation Codes for Western Australian Flora and Fauna (Western Australian Herbarium 1998–, https://florabase.dpaw.wa.gov.au/ (accessed on 6 December 2022)). It is assessed as Vulnerable (VU) under IUCN criterion D1 following IUCN [29]. The populations of *D. atrata* frequently comprise extremely small population sizes of just 1–30 plants. Only a single larger population of ca. 200 plants is known from an unprotected road reserve near Warradarge. Six of the eleven known locations occur on land managed by the Western Australian Department of Biodiversity, Conservation and Attractions (DBCA). Unlicenced collectors and illegal commercial/horticultural trade could pose a threat to *D. atrata* in the future given its extremely small population sizes and the tendency for poachers to target rare carnivorous plant species to supply a demand driven by the horticultural market and carnivorous plant collectors in particular [16]. Further surveys are recommended to gain a better understanding of this taxon’s biology, distribution, number and size of populations, and to identify additional potential threats.

**Additional specimens examined (paratypes):** AUSTRALIA. Western Australia: Coomallo Nature Reserve [precise locality withheld for conservation purposes], breakaway, brown dry ironstone gravel, 24 July 2011*, J.E. Wajon 2435* (PERTH 09050876!); Warradarge [precise locality withheld for conservation purposes], in lateritic sandy soil on lower slope, 16 August 2018, *J. Keeble JK 73* (PERTH 09189807!); Badgingarra [precise locality withheld for conservation purposes], laterite hill top, sand and pebbles, 16 June 2022, *F. Hort & J. Hort FH 4502* (PERTH 09482849!); Badgingarra [precise locality withheld for conservation purposes], white sand with coarse pebbles, 16 June 2022, *F. Hort & J. Hort FH 4499* (PERTH 09482717!); Badgingarra [precise locality withheld for conservation purposes], sand and laterite rise mid slopes, 16 June 2022, *F. Hort & J. Hort FH 4503* (PERTH 09482806!); Badgingarra [precise locality withheld for conservation purposes], small breakaway, weathered stone, clay, sand, gravel, 26 June 2022, *F. Hort & J. Hort FH 4509* (PERTH 09482636!); Badgingarra [precise locality withheld for conservation purposes], upslope from shallow breakaway: sand gravel, 27 June 2022, *F. Hort & J. Hort FH 4514* (PERTH 09482768!); Badgingarra [precise locality withheld for conservation purposes], hill top grey sand with laterite rubble/gravel, 27 June 2022, *F. Hort & J. Hort FH 4513* (PERTH 09482679!).

**Additional localities examined:** Brand Highway, Badgingarra [precise locality withheld for conservation purposes], mid slope of laterite hill, June 2003, G. Bourke pers. obs.; Lesueur National Park [precise locality withheld for conservation purposes], upper slopes of laterite hill, 11 August 2022, T. Krueger pers. obs.; Cataby [precise locality withheld for conservation purposes], upper slopes of laterite hill, 26 June 2022, T. Krueger pers. obs.

### 3.2. **Drosera calycina**
*Planch., Ann. Sci. Nat., Bot., sér. 3, 9: 299 (1848). (Figure 3, Figure 4, Figure 5 and Figure 6)*

**Lectotype (designated here):** [AUSTRALIA. Western Australia:] Swan River, without date [likely part of the Drummond I collection, hence a collection date between 1839 and 1841 is probable [30]], *J. Drummond n. 1* (left individual of K000215039! isolectotype: right individual of K000215039! [both individuals mounted on the same sheet as K000215091 (not a type)]).

**Figure 6 biology-12-00141-f006:**
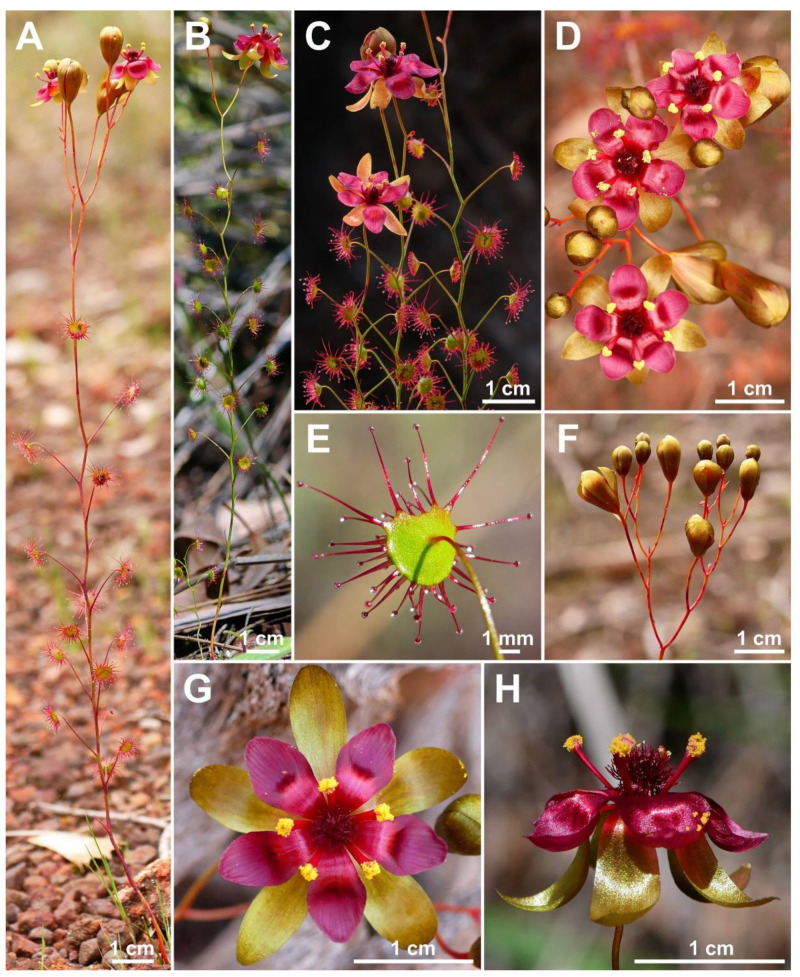
*Drosera calycina* Planch. (**A**–**C**) habit; (**D**) flowers in diffuse light; (**E**) lamina, abaxial view; (**F**) branched inflorescence; (**G**) flower in diffuse light; (**H**) flower in bright sunlight, lateral view. (**A**–**D**,**F**–**H**) from John Forrest National Park, Western Australia, 8 September 2022; (**E**) from Roleystone, Western Australia, 11 July 2021. Images: T. Krueger.

**Description:** Tuberous perennial herb, (8–)14–37(–42) cm tall above ground including inflorescence. **Tuber** subglobose, 6–8 mm in diameter, enclosed in black papery sheaths from previous seasons’ growth. **Stem** (subterranean part) 3.5–8.0 cm long, 1.5–3.0 mm in diameter, enclosed in brown, fibrous tunic formed from previous seasons’ stems and roots. **Roots** few, fibrous, emerging laterally from along subterranean part of stem, mostly immediately above tuber. **Stem** (epigeous part) erect, self-supporting, simple, terete, slightly to strongly fractiflex, glabrous, (9–)11–35(–39) cm tall, (0.4–)0.7–1.8 mm in diameter near soil surface, 0.5–1.2 mm in diameter at internodes, yellowish green, often irregularly blotched with red, always orange to red near soil level; sometimes 2–6 stems emerging from the same tuber. **Cataphylls** subulate, 4–9(–12) present on lower part of stem, 1–5 mm long, ca. 0.5 mm wide, red to orangey yellow. **Leaves** solitary on each node, alternate, (12–)15–30 present in flowering individuals; internodes (1–)3–17(–26) mm. **Petioles** terete, semi-erect, mostly ± straight or very slightly arcuated abaxially (downwards), strongly arcuated abaxially near tip, glabrous, 9–25(–30) mm long, (0.3–)0.5–0.8 mm wide at base, tapering to 0.1–0.2 mm towards the lamina, yellowish green, often irregularly blotched with red, base yellowish green to red, tip orangey yellow to red. **Lamina** peltate, reniform, or orbiculate with flattened, often truncated adaxial lateral (upper) margin, shallowly concave, adaxial surface facing outwards or slightly downwards, 2.2–3.6 mm long, 2.4–4.1 mm wide; lamina adaxial surface covered with stalked, carnivorous, secretive capitate glands (tentacles); tentacles 2–7 mm long at lamina margin, decreasing in size towards centre of lamina, with red stalk; lamina abaxial surface glabrous. **Inflorescence** a 1–9-flowered scorpioid cyme, terminal, simple or branched, sometimes with an additional scape emerging from axil of uppermost leaf, single-sided, (2.8–)4.0–9.0 cm long. **Peduncle** terete, 1.3–4.0 cm long, 0.4–0.8 mm in diameter, microscopically glandular (appearing glabrous), yellowish green, sometimes blotched with red. **Pedicels** terete, erect in fruit, (6–)9–26(–31) mm long in fruit, 0.3–0.7 mm in diameter, spaced by 3–17 mm along rhachis, microscopically glandular (appearing glabrous), yellowish green to orangey yellow in lower half, reddish orange to red in upper half, uppermost pedicels often completely tinged red. **Bracts** spathulate, narrowly obovate, elliptic, or subulate, often concave and arcuated adaxially (upwards), apex entire or irregularly crenulate, 1.4–3.5 mm long, 0.3–1.2 mm wide, glabrous. **Sepals** 5, narrowly elliptic to narrowly obovate, arcuated adaxially (upwards), slightly concave, often reflexed during anthesis, apex entire or crenulate, 8–13 mm long, 2.8–5.0 mm wide, abaxial surface microscopically glandular, yellowish brown to yellowish green, often with 3–5 red veins, minute black spots sometimes apparent. **Corolla** 14–20 mm in diameter. **Petals** 5, deep red in inner half transitioning to purplish red in outer half, deep red veins often apparent, obovate, deeply concave and slightly arcuated adaxially (upwards), apex rounded and entire, 6.5–9.5 mm long, 4.5–6.0 mm wide. **Stamens** 5, 4.0–6.5 mm long. **Filaments** dilated towards apex, 0.3–0.5 mm wide at base, 0.5–1.1 mm wide near apex, deep red (sometimes purplish red in upper half). **Anthers** bithecate, retrorse, 1.0–1.4 mm wide. **Pollen** yellow. **Ovary** obovoid, 3-carpellate, fused, 1.3–1.7 mm in diameter, deep red. **Styles** 3, divided into a few filiform segments just above the base, style segments again divided into many terete style segments, forming a crowded tuft, not extending laterally beyond filaments, 1.3–1.9 mm long, dark red. **Stigmas** simple, at tips of style segments, papillose, ca. 0.2 mm long, deep red to dark red. **Seeds** slightly falcate to slightly allantoid, flattened, funicular (upper) apex truncate to obtuse, chalazal (lower) end tapering to obtuse apex, 2.4–2.7 mm long, 0.5–0.7 mm wide, testa dark brown with chalazal and funicular ends pale brown); testa longitudinally reticulate, with anticlines thin and only shallowly raised.

**Etymology:** The specific epithet is derived from the Latin *calycinus* (having a well-developed calyx) and was selected by Planchon [17] to refer to the very large sepals/calyx of this species.

**Taxonomic notes:** *Drosera calycina* can easily be distinguished from the remainder of the *D. microphylla* complex (especially in herbarium material) by the combination of ± straight petioles (which are only arched near the tip; Figure 6C,E) with the absence of any axillary leaves. It is morphologically similar to *D. microphylla*, *D. hortiorum*, *D. macropetala*, and *D. rubricalyx*. It can be distinguished from *D. microphylla* by (contrasting characters in parentheses) (1) its lamina shape, which is reniform or orbiculate with flattened, often truncated upper margin (lamina orbiculate or sometimes orbiculate with very slightly flattened upper margin); (2) its comparatively large flowers with a corolla diameter of 14–20 mm (corolla diameter 8–15 mm); (3) its petal colour, which is deep red in inner half transitioning to purplish red in outer half (petals reddish orange with deep red bases); (4) its stamen length, which reaches 4.0–6.5 mm (stamens 2.5–3.5 mm long); and (5) its styles, which do not extend laterally beyond the filaments and have deep red stigmas (styles laterally extending beyond the filaments with reddish purple stigmas).

*Drosera calycina* is further distinguished from *D. hortiorum*, *D. macropetala*, and *D. rubricalyx* by (contrasting characters in parentheses): (1) its solitary leaves (leaves of upper 1–9 nodes in groups of 2–5 due to the presence of usually two shorter axillary leaves); (2) its lamina shape, which is reniform or orbiculate with flattened, often truncated upper margin (lamina orbiculate or orbiculate with slightly flattened upper margin); and (3) its petal colour, which is deep red in inner half transitioning to purplish red in outer half (petals deep red in inner half, dark purplish red in outer half in *D. hortiorum*; white with deep red bases in *D. macropetala*; or deep red in inner half, deep pink in outer half in *D. rubricalyx*; Figure 4). The four species are also ecologically and geographically well separated (Figure 3). While *D. calycina* has been observed growing within a few hundred metres of *D. hortiorum* near Glen Forrest, just east of Perth, they likely do not co-occur due to their different habitat requirements. In this area, *Drosera calycina* is restricted to laterite soils in Jarrah forests while *D. hortiorum* grows in clay loam around granite slopes and boulders.

The distinctive lamina shape of *Drosera calycina* (Figure 6E) has been described as “subtruncate” (Planchon, in annot. K000215039), “suborbiculate-lunate” by Planchon [17], or “crescent-shaped and/or broadly reniform” by Lowrie [1] (the latter likely included both *D. hortiorum* and *D. rubricalyx* in his description of *D. calycina*). In all other species but *D. atrata*, the lamina is usually entirely orbiculate or orbiculate with a slightly flattened upper margin but not truncated. Only *D. atrata* also often produces reniform or truncated laminae but that species is readily distinguished by the presence of axillary leaves in all nodes, its sparsely branching styles extending laterally beyond the filaments, and its very dark red to blackish-red petals (Figure 4). *Drosera atrata* and *D. calycina* additionally share a falcate to allantoid (i.e., slightly depressed and curved) seed shape, which is another taxonomically informative character to distinguish these two species from the remainder of the *D. microphylla* complex, which mostly have straight, pin-shaped, or bone-shaped seeds (Figure 5).

*Drosera calycina* was previously illustrated by Erickson [22] (p. 40, drawing 2) as “*D. microphylla* var. *macropetala*”. While both *D. calycina* and *D. macropetala* indeed are very tall, large-flowered plants, the petals of *D. calycina* are never “drying palish” as stated by Erickson [22] (likely based on Diels’ [8] description of *D. microphylla* var. *macropetala*). Indeed, both of Erickson’s specimens in the Western Australian Herbarium (PERTH 00666416!, PERTH 00666874!) are *D. calycina* and it seems unlikely that she observed *D. macropetala* during her studies.

The illustration of *D. calycina* provided by Lowrie in 2014 [1] (p. 355; he previously published a similar illustration as *D. microphylla* in his 1987 book [12] (p. 65)) is evidently based on several different specimens. While most of the illustration matches the cited specimen *A. Lowrie 3043* (PERTH 08988110!, MEL 2443236A!), the presence of axillary leaves on the habit drawing A is puzzling. Crucially, none of the individuals of the *A. Lowrie 3043* collection feature axillary leaves. The specimens also lack tubers, in contrast with the illustration. It is therefore possible that Lowrie’s illustration also incorporates specimens of either *D. hortiorum*, *D. macropetala*, or *D. rubricalyx*, all of which have axillary leaves in their upper parts, i.e., the author seems to have used some artistic licence.

**Distribution and habitat:** Darling Scarp (westernmost part of the Darling Range) between Gidgegannup and Dwellingup (Figure 3). Grows in Jarrah forest mixed with *Banksia sessilis* (Knight) A.R.Mast & K.R.Thiele (Proteaceae), usually on slightly sloping hillsides and hilltops high up on the Darling Scarp. The soils are usually sandy clay with laterite gravel.

**Phenology:** Flowering has been recorded from August and September.

**Conservation status:** Not eligible for Conservation Code listing and Least Concern (LC) following Cross [31]. *Drosera calycina* is relatively common in its preferred Jarrah forest habitat and at least twenty localities have been recorded, most of which are on land managed by the Western Australian Department for Biodiversity, Conservation and Attractions (DBCA). While frequently occurring in small population sizes of <50 plants, at least two large populations of >200 plants are known to exist. Unlicensed collection by plant collectors may represent a threatening process but regular monitoring of several populations between 2019 and 2022 indicated there are no current threats to this taxon (T. Krueger pers. obs.).

**Notes on the lectotypification:** Lectotypification of *D. calycina* is required as Planchon [17] did not select as the type a single specimen out of Drummond’s gathering (*J. Drummond n. 1*). He cited two duplicates that he had studied (constituting syntypes), namely “*Drummond* in herb. Hook. et Soc. Linn. Londres” [17] (p. 299), which is the specimen from Herbarium Hookerianum (K000215039) and the one from the Linnean Society of London (Herbarium LINN, some specimens have been transferred to BM [32], and some apparently also to K, see below). However, the second specimen could not be found either at LINN or at BM and it may indeed be lost. However, it is also possible that this second specimen was transferred to K when the “Herbarium Australiense” specimens of the Linnean Society of London herbarium were included in the Kew collections in 1915, including Drummond material (”Herbarium Australiense, presented by the Linnean Society, 1915”; Anonymous in annot. K000843361 photo!). As no other matching specimen could be found at K, this would likely mean that this second specimen was added to the same sheet, which is now K000215039, and that the two plants represented there are indeed the two syntype specimens cited by Planchon, housed at different herbaria at the time. While it might seem counterintuitive to combine specimens this way, the practice was not uncommon at that time and another Drummond collection, *J. Drummond n. 282*, which belongs to the Drummond V collection and represents a different species, *D. microphylla*, was even added to this same sheet at a later date.

These two specimens of *J. Drummond n. 1.* cited by Planchon [17] by definition constitute syntypes, hence lectotypification is required (ICN Arts. 7.11 and 9.17 [26]), even if both are found mounted together on the same herbarium sheet today. K000215039 holds a handwritten personal annotation by Planchon, which represents the sketch of a differential diagnosis noted by the author: “*Drosera calycina* Planch. nov. sp.; Folia nunc subtruncatam. Droserae filicauli Endl. affinis sed petala violacea, et sepala eciliata [leaves now subtruncate. Related to *Drosera filicaulis* Endl. [*D. menziesii*] but with violet petals and hairless sepals]”. It is thus clearly evident that K000215039 is original material of *D. calycina*; what cannot be determined is which of the two individuals corresponds to the specimen from Hooker’s herbarium, and which may originate from LINN. Accordingly, we have selected the more complete individual as the lectotype, that is, the individual to the left that includes open flowers (the other specimen only has flowers in bud).

Marchant et al. [23] unnecessarily selected an “isotype” at K for *D. calycina*, referring to Planchon’s type. In addition, Marchant incorrectly annotated a different specimen at Montpellier Herbarium (MPU1254140) as the “holotype” (Marchant 1985 in sched.) but this was never effectively published (as required by ICN Art 7.10 [26]). Choosing MPU1254140 would have been an incorrect type designation in any case because this specimen is not original material of *Drosera calycina*. It was not cited by Planchon in 1848 [17] and it was not ascribed to the name *D. calycina* by Planchon himself (evident from the label on MPU1254140 in Planchon’s hand, which reads, “*Drosera calycina* ? Planch.”), and it was annotated by Planchon after he had described *D. calycina* in 1848 (Planchon became assistant professor at Montpellier in 1853 and director of MPU in 1881, while from 1844–1848 he was based at Kew [32]). Thus, MPU1254140 cannot constitute a type for *D. calycina* and, in fact, represents a different species (likely *D. rubricalyx*, see “Notes on Drummond’s type collection” under *D. rubricalyx*). Lowrie [1], simply referring to Marchant et al. [23], also incorrectly lists the MPU specimen as the “holotype”.

It should also be noted that *J. Drummond n. 1* was likely the only gathering of the *D. microphylla* complex collected by James Drummond that was available to Planchon in 1848 for his revision of Droseraceae. The other specimens *J. Drummond coll. V n. 282*, *J. Drummond coll. VI n. 109*, and *J. Drummond coll. VI n. 110* were collected later (collected in 1847 or 1848 and dispatched to Europe in 1849 for coll. V; and collected in 1850 or 1851 for coll. VI [30]) and, thus, were not considered by Planchon in his 1848 taxonomic treatment of *Drosera* [17].

**Notes on Drummond’s type collection:** Unfortunately, neither the type material nor Drummond’s scarce publication records provide any evidence for where exactly in the former Swan River colony the type collection was made. The contemporary botanist Diels [33] (p. 50, literally translated) has already asserted that, “in short, one will never know [exactly] where Drummond’s plants were collected; and just in rare cases it can be achieved by the aid of literature to pinpoint at least the approximate habitat”. Two of these cases, for which the authors of the present work could trace back the *locus classicus* from Drummond’s historic notes [18,19], are *D. macropetala* and *D. rubricalyx* (see “Notes on Drummond’s type collection” under the headings of the two respective species).

**Additional specimens examined:** AUSTRALIA. Western Australia: Bellevue, Darling Range, *C.P. Conigrave s.n.* (E00794030 photo!); Darlington, Darling Range, soil of ironstone gravel, 6 September 1900, *A. Morrison 1238* (PERTH 666920!, E00138677 photo!); Gooseberry Hill, Darling Range, E of Perth, September 1908, *C. Andrews s.n.* (PERTH 666440!); Kalamunda, 1 September 1913, *W.B. Alexander s.n.* (PERTH 666467!); Mundaring Weir, 1 September 1945, *C.D. Hamilton 43* (PERTH 666394!); Pomeroy Road near Welshpool Road, Bickley, laterite soil, 16 August 1965, *N.G. Marchant 6585* (PERTH 666882!); Lesmurdie, near junction of Welshpool and Pomeroy roads, gravelly loam, 16 August 1965, *N.G. Marchant s.n.* (PERTH 666890!); Gooseberry Hill, S of The Knoll near Perth, 28 August 1965, *A.C. Beauglehole ACB 12338* (PERTH 666386!); Gooseberry Hill, Darling Range, E of Perth, 28 August 1965, *R. Erickson s.n.* (PERTH 666874!); Gooseberry Hill, Darling Range, E of Perth, 5 September 1965, *R. Erickson s.n.* (PERTH 666416!); At the intersection of Mundaring Weir Road and Spring Road, Gooseberry Hill, in jarrah forest, 24 August 1974, *S. Carlquist 5398* (RSA0229906 photo!); At the corner of Spring Rd. and Mundaring Weir Rd. in Kalamunda. On the Darling Scarp, Growing in laterite soil with some sand mixture in Eucalyptus forest, 14 September 1974, *L. Debuhr 3606* (RSA0229907 photo!); On Gooseberry Hill, Darling Scarp, 15 September 1974, *S. Carlquist 5631* (RSA0229905 photo!); Junction of Canning Mills Road and Canning Road, Kalamunda, 25 km E of Perth, lateritic sand, 3 September 1984, *G.J. Keighery 7370* (PERTH 5863031!, CANB 363020.1); NE side of minor track in Park Forest Block,W of Stawell Road—Waroona Road intersection, Quadrat P7/1, on black gravel soil, 19 September 1994, *K. McDougall 414* (PERTH 6141110!); Site 46, ca 4 km W of Teesdale Hill, bearing NE, upland, very disturbed, soil surface: littered, gravelly, soil colour: dark brown, soil texture: sandy loam, 4 September 1997, *A. Gundry 1309* (PERTH 4828135!); Between Kalamunda and Mundaring Weir on Mundaring Weir Road, hillside, brown lateritic loam, dense litter cover, 5 September 2001, *K. Macey 380* (PERTH 5910048!); Bodhinyana Monastery, 216 Kingsbury Drive, Serpentine, topography: plain and ridge, soil colour: brown, soil: ironstone gravel, 7 September 2002, *B. Nyanatusita 140* (PERTH 681050!); Pinjarra—Dwellingup Road, grows in laterite-loam soils, 10 September 2004, *A. Lowrie 3043* (PERTH 8988110!, MEL 2443236A!); Beelu National Park, off of Moola road, ironstone gravels, Jarrah woodland with open shrub and sedge/grass understory, 28 August 2019, *D.E. Murfet & A. Lowrie 9406* (MEL 2477153A); West. Australia, without date, *C.A. Gardner 9584* (L.1858657 photo!); without locality, without date, without collector (PERTH 666424!); without locality, without date, without collector (PERTH 666432!).

### 3.3. **Drosera esperensis**
*Lowrie, Carniv. Pl. Austral. Magnum Opus 3: 1270 (2014). (Figure 3, Figure 4, Figure 5 and Figure 7)*

**Type:** AUSTRALIA. Western Australia: Cape Le Grand, E of Esperance, 31 August 2000, *A. Lowrie 2566* (holotype PERTH 08988307 photo!; isotype MEL 2457584!).

**Figure 7 biology-12-00141-f007:**
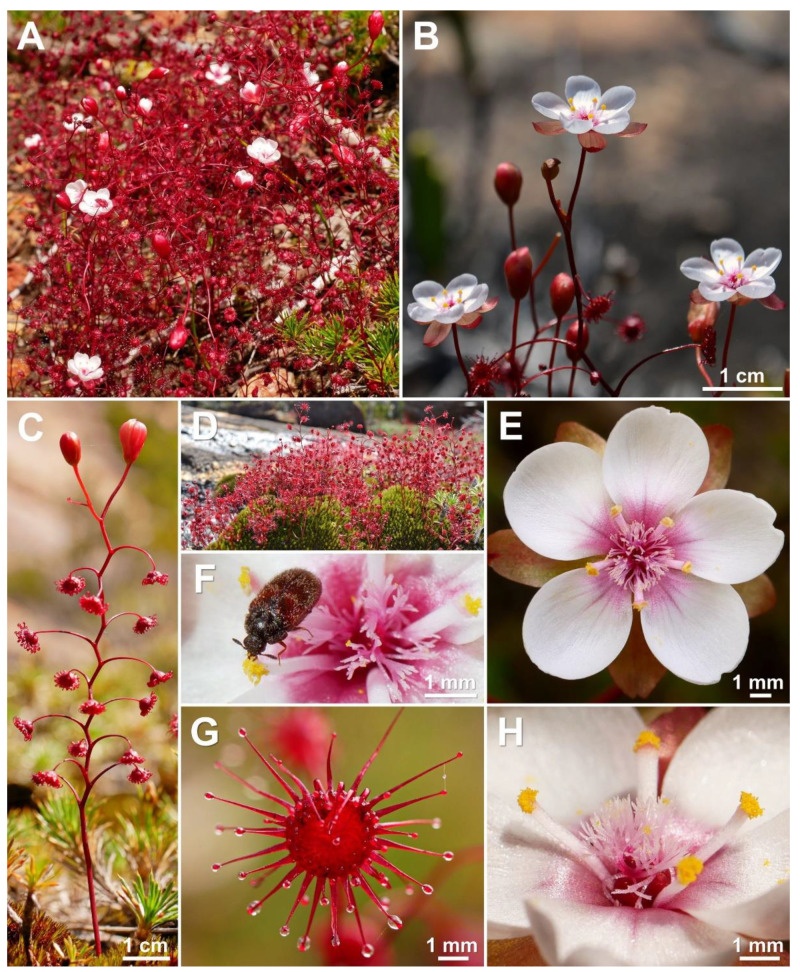
*Drosera esperensis* Lowrie. (**A**–**D**) habit; (**E**) flower in diffuse light; (**F**) flower with observed pollinator (a pollinivorous beetle of the family Dermestidae); (**G**) lamina; (**H**) stamens and styles (one stamen is missing). (**A**,**C**,**E**–**H**) from Cape Le Grand National Park, Western Australia, 19 September 2022; (**B**,**D**) from Cape Le Grand National Park, Western Australia, 16 September 2014. Images: T. Krueger.

**Description:** Tuberous perennial herb, often forming dense colonies, 7–20(–33) cm tall above ground including inflorescence. **Tubers** not seen. **Stem** (epigeous part) erect, self-supporting, simple, terete, strongly fractiflex, glabrous, 4–17(–29) cm tall, 0.8–1.3(–1.8) mm in diameter near soil surface, 0.5–0.9 mm in diameter at internodes, red or rarely yellowish green. **Cataphylls** subulate, 2–7 present on lower part of stem, (1.0–)1.7–6.5(–9.0) mm long, ca. 0.5 mm wide, red. **Leaves** solitary on each node, rarely uppermost 1–5 nodes with 2 shorter axillary leaves, alternate, 8–22 present in flowering individuals; internodes 2–18 mm. **Petioles** terete, semi-erect, arcuated abaxially (downwards) along whole length or rarely straight, glabrous, 8–22 mm long, 0.3–0.9 mm wide at base, tapering to 0.1–0.3 mm towards the lamina, red. **Lamina** peltate, orbiculate or sometimes orbiculate with slightly flattened adaxial lateral (upper) margin, shallowly to deeply concave, adaxial surface facing downwards or sometimes outwards, 2.0–4.4 mm long, 2.1–4.5 mm wide; lamina adaxial surface covered with stalked, carnivorous, secretive capitate glands (tentacles); tentacles 2.0–5.5 mm long at lamina margin, decreasing in size towards centre of lamina, with red stalk; lamina abaxial surface glabrous. **Inflorescence** a 1–5(–7)-flowered scorpioid cyme, terminal, simple, single-sided, 1.5–4.2 cm long. **Peduncle** terete, (0.2–)0.5–2.7 cm long, 0.5–0.9 mm in diameter, glabrous, red. **Pedicels** terete, erect or semi-erect in fruit, 10–20 mm long in fruit, 0.5–0.8 mm in diameter, spaced by (1–)2–5 mm along rhachis, glabrous, red. **Bracts** spathulate, narrowly spathulate or subulate, often slightly concave and arcuated adaxially (upwards), apex entire or irregularly crenulate, sometimes truncate, 1.8–4.4 mm long, 0.3–1.4 mm wide, glabrous. **Sepals** 5, narrowly obovate to narrowly elliptic, arcuated adaxially (upwards), slightly concave, often reflexed during anthesis, apex entire or crenulate, 5–10 mm long, 2.6–4.1 mm wide, abaxial surface microscopically glandular (appearing glabrous), red or rarely yellowish green, minute black spots often apparent. **Corolla** 10–14 mm in diameter. **Petals** 5, white with pale purplish red base, obovate to broadly obovate, deeply concave and slightly arcuated adaxially (upwards), apex rounded and entire, 5.2–6.5 mm long, 3.9–5.3 mm wide. **Stamens** 5, 2.8–3.4 mm long. **Filaments** ± linear, 0.3–0.6 mm wide, white (often pale purplish red at base). **Anthers** bithecate, retrorse, 0.6–0.9 mm wide, thecae pale yellow. **Pollen** yellow to orangey yellow. **Ovary** obovoid, 3-carpellate, fused, 1.0–1.5 mm in diameter, deep red. **Styles** 3, divided into a few filiform segments just above the base, style segments again divided into many terete style segments, forming a crowded tuft, extending laterally to reach or slightly exceed the filaments, 1.2–2.0 mm long, red at base, gradually transitioning to white near stigma. **Stigmas** shortly branched or simple, at tips of style segments, papillose, 0.2–0.5 mm long, white. **Seeds** narrowly obtrullate to narrowly obovate, outline sinuate, rarely straight, with slight ellipsoid swelling in the proximal and distal half (“bone-shaped seed”), funicular (upper) end truncate (rarely acute), lower (chalazal) end pointed with obtuse tip, 1.9–2.2 mm long, 0.3–0.5 mm wide, testa pale brown, only the median with a blackish-brown rectangular part; testa longitudinally reticulate, with anticlines thin and only shallowly raised.

**Etymology:** The specific epithet refers to the Esperance region of southern Western Australia where this species is endemic.

**Taxonomic notes:***Drosera esperensis* is morphologically similar to *D. koikyennuruff*, *D. microphylla*, and *D. reflexa*. It is distinguished from these three species by (contrasting characters in parentheses): (1) its tendency to form dense, clonal, mat-like colonies (plants not colony forming or only forming relatively sparse [not mat-like] colonies); (2) its ± linear filament shape (filaments increasing in width towards apex); (3) its petal colour, which is white with a pale purplish red base (dark red in *D. koikyennuruff*, reddish orange with deep red base in *D. microphylla*, or purplish pink with deep red base in *D. reflexa*; Figure 4); and (4) its style and filament colour, which is white with red or purplish red base (styles and filaments red, deep red, purplish red, or reddish purple). The distinctive white petal colour of *D. esperensis* is paralleled in *D. macropetala*, from which it can be distinguished by (contrasting characters in parentheses): (1) its mostly solitary leaves (leaves of upper 1–9 nodes in groups of 3 due to the presence of two shorter axillary leaves); (2) its tendency to form dense, clonal, mat-like colonies (plants not colony forming); (3) its ± linear filament shape (filaments dilated towards apex); and (4) its style colour, which is white with red or purplish red base (styles very dark red).

*Drosera esperensis* is geographically the most isolated species of the *D. microphylla* complex, occurring ca. 350 km east of the nearest confirmed population of *D. microphylla* (Figure 3; for discussion of the more proximate collection from Hopetoun, see Taxonomic notes under *D. microphylla*).

Plants from the Cape Arid area have been observed to frequently produce axillary leaves. Further studies of these populations are recommended to determine whether they represent a taxon distinct from *D. esperensis* (the type of which was collected from the Cape Le Grand area, where this species almost never produces axillary leaves).

*Drosera esperensis* was previously illustrated by Gibson [14] (p. 41) and Lowrie [1] (p. 435).

**Distribution and habitat:** Only known to occur within the Cape Le Grand and Cape Arid National Parks, east of Esperance (Figure 3). Grows in wet, mossy areas on and near granite hills in sandy clay or peat.

**Phenology:** Flowering has been recorded from August to October. In exceptionally wet habitats or seasons, flowering has been observed to continue until at least December (T. Krueger pers. obs.).

**Conservation status:** Not eligible for Western Australia Flora and Fauna Conservation Code listing and Least Concern (LC) under IUCN classification, following Cross [34]. *Drosera esperensis* frequently forms very large populations on the large, coastal granite hills east of Esperance. At least nine populations have been recorded, all of which are located on land managed by the Western Australian Department for Biodiversity, Conservation and Attractions (DBCA).

**Additional specimens examined:** AUSTRALIA. Western Australia: Cape Le Grande [Grand], 6 October 1966, *T.B. Muir 4246* (MEL 0097050A!); Frenchman Peak, in granitic sand on granite outcrops, 20 September 1991, *I. Solomon 512* (PERTH 01675931!); Cape Arid National Park, Mt Arid, SW from Thomas Fisheries, Hillside aspect S, brown loam over granite, 22 August 2014, *M. Hoggart & J. Waters 3/814* (PERTH 08780021!); Cheetup Hill, accessed via track off Saddleback Rd, NE edge of Cape le Grand NP, granite slope aspect SW, mossy brown loam over granite, 26 September 2014, *M. Hoggart 3/914* (PERTH 08780013!); Around the base of Cape Arid, without date, without collector (MEL 0096537A!).

### 3.4. **Drosera hortiorum**
*T.Krueger & G.Bourke*, **sp. nov.** (*Figure 3, Figure 4, Figure 5, Figure 8 and Figure 9*)

**Type:** AUSTRALIA. Western Australia: Wandoo National Park [precise locality withheld for conservation purposes], open granitic area, winter damp, semi-shaded, 20 August 2022, *F. Hort, J. Hort & T. Krueger FH 4575* (holotype PERTH!).

**Figure 8 biology-12-00141-f008:**
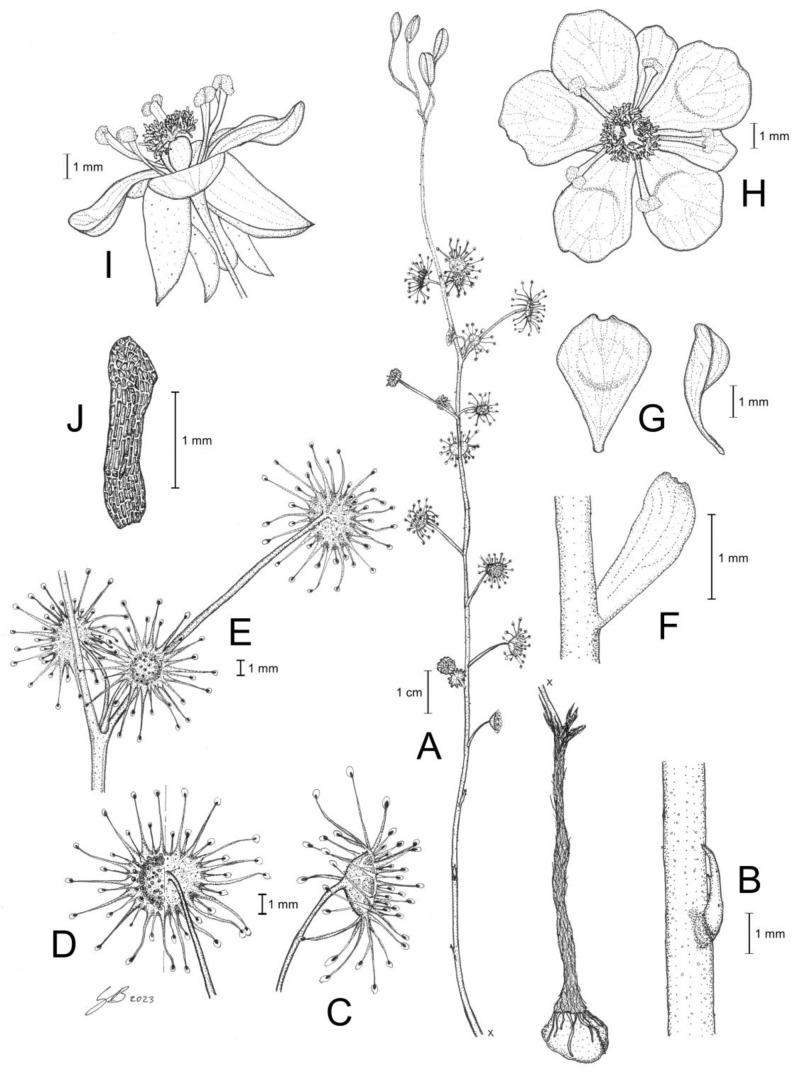
*Drosera hortiorum* T.Krueger & G.Bourke. (**A**) habit; (**B**) stem base with cataphyll; (**C**) lamina, lateral view; (**D**) lamina, left half adaxial view, right half abaxial view; (**E**) group of leaves from upper node of the stem, consisting of one cauline leaf and two axillary leaves; (**F**) bract; (**G**) petals, left adaxial view, right lateral view; (**H**) flower, top view; (**I**) flower, side view; and (**J**) seed. (**A**–**I**) from photographs of living plants from the type location, Wandoo National Park, Western Australia; (**J**) from near York, Western Australia. Drawing: G. Bourke.

**Figure 9 biology-12-00141-f009:**
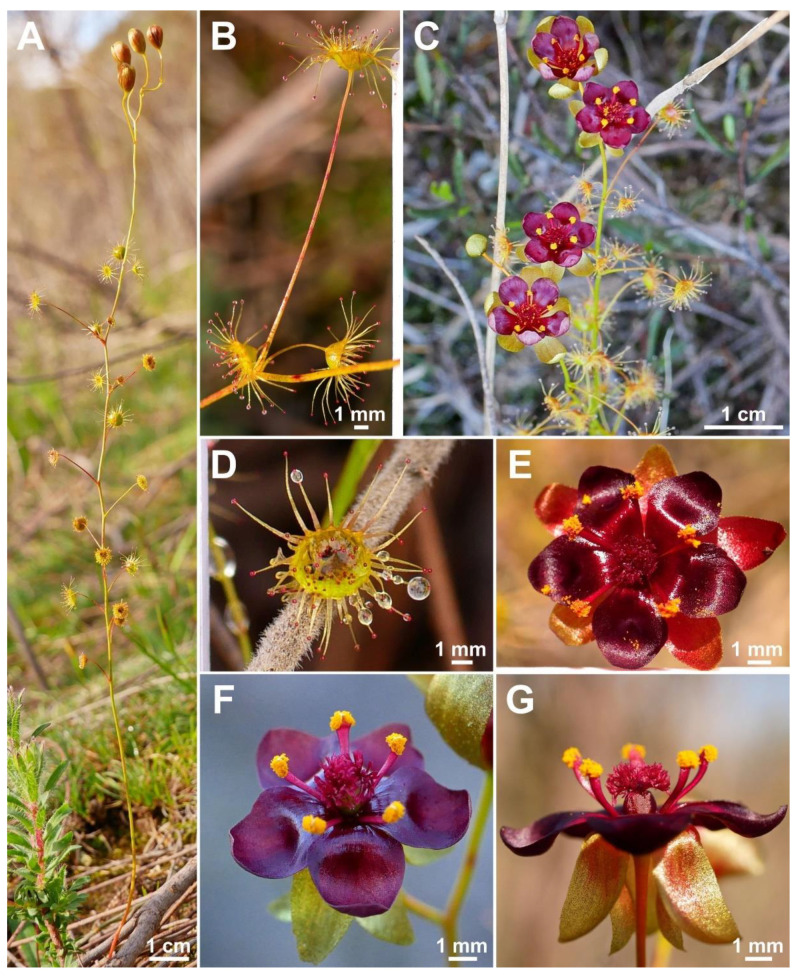
*Drosera hortiorum* T.Krueger & G.Bourke. (**A**) habit; (**B**) leaf on the upper part of the stem exhibiting two smaller axillary leaves emerging from the leaf axil; (**C**) group of flowering plants; (**D**) lamina, this species has an orbiculate lamina shape (sometimes with a sightly flattened upper margin); (**E**) flower in bright sunlight; (**F**) flower in diffuse light; and (**G**) flower, lateral view. (**A**,**B**,**E**) from Wandoo National Park, Western Australia, 20 August 2022. (**C**,**F**,**G**) from near Wickepin, Western Australia, 1 July 2022. (**D**) from near York, Western Australia, 4 September 2022. Images: T. Krueger.

**Diagnosis:** *Drosera hortiorum* is morphologically most similar to *D. rubricalyx* T.Krueger & A.Fleischm. and *D. macropetala* (Diels) T.Krueger & A.Fleischm. from which it differs by (contrasting characters in parentheses): (1) its small corolla diameter of 8–11 mm (corolla diameter 11–22 mm) and (2) its petal colour, which is deep red in inner half transitioning to dark purplish red in outer half (petals white with deep red base [*D. macropetala*] or petals deep red in inner half transitioning to deep pink in outer half [*D. rubricalyx*]; Figure 4). From *D. macropetala*, it is additionally distinguished by (contrasting characters in parentheses): (1) its filament shape, which are only slightly dilated towards apex, 0.3–0.5 mm wide near the apex (filaments strongly dilated towards apex, 0.5–0.9 mm wide near apex); (2) its tentacle stalk colour, which is greenish yellow (tentacle stalks red in lower half, greenish yellow in upper half, or red throughout); and (3) its filament colour, which is deep red (filaments deep red in lower half, white, or sometimes red in the upper half). *Drosera hortiorum* further shares morphological similarities with *D. calycina* Planch., from which it is distinguished by (contrasting characters in parentheses): (1) the presence of two smaller axillary leaves in the axils of the upper 1–7 cauline leaves (all cauline leaves solitary); (2) its lamina shape, which is orbiculate or orbiculate with a slightly flattened upper margin (lamina reniform or orbiculate with flattened, often truncated upper margin); (3) its filament shape, which only slightly dilated towards apex, 0.3–0.5 mm wide near apex (filaments strongly dilated towards apex, 0.5–1.1 mm wide near apex); and (4) its straight, pin- to bone-shaped seeds (seeds flattened, slightly falcate to slightly allantoid).

**Description:** Tuberous perennial herb, 14–32(–41) cm tall above ground including inflorescence. **Tuber** subglobose, ca. 10 mm in diameter, enclosed in black papery sheaths from previous seasons’ growth. **Stem** (subterranean part) ca. 6 cm long, ca. 2.0 mm in diameter, enclosed in brown, fibrous tunic formed from previous seasons’ stems and roots. **Roots** few, fibrous, emerging laterally from along subterranean part of stem, mostly immediately above tuber. **Stem** (epigeous part) erect, self-supporting, simple, terete, slightly fractiflex, glabrous, (10–)14–27(–34) cm tall, 0.7–1.3 mm in diameter near soil surface, 0.4–0.8 mm in diameter at internodes, yellowish green or sometimes red, red near soil level; sometimes 2–5 stems emerging from the same tuber. **Cataphylls** 4–9 on lower part of stem, subulate, 1.4–3.1 mm long, ca. 0.5 mm wide, red to orangey yellow. **Leaves** solitary in lower part of stem but upper (0–)10–50% of leaves in groups of three per node, due to two much shorter axillary leaves emerging from the axils; internodes 4–22 mm and 9–14 nodes bearing leaves (foliose nodes) present in flowering individuals. **Petioles** terete, semi-erect, straight or slightly arcuated abaxially (downwards), strongly arcuated abaxially near tip, glabrous, 8–23(–27) mm long, 0.4–0.7 mm wide at base, tapering to 0.1–0.3 mm towards lamina, yellowish green or sometimes red, tip often tinged orangey yellow. **Lamina** peltate, orbiculate or orbiculate with slightly flattened adaxial lateral margin, shallowly concave, adaxial surface facing outwards or slightly downwards, 2.6–4.0 mm long, 2.7–4.2 mm wide; lamina adaxial surface covered with stalked, carnivorous, secretive capitate glands (tentacles); tentacles 2–5 mm long at lamina margin, decreasing in size towards centre of lamina, with greenish yellow stalk (sometimes red at base); lamina abaxial surface glabrous. **Petioles of axillary leaves** terete, semi-erect, arcuated downwards along whole length, glabrous, 4–6(–8) mm long, 0.2–0.4 mm wide at base, tapering to 0.1–0.2 mm towards lamina, yellowish green or sometimes red. **Lamina of axillary leaves** of same shape as the lamina described above, 2.0–2.9 mm long, 2.0–3.0 mm wide. **Inflorescence** a 2–6-flowered scorpioid cyme, terminal, simple, single-sided, (2.6–)3.2–6.7(–8.8) cm long. **Peduncle** terete, 1.2–4.1 cm long, 0.4–0.6 mm in diameter, microscopically glandular (appearing glabrous), yellowish green, sometimes red. **Pedicels** terete, erect in fruit, 6–21 mm long in fruit, 0.3–0.5 mm in diameter, spaced by 2–8 mm along rhachis, microscopically glandular (appearing glabrous), yellowish green, sometimes red. **Bracts** spathulate, narrowly obovate, elliptic or subulate, arcuated adaxially (upwards), often concave, apex entire or irregularly crenulate, 1.4–3.0 mm long, 0.5–0.9 mm wide, abaxial surface microscopically glandular. **Sepals** 5, narrowly elliptic to narrowly obovate, arcuated adaxially (upwards), slightly concave, often reflexed during anthesis, apex entire or crenulate, 5–9 mm long, 2.5–4.3 mm wide, abaxial surface microscopically glandular, yellowish brown to yellowish green or sometimes red, minute black spots often apparent. **Corolla** 8–11 mm in diameter. **Petals** 5, deep red in inner half transitioning to dark purplish red in outer half, obovate, deeply concave and slightly arcuated adaxially (upwards), apex rounded and entire, 4.1–5.0 mm long, 3.1–4.0 mm wide. **Stamens** 5, 3.0–3.5 mm long. **Filaments** very slightly dilated towards apex, straight or slightly falcate, 0.2–0.4 mm wide at base, 0.3–0.5 mm wide near apex, deep red. **Anthers** bithecate, retrorse, 0.8–1.1 mm wide, thecae reddish orange. **Pollen** yellow. **Ovary** obovoid, 3-carpellate, fused, 1.3–1.6 mm in diameter, deep red or dark olive. **Styles** 3, divided into a few filiform segments just above the base, style segments again divided into many terete style segments, forming a crowded tuft, not extending laterally beyond filaments, 1.0–1.4 mm long, deep red. **Stigmas** simple, at tips of style segments, papillose, ca. 0.2 mm long, deep red. **Seeds** narrowly obtrullate to narrowly obovate, straight or slightly curved, outline rectangular with slight ellipsoid swelling in the proximal and distal half, funicular (upper) end truncate, basal (chalazal) end pointed with obtuse tip, 1.8–2.2 mm long, 0.3–0.5 mm wide, testa pale brown, only the rectangular middle part blackish brown; testa more or less isodiametrically (to slightly longitudinally) reticulate, with anticlines thin and only shallowly raised.

**Etymology:** The specific epithet honours Fred Hort (1937–) and Jean Hort (1952–), enthusiastic field botanists, nature photographers, and volunteers at the Western Australian Herbarium who found this species at the Wandoo National Park type location in 1987 and brought it to the attention of the authors of the present work. Their prolific collections from the eastern Darling Range have led to the recognition of many new species, several of which have already been named in their honour (e.g., [35,36,37,38]).

**Taxonomic notes:** The presence of axillary leaves in the upper parts of the stem, as well as seed characters (Figure 5), link *D. hortiorum* to the morphologically similar *D. macropetala* and *D. rubricalyx*. However, its corolla is of a much smaller size and its distinctive dark purplish red petal colour easily distinguishes it from these two species (Figure 4). In addition, all three species are geographically well separated, with *D. macropetala* and *D. rubricalyx* occurring well north of Perth while *D. hortiorum* is only known from areas to the east and south-east of Perth (Figure 3).

Despite its usually much smaller size, the corolla shape and colour of *D. hortiorum* closely resembles that of *D. calycina*. Both species further occur in close geographic proximity (Figure 3). However, *D. hortiorum* is easily distinguished from *D. calycina* by the presence of axillary leaves in the upper parts of the stem (D*. calycina* has solitary leaves and always lacks axillary leaves). While *D. hortiorum* has been observed growing within a few hundred metres of *D. calycina* near Glen Forrest, they do not co-occur syntopically due to their different habitat requirements. In that area, *D. calycina* is restricted to laterite soils in Jarrah forests while *D. hortiorum* grows in clay loam around granite slopes and boulders.

A photograph of *D. hortiorum* was published in 1987 by Lowrie [12] (p. 67) who, at the time, treated all taxa of the complex under *D. microphylla*. In his 2014 taxonomic treatment, Lowrie likely included *D. hortiorum* under *D. calycina*, as he described the presence of axillary leaves for this species (“sometimes forming leaves in groups of 2 to 3 in the upper parts” [1] (p. 354)). *Drosera hortiorum* is further illustrated in *Drosera* of the World [15] (p. 228), but with an erroneous location description (Badgingarra). The pictured plant actually represents a specimen cultivated by G. Bourke and originated from the late Allen Lowrie.

**Distribution and habitat:** Known from Glen Forrest, Wandoo National Park (near York, east of Perth) and two additional sites in the wheatbelt region near York and Wickepin (Figure 3). In the western part of its range, *D. hortiorum* appears to be associated with low granite outcrops and granite slopes where it grows in poorly drained clay loam with *Borya* sp. In the eastern part of its range, *D. hortiorum* has been recorded from within and near shallow drainage channels and moist sandplains in sandy clay.

It is curious to note that *D. hortiorum* has been observed in such a wide range of different habitats, as this is unusual for the complex. Only *D. microphylla* is also known from very different types of habitat.

**Phenology:** Flowering has been recorded from June to September.

**Conservation status:** Recommended for listing as Priority Two (poorly known species) under Conservation Codes for Western Australian Flora and Fauna (Western Australian Herbarium 1998–; https://florabase.dpaw.wa.gov.au/ (accessed on 6 December 2022)). Data deficient (DD) following IUCN [29]. Three of the four known locations occur on land managed by the Western Australian Department of Biodiversity, Conservation and Attractions (DBCA). The type population currently comprises ca. 30–40 mature individuals. Additional populations were found by wildflower enthusiasts near Wickepin (“foxydoug” 2022. iNaturalist observation: https://www.inaturalist.org/observations/123515288 (accessed on 9 January 2023)) and near York (photograph posted by Patricia Paull on Facebook). Both populations consist of only ca. 15–30 flowering-sized individuals. The population near Glen Forrest discovered by L. Diels and E. Pritzel in 1901 (*Diels & Pritzel 534/B.59*) was re-located in September 2022 by T. Krueger. At this site, ca. 30 flowering individuals occur in an unprotected area. Given the small number of mature individuals known to occur, *D. hortiorum* could be threatened by unlicensed collection and poaching for the horticultural trade. Further surveys are recommended to gain a better understanding of this taxon’s biology, distribution, number and size of populations, and to identify additional threats.

**Additional specimens examined (paratypes):** AUSTRALIA. Western Australia: Swan Distr.: Smith’ Mill [Glen Forrest], Sept. 1901, *Diels & Pritzel 534/B.59* (PERTH 00666904!); Wandoo National Park [precise locality withheld for conservation purposes], open granitic area, winter damp, semi-shaded, 15 August 2022, *F. Hort & J. Hort FH 4574* (PERTH!).

**Additional localities examined:** Wickepin [precise locality withheld for conservation purposes], poorly drained, seasonally moist drainage channel, 1 July 2022, T. Krueger pers. obs.*;* York [precise locality withheld for conservation purposes], open Wandoo woodland with low heath, poorly drained seasonally moist sandplain, 4 September 2022, T. Krueger pers. obs.

### 3.5. **Drosera koikyennuruff**
*T.Krueger & A.S.Rob.*, **sp. nov.** (*Figure 3, Figure 4, Figure 5, Figure 10 and Figure 11*)

**Type:** AUSTRALIA. Western Australia: Stirling Range National Park [precise locality withheld for conservation purposes], grey clayey sand over sandstone, 23 June 1988, *A. Rose 1029* (holotype PERTH 05812402!).

**Figure 10 biology-12-00141-f010:**
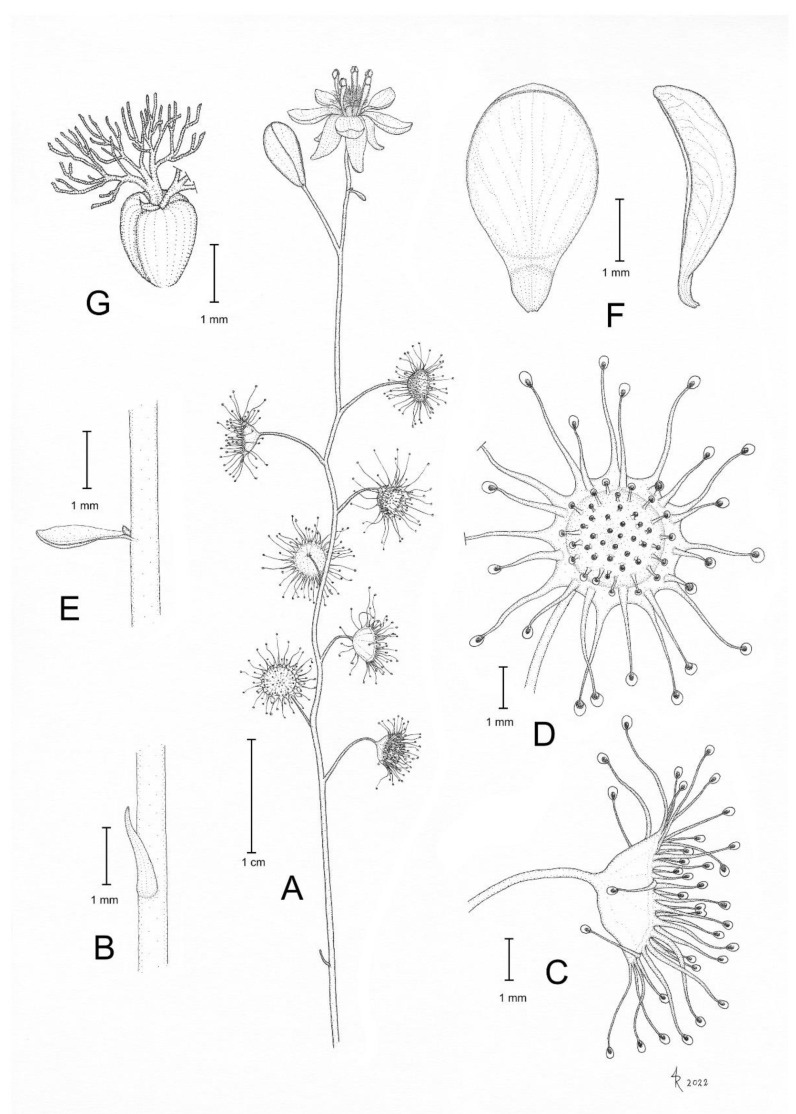
*Drosera koikyennuruff* T.Krueger & A.S.Rob. (**A**) habit; (**B**) cataphyll from stem base; (**C**) lamina, lateral view; (**D**) lamina, adaxial view; (**E**) bract; (**F**) petals, adaxial view (left), lateral view (right); and (**G**) gynoecium, with two styles removed. (**A**,**D**–**G**) from type and photographs of living plants and (**B**) from photographs of living plants only. Drawing: A. Robinson.

**Figure 11 biology-12-00141-f011:**
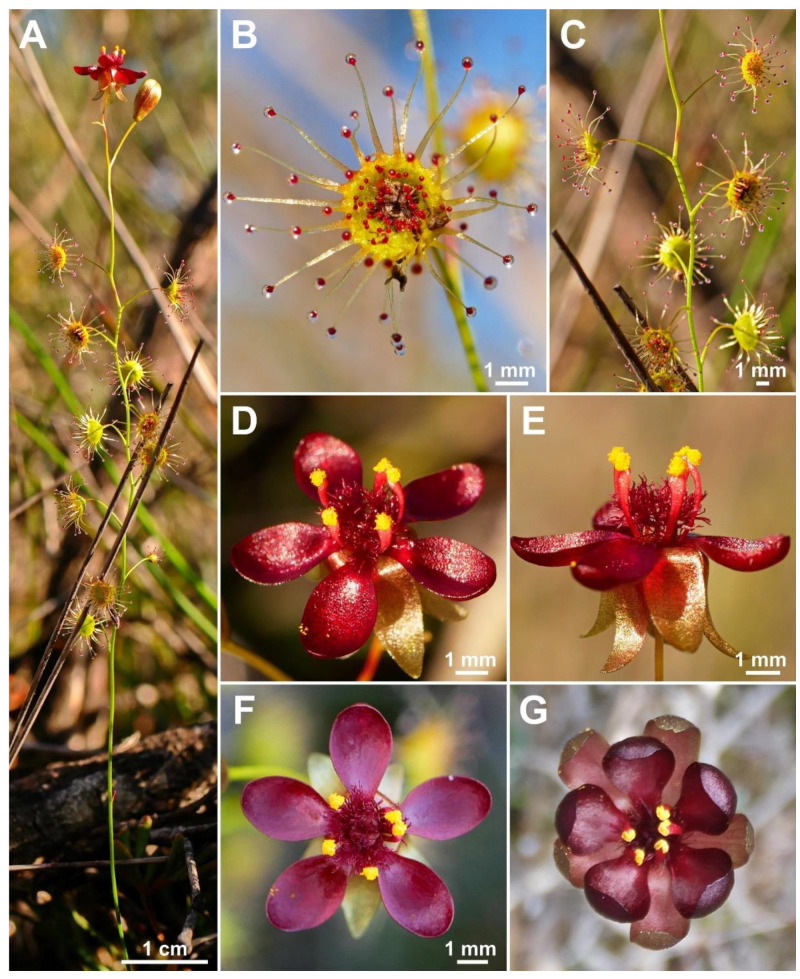
*Drosera koikyennuruff* T.Krueger & A.S.Rob. (**A**) habit; (**B**) lamina; (**C**) stem and leaves; (**D**) flower in bright sunlight; (**E**) flower, lateral view; and (**F**,**G**) flowers in diffuse light. (**A**–**F**) from Stirling Range National Park, Western Australia, 2 July 2022; images by T. Krueger. (**G**) from near Woogenellup, Western Australia, July 2021; image by P. Luscombe.

**Diagnosis:***Drosera koikyennuruff* is morphologically most similar to *D. microphylla* Endl., from which it is distinguished by (contrasting characters in parentheses): (1) its much earlier flowering time from June to July (flowering from August to October); (2) its dark red petal colour (petals reddish orange with deep red bases); (3) its deep red stigma colour (stigmas reddish purple), (4) its yellowish green tentacle stalk colour (tentacle stalks red or red in lower half with upper half yellowish green); and (5) its preference for relatively dry sandy habitats in open Mallee woodlands (mossy wet habitat areas on and near granite outcrops, seasonally wet swamps, or rocky mountain slopes). It is further distinguished from the morphologically similar *D. reflexa* G.Bourke & A.S.Rob. by (contrasting characters in parentheses): (1) its sparse populations, which are not colony-forming (plants forming dense populations via adventitious stolons); (2) its petal shape, which is narrowly obovate to broadly spathulate (petals obovate to very broadly obovate); (3) its dark red petal colour (petals purplish pink with deep red base); (4) its yellowish brown to yellowish green sepal colour (sepals red to purplish red); and (5) its preference for relatively dry sandy habitats in open Mallee woodlands in and around the Stirling Range (plants occurring in shallow moss on granite outcrops between Walpole and Denmark).

**Description:** Tuberous perennial herb, ca. 15 cm tall above ground including inflorescence. **Tuber** not seen. **Stem** (epigeous part) erect, self-supporting, simple, terete, slightly fractiflex, glabrous, 10–12 cm tall, 0.4–0.5 mm in diameter near soil surface, 0.3–0.4 mm in diameter at internodes, yellowish green. **Cataphylls** subulate, few present on lower part of stem, ca. 1.5 mm long, ca. 0.3 mm wide, red to orangey yellow. **Leaves** solitary on each node, alternate, 2–11 present in flowering individuals; internodes 2–11 mm. **Petioles** terete, semi-erect, arcuated abaxially (downwards) along whole length or arching increasing gradually towards the lamina, glabrous, 5–9 mm long, 0.3–0.4 mm wide at base, tapering to 0.1–0.2 mm towards the lamina, yellowish green with orangey yellow or red tip. **Lamina** peltate, orbiculate, shallowly concave, adaxial surface facing outwards or slightly downwards, 2.3–3.4 mm long, 2.3–3.4 mm wide; lamina adaxial surface covered with stalked, carnivorous, secretive capitate glands (tentacles); tentacles 2–4 mm long at lamina margin, decreasing in size towards centre of lamina, with yellowish green stalk; lamina abaxial surface glabrous. **Inflorescence** a 1–2-flowered scorpioid cyme, terminal, simple, single-sided, 2.5–4.2 cm long. **Peduncle** terete, 1.2–1.5 cm long, 0.3–0.4 mm in diameter, glabrous, yellowish green, often blotched with red. **Pedicels** terete, erect in fruit, 5–20 mm long in fruit, 0.2–0.3 mm in diameter, spaced by 5–13 mm along rhachis, glabrous, yellowish green in lower half, orangey yellow to reddish orange in upper half. **Bracts** spathulate, narrowly obovate or subulate, apex entire or irregularly crenulate, 1.5–2.5 mm long, 0.3–0.4 mm wide, glabrous. **Sepals** 5, narrowly obovate, arcuated adaxially (upwards), slightly concave, often reflexed during anthesis, apex entire or sometimes crenulate, 4–6 mm long, 1.8–2.1 mm wide, abaxial surface microscopically glandular (appearing glabrous), yellowish brown to yellowish green, minute black spots often apparent. **Corolla** 8–12 mm in diameter. **Petals** 5, dark red, broadly spathulate to narrowly obovate, deeply concave and slightly arcuated adaxially (upwards), apex rounded and entire, 3.7–5.4 mm long, 2.0–2.2 mm wide. **Stamens** 5, 2.9–3.5 mm long. **Filaments** slightly dilated towards apex, ca. 0.2 mm wide at base, ca. 0.5 mm wide near apex, deep red. **Anthers** bithecate, retrorse, 0.5–0.6 mm wide. **Pollen** yellow. **Ovary** obovoid, 3-carpellate, fused, 1.0–1.6 mm in diameter, deep red. **Styles** 3, divided into a few filiform segments just above the base, style segments again divided into many terete style segments, forming a crowded tuft, extending laterally beyond the filaments, 1.3–2.3 mm long, deep red. **Stigmas** simple or shortly branched, at tips of style segments, ca. 0.2 mm long, deep red. **Seeds** not seen.

**Etymology:** The specific epithet refers to *koikyennuruff*, the Noongar Aboriginal name for the Stirling Range, where this taxon occurs. The name means “mist over hills” [39].

**Taxonomic notes:** The overall habit as well as petiole, lamina, and style shape of *D. koikyennuruff* indicate that it is morphologically most similar to *D. microphylla*. Both species grow in close proximity at sites in Stirling Range National Park but favour a different habitat type. While *D. koikyennuruff* grows in low-lying areas with sandy soils in open Mallee woodlands, *D. microphylla* appears (in this area) to be restricted to the middle and upper slopes of the Stirling Range mountains where it typically grows in rocky or lateritic soils. In addition, the two species also differ phenologically and thus are reproductively isolated by non-overlapping flowering times, with *D. koikyennuruff* flowering from June to July while *D. microphylla* flowers from late August to October. *Drosera koikyennuruff* is easily distinguished from *D. microphylla* by its dark red petal colour (*D. microphylla* has reddish orange petals with deep red bases). The flower colour of the type specimen *A. Rose 1029* (PERTH 05812402!) is denoted as “burgandy” (burgundy), which is an apt description for the distinctive dark red petal colour of this species.

*Diels 3009* (Plantagenet: westlich des Sucky Peeks [west of “Sucky Peek” =Sukey Hill], B 10 0755996!) represents an intriguing collection from near Cranbrook (marked with “1” in Figure 3). The exceptionally small and consistently single-flowered plants were collected in late May, which is potentially within the flowering time of *D. koikyennuruff*. In addition, the petal colour is described by Diels as “dunkelkarmin” (dark carmine/dark crimson), which might match the dark red flower colour of *D. koikyennuruff*. However, the plants are overall much smaller and appear to have shorter styles. Since this population could not be re-located by the authors prior to submission, it is not currently known whether it represents *D. koikyennuruff* or a closely allied, undescribed species. It is thus not included under *D. koikyennuruff* in the present work.

**Distribution and habitat:** *Drosera koikyennuruff* is only known from two locations, one in Stirling Range National Park and one from nearby Woogenellup (Figure 3). It grows in low heath amongst Mallee scrub in sandy clay soils.

**Phenology:** Flowering has been recorded in June and July.

**Conservation status:** Recommended for listing as Priority Two (poorly known species) under Conservation Codes for Western Australian Flora and Fauna (Western Australian Herbarium 1998–; https://florabase.dpaw.wa.gov.au/ (accessed on 6 December 2022)). Data deficient (DD) following IUCN [29]. The Stirling Range National Park population (recorded in 1988; *A. Rose 1029*) is the only known population on land managed by the Western Australian Department of Biodiversity, Conservation and Attractions (DBCA). This population was surveyed in July 2022 but despite considerable effort, only a single mature individual and four juvenile plants were located (T. Krueger pers. obs.). In July 2021, photos from a second population near Woogenellup were posted on Facebook by local resident Peter Luscombe. The population size of this second population is currently unknown. Given the extremely small number of individuals that are known to exist, any disturbance of the habitat or unlicenced collection could be disastrous to the species’ long-term survival. Further surveys are strongly recommended to gain a better understanding of this taxon’s distribution, number and size of populations, and to identify potential additional threats.

### 3.6. **Drosera macropetala**
*(Diels) T.Krueger & A.Fleischm.*, **comb. nov. & stat. nov.** (*Figure 3, Figure 4, Figure 5, Figure 12 and Figure 13*)

**Basionym:** *Drosera microphylla* var. *macropetala* Diels, Das Pflanzenreich Heft 26: 121 (1906).

**Lectotype (designated here):** [AUSTRALIA]. Westaustralien [Western Australia: “between Moore River and Murchison Rivers”—the collection locality is not provided on the lectotype specimen at B, but on all other syntypes. As evident from Drummond [18,19], the locus classicus is “about 4 miles to the north of Dundaragan”, which is ca. 15 km north of today’s townsite of Dandaragan, see under “Notes on Drummond’s type collection” below], without date [collected 1850 or 1851; [30]], *J. Drummond coll. VI n. 109* (B100755976!; isolectotypes: BM000752962 photo!; E00279841 photo!; FI011168 photo!; G00410322! [one specimen mounted on three sheets, two of them without collector’s number and locality, but definitely from the gathering *Drummond 109*]; K000659189!; K000659190!; K000659191!; LD1974467 photo!; LD1971651 photo!; LD1971715 photo! (wrongly labelled as “110”); MEL97059!; NSW146696 photo!; OXF00140703!; P00713916 photo!; P00749106 photo!; W0131702!).

= *Drosera calycina* var. *macropetala* (Diels) N.G.Marchant in annot., nomen nudum.

**Figure 12 biology-12-00141-f012:**
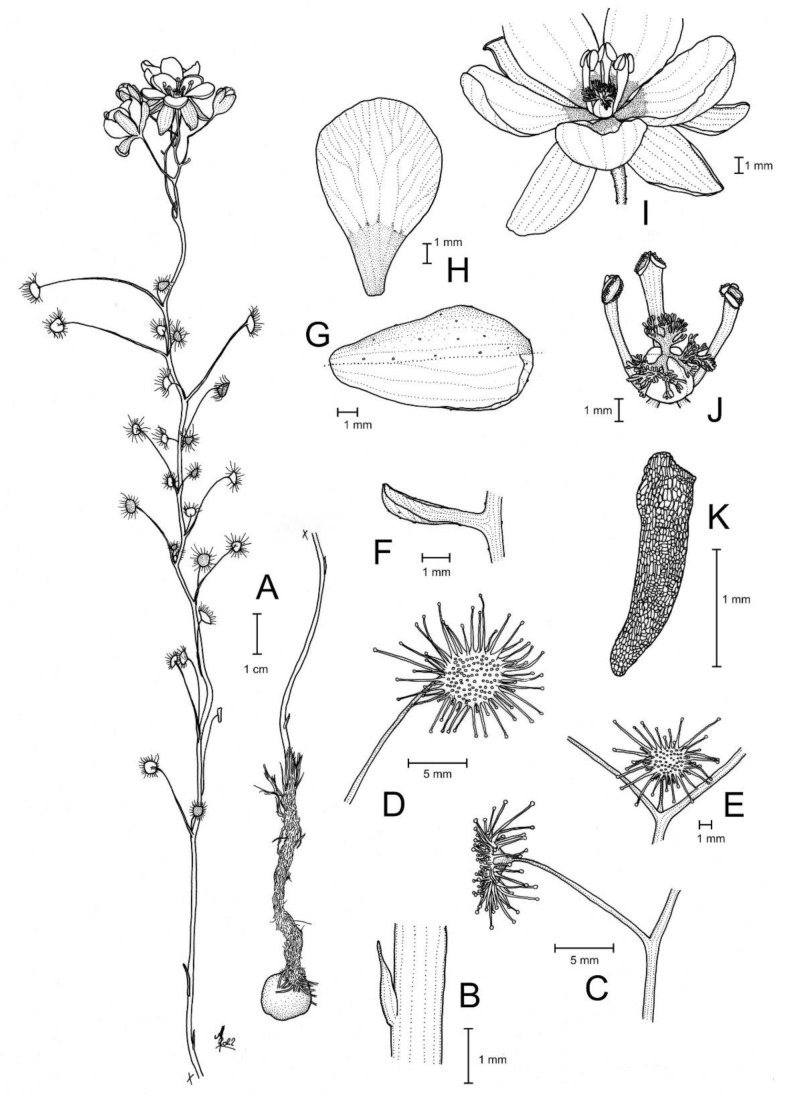
*Drosera macropetala* (Diels) T.Krueger & A.Fleischm. (**A**) habit; (**B**) cataphyll from stem base; (**C**) cauline leaf from lower part of the stem; (**D**) lamina, adaxial view; (**E**) axillary leaf from a stem upper node; (**F**) bract; (**G**) sepal, top half abaxial view, lower half adaxial view; (**H**) petal; (**I**) flower (two stamens removed to reveal the ovary); (**J**) gynoecium, with two stamens removed; and (**K**) seed. (**A**) from the type (*J. Drummond coll. VI n. 109*); (**B**,**D**,**G**,**H**) from *T. Krueger 29*; (**C**,**E**,**F**,**H**–**K**) from photographs of living plants from near Dandaragan, Western Australia. Drawing: A. Fleischmann.

**Figure 13 biology-12-00141-f013:**
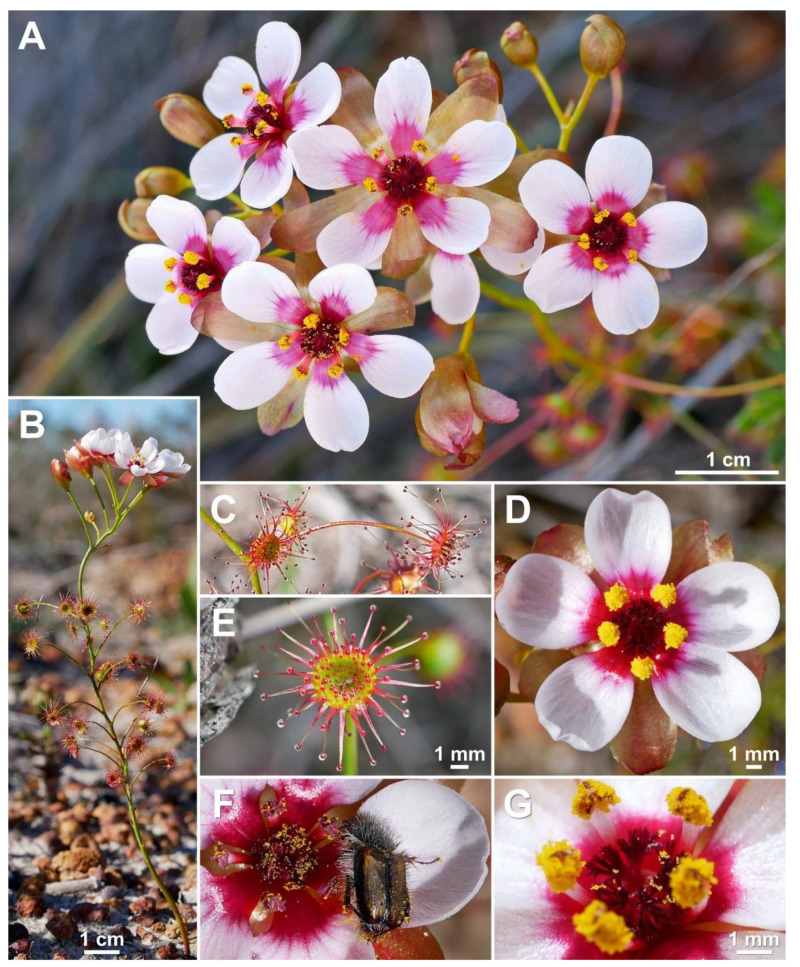
*Drosera macropetala* (Diels) T.Krueger & A.Fleischm. (**A**) flowers of a single plant in diffuse light, this species often has 3–5 flowers open simultaneously; (**B**) habit of a relatively small individual; (**C**) upper leaf exhibiting two smaller axillary leaves emerging from the leaf axil; (**D**) flower in bright sunlight; (**E**) lamina; (**F**) flower with observed pollinator (a beetle of the family Scarabaeidae); and (**G**) stamens and styles. All from near Dandaragan, Western Australia, 16 August 2021. Images: T. Krueger.

**Description:** Tuberous perennial herb, (12–)15–38(–44) cm tall above ground including inflorescence. **Tuber** subglobose, 8–15 mm in diameter, enclosed in black papery sheaths from previous seasons’ growth. **Stem** (subterranean part) 4.2–6.8 cm long, 1.5–4.0 mm in diameter, enclosed in brown, fibrous tunic formed from previous seasons’ stems and roots. **Roots** few, fibrous, emerging laterally from along subterranean part of stem, mostly immediately above tuber. **Stem** (epigeous part) erect, self-supporting, simple, terete, slightly fractiflex, glabrous, (8–)12–32(–37) cm tall, (0.8–)1.0–1.5 mm in diameter near soil surface, (0.4–)0.6–1.2 mm in diameter at internodes, yellowish green or sometimes red, red near soil level; sometimes 2–5 stems emerging from the same tuber. **Cataphylls** 4–9 on lower part of stem, subulate, 1.5–3.5 mm long, ca. 0.5 mm wide, red to orangey yellow. **Leaves** solitary in lower part of stem but upper (0–)15–50(–65)% of leaves in groups of three per node, due to two much shorter axillary leaves emerging from the axils; internodes 2–21 mm and 10–18(–22) nodes bearing leaves (foliose nodes) present in flowering individuals. **Petioles** terete, semi-erect, slightly arcuated abaxially (downwards), strongly arcuated abaxially near tip, glabrous, 10–30(–45) mm long, 0.5–0.9(–1.1) mm wide at base, tapering to 0.1–0.3 mm towards lamina, yellowish green or sometimes red, tip often tinged orangey yellow or red. **Lamina** peltate, orbiculate or orbiculate with slightly flattened adaxial lateral margin, shallowly concave, adaxial surface facing outwards or slightly downwards, 2.6–3.6(–4.2) mm long, 2.8–4.4 mm wide; lamina adaxial surface covered with stalked, carnivorous, secretive capitate glands (tentacles); tentacles 2–6 mm long at lamina margin, decreasing in size towards centre of lamina, stalk red in lower half, yellowish green in upper half or red throughout; lamina abaxial surface glabrous. **Petioles of axillary leaves** terete, semi-erect, arcuated downwards along whole length, glabrous, 3–7(–9) mm long, 0.3–0.5 mm wide at base, tapering to 0.1–0.3 mm towards lamina, yellowish green or sometimes red. **Lamina of axillary leaves** of same shape as the lamina described above, 1.9–3.0(–3.4) mm long, 2.1–3.1(–3.6) mm wide. **Inflorescence** a 2–8-flowered scorpioid cyme, terminal, simple, single-sided, 3.5–7.5 cm long. **Peduncle** terete, 1.3–4.2 cm long, 0.5–0.9 mm in diameter, microscopically glandular (appearing glabrous), yellowish green or red. **Pedicels** terete, erect in fruit, 6–24 mm long in fruit, 0.4–0.8 mm in diameter, spaced by (1–)2–8 mm along rhachis, microscopically glandular (appearing glabrous), yellowish green or red. **Bracts** spathulate, narrowly obovate or subulate, often arcuated adaxially (upwards) and concave, apex entire or irregularly crenulate, 1.9–4.0 mm long, 0.5–1.3 mm wide, glabrous. **Sepals** 5, obovate, narrowly obovate or narrowly elliptic, arcuated adaxially (upwards), slightly concave, often reflexed during anthesis, apex entire or crenulate, 7–12 mm long, (2.7–)3.0–5.3 mm wide, abaxial surface microscopically glandular, yellowish brown to red, minute black spots often apparent. **Corolla** (11–)13–22 mm in diameter**. Petals** 5, white with deep red base, obovate to very broadly obovate, sometimes broadly spathulate, deeply concave and slightly arcuated adaxially (upwards), apex rounded and entire, (5.5–)6.5–11.0 mm long, (4.0–)4.5–7.5 mm wide. **Stamens** 5, 3.3–4.9 mm long. **Filaments** dilated towards apex, 0.3–0.5 mm wide at base, 0.5–0.9 mm wide near apex, deep red in lower half, white or red in upper half. **Anthers** bithecate, retrorse, 0.8–1.5 mm wide, thecae orange or reddish orange, rarely pale yellow. **Pollen** yellow. **Ovary** obovoid, 3-carpellate, fused, 1.3–1.8 mm in diameter, deep red. **Styles** 3, divided into a few filiform segments just above the base, style segments again divided into many terete style segments, forming a crowded tuft, not extending laterally beyond filaments, 1.2–1.6 mm long, very dark red. **Stigmas** simple or shortly branched, at tips of style segments, papillose, ca. 0.2–0.3 mm long, very dark red. **Seeds** narrowly obtrullate to narrowly obovate, outline narrowly conical with slight ellipsoid swelling in the lower half, funicular (upper) end truncate, basal (chalazal) end obtuse to acute (=pin- or nail-shaped seeds), 1.7–2.0 mm long, 0.3–0.5 mm wide, testa black-brown (often chalazal and funicular ends pale brown); testa more or less isodiametrically reticulate, with anticlines thin and only shallowly raised.

**Etymology:** The specific epithet, from the Greek *macro-* (=large) and *petalum* (=petal), refers to the comparatively large petals of this species. Indeed, with a length of up to 11 mm and a width of up to 7.5 mm, *D. macropetala* produces the largest petals known in the *D. microphylla* complex. Only *D. calycina* sometimes produces similar-sized (but usually distinctly narrower) petals.

**Taxonomic notes:** *Drosera macropetala* is morphologically similar to *D. calycina*, *D. hortiorum*, and *D. rubricalyx* from which it can be quickly and reliably differentiated—even in herbarium material—by its petal colour, which is white with a deep red base (petals deep red in inner half transitioning to purplish red in outer half in *D. calycina*; deep red in inner half transitioning to dark purplish red in outer half in *D. hortiorum*; deep red in inner half transitioning to deep pink in outer half in *D. rubricalyx*). It is further distinguished from *D. calycina* by (contrasting characters in parentheses): (1) the presence of two smaller axillary leaves in the axils of the upper 1–9 cauline leaves (all cauline leaves solitary); (2) its lamina shape, which is orbiculate or orbiculate with slightly flattened upper margin (lamina reniform or orbiculate with flattened, often truncated upper margin); and (3) its straight, pin-like seeds (seeds flattened, falcate in *D. calycina*). It is further distinguished from *D. hortiorum* by (contrasting characters in parentheses): (1) its much larger corolla diameter of 13–22 mm (corolla diameter 8–11 mm) and (2) its filament shape, which is markedly dilated towards apex, 0.5–0.9 mm wide near apex (filament width only slightly dilated towards apex, 0.3–0.5 mm wide near apex). *Drosera macropetala* is further distinguished from *D. rubricalyx* by (contrasting characters in parentheses): (1) its broader petals, which are 4.5–7.5 mm wide (petals 3.3–4.8 mm wide); (2) its usually much longer peduncles, which are 1.3–4.2 cm long (peduncles 0.8–2.2 cm long); (3) its filament shape, which is markedly dilated towards apex, 0.5–0.9 mm wide near apex (filaments only slightly dilated towards apex, 0.4–0.6 mm wide near apex); and (4) its yellowish brown or red sepals, which are not strongly contrasting the yellowish green or red stem colour (sepals red, strongly contrasting the yellowish green stem). The distinctive white petal colour of *D. macropetala* is paralleled in *D. esperensis*, from which it can be distinguished by (contrasting characters in parentheses): (1) the presence of two smaller axillary leaves in the axils of the upper 1–9 cauline leaves (all cauline leaves usually solitary); (2) plants not colony forming (plants forming dense, mat-like colonies); (3) its filament shape, which is markedly dilated towards apex, 0.5–0.9 mm wide near apex (filaments ± linear, 0.3–0.6 mm wide near apex); and (4) its style colour, which is very dark red (styles white with red base).

Gibson [14] erroneously lists the petal colour of *D. macropetala* as “purple” even though Diels [8] (p. 121) clearly states “petala […] siccata pallida (non atropurpurea)” (dried petals pale white [not dark red]) in his description of *D. microphylla* var. *macropetala*. However, the wording of Diels also may indicate that he was not sure of the petal colour in their living state and thus only stated that they are white in the dried condition. The deep red, purplish red, reddish orange, or deep pink petal colours found in other members of the complex are usually well-preserved even in 100+ year-old specimens, provided they have been stored under favourable conditions. Many of the *J. Drummond coll. VI n. 109* specimens of *D. macropetala* still clearly show the reddish inner part of their petals. It is therefore astonishing that the unique flower colour pattern escaped the notice of Diels, the taxon’s author, and indeed of later botanists who studied the widely available material; the gathering *J. Drummond coll. VI n. 109* consists of numerous duplicates (the herbarium sheets studied by the authors of the present study [see “Types”] comprise ca. 140 individuals of that taxon) and it was apparently distributed to several major European herbaria by Drummond at the time (syntypes were found in twelve herbaria, see “isolectotypes”).

The initial collector, James Drummond, referred to *D. macropetala* as “[t]he most beautiful of the genus *Drosera*” [18,19] and he also clearly described the distinctive flower colour pattern of this taxon as “flowers […] white with a crimson eye, and they are beautifully variegated with crimson veins” [18,19]. These earlier mentions were however apparently overlooked by Ludwig Diels when he described this distinctive *Drosera* as new to science in 1906 [8].

**Distribution and habitat:** *Drosera macropetala* is known from the Dandaragan Plateau between Dandaragan and Mogumber, about 100–150 km north of Perth (Figure 3). The species appears to be restricted to the upper slopes of lateritic hills where it grows in low heath in poorly drained, sandy clay with laterite. Possibly also occurs in open *Eucalyptus* (“white gum”) forest [18,19].

**Phenology:** Flowering has only been recorded in August.

**Conservation status:** Recommended for listing as Priority One (poorly known species) under Conservation Codes for Western Australian Flora and Fauna (Western Australian Herbarium 1998–; https://florabase.dpaw.wa.gov.au/ (accessed on 6 December 2022)). Endangered (EN) under IUCN Red List criteria B1ab(iii,iv,v)+2ab(iii,iv,v) and C2a(i) following IUCN [29]. The extent of occurrence (EOO) and area of occurrence (AOO) of *D. macropetala* is estimated at ca. 200 km^2^ and 16 km^2^, respectively. These numbers assume that the Mogumber population, last documented in 1904 by Alexander Morrison (see “Specimens examined”), still exists today. Given the extensive vegetation clearing in this area after that year [40], it is possible that this population has been destroyed, in which case both EOO and AOO would be <10 km^2^, meeting the Critically Endangered (CR) criteria [29]. *Drosera macropetala* is not known to occur on any land managed by the Western Australian Department of Biodiversity, Conservation and Attractions (DBCA) and is thus potentially threatened by future vegetation clearing.

This species was re-located by Declared Rare Flora monitors Gail and Dannielle Reed in August 2020 near Dandaragan, having not been recorded or documented since 1904. Their photographs were uploaded to Facebook and the plants depicted were immediately recognised as an unknown taxon by the authors. Subsequent targeted surveying of this area during 2020, 2021, and 2022 located a total of four (sub-)populations in a single, very narrow strip of unprotected remnant roadside vegetation (T. Krueger pers. obs.). These narrow, linear vegetation corridors, which transect completely cleared agricultural and urban areas, are highly susceptible to road maintenance and construction, altered hydrology, and weed infestation [16]. Population sizes of these four (sub-)populations vary from 2 to ca. 200 mature individuals and the total population in this area is estimated to consist of ca. 500 mature individuals. The historically reported population(s) from near Mogumber (which is ca. 50 km south-east of Dandaragan) have not yet been re-located by the authors as of this publication and their population size and persistence is currently unknown. Unlicenced collectors and illegal commercial/horticultural trade could pose an additional threat to *D. macropetala* in the future given its extremely small population sizes and the unfortunate tendency for poachers to target rare carnivorous plant species [16]. Further surveys are strongly recommended to gain a better understanding of this taxon’s distribution, number and size of populations, and to identify further potential threats.

**Notes on the lectotypification:** Lectotypification of *D. macropetala* is required as Diels [8] did not select a type specimen from the duplicates of *J. Drummond coll. VI n. 109*, which all constitute syntypes. Marchant et al. [23] designated an “isotype” (which is not an inadvertent lectotypification following ICN Arts. 7.11 and 9.10 [26]) and, while K000659191 was labelled as the “holotype” by Marchant in 1985, this was not effectively published and also does not constitute a lectotypification (ICN Arts. 7.10, 7.11, and 9.10 [26]). Even if it had been validly published, the K specimen was incorrectly selected by Marchant as it cannot be the holotype (i.e., the material consulted by the taxon author for the description); Diels visited K to annotate the specimen after the publication of his *D. microphylla* var. *macropetala* in 1906 (evident from the fact that his annotation slip on the K specimen—in contrast with that on the B material—does not bear the label head “bearbeitet für das “Pflanzenreich”” [seen for Diels’s taxonomic revision of *Drosera*, i.e., [8]]), which Diels made strict use of for all specimens he examined for his *Drosera* monograph [8]. Lowrie [1] (p. 354) erroneously assumed that both Bentham’s *D. calycina* var. *minor* and Diels’ *D. microphylla* var. *macropetala* are based on the same sheet (K000215038), which is *J. Drummond coll. VI n. 110*, even though Diels [8] clearly states in his description of *D. microphylla* var. *macropetala* that it is based on *J. Drummond coll. VI n. 109*.

Specimen B 100755976 features an identification slip in the hand of the taxon’s author Ludwig Diels and, also given that he worked at B, represents the obvious choice for a lectotype.

At KFTA herbarium, a Drummond specimen has been indicated as “type material” of *D. macropetala* (KFTA0003370 photo!; identified as an “isotype” of “*Drosera macrosepala*” [*sic.!*] in 2013), however, this specimen is not *J. Drummond coll. VI n. 109*, nor does it agree with the locus classicus for the taxon. Rather, it corresponds to *Drummond s.n.*, a collection of *D. menziesii* (the original label reads “Swan River Drummond”, to which a pencil-written “n. 109” has been added in error later).

**Notes on Drummond’s type collection:** The syntypes of *J. Drummond coll. VI n. 109* only provide the rough locality information “Western Australia, between Moore River and Murchison Rivers” (locality not indicated on the lectotype specimen in B). However, more precise information on the locus classicus comes from Drummond’s newspaper contributions ”The Botany of the North-western Districts of Western Australia“ [18], republished by Hooker [19]. There, he describes a *Drosera* species with white flowers with crimson centres, large glabrous sepals exceeding the petals in size, and flowers that close at night or during rainy weather, a description that exactly matches *D. macropetala*. Drummond [18,19] mentions that this species “[...] grows abundantly in a White Gum forest about four miles to the north of Dundaragan [Dandaragan]”, a locality very close to where it still can be found today (T. Krueger pers. obs.). As *J. Drummond coll. VI n. 109* is the only collection of *D. macropetala* provided by Drummond, it is safe to conclude that this is the collection locality. Additional support for this comes from Barker [41], who evidenced that Drummond’s newspaper contribution [18] is referring to Drummond’s VI collection series, i.e., the series containing the type collection of *D. macropetala*. This means that the year of collection (not given on any of the syntype specimens) is 1850 or 1851, as for all specimens comprising the VI collection [30].

**Additional specimens examined:** AUSTRALIA. Western Australia: Between Gillingara [*Gillingarra*] + Mogumber, Moore River, 18 August 1904, *A. Morrison s.n.* (E00794029 photo!); Mogumber, Moore River, 18 August 1904, *A. Morrison s.n.* (E00794031 photo!); Dandaragan [precise locality withheld for conservation purposes], upper hillslope, low kwongan heath, sandy clay with laterite gravel, 22 August 2021, *T. Krueger 29* (PERTH!).

### 3.7. **Drosera microphylla**
*Endl., Enum. Pl. (Endlicher): 6 (1837). (Figure 3, Figure 4, Figure 5 and Figure 14)*

**Holotype:** AUSTRALIA. Western Australia: King Georges [George] Sound, without date [likely collected in 1833 or 1834], *C.A.A. von Hügel s.n.* (W 0009732!).

≡ *Sondera microphylla* (Endl.) Chrtek & Slavíková, Novit. Bot. Univ. Carol. 13: 44 (2000).

**Figure 14 biology-12-00141-f014:**
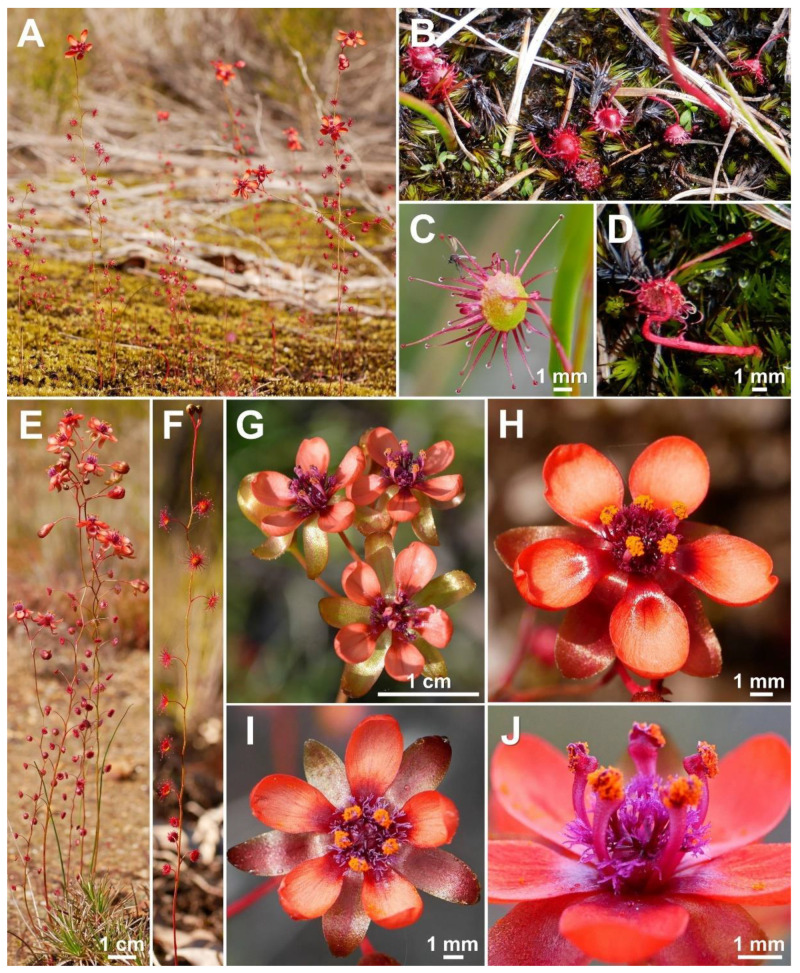
*Drosera microphylla* Endl. (**A**) group of plants; (**B**) detached leaves; (**C**) lamina, abaxial view; (**D**) detached leaf with adventitious dropper shoot emerging from central adaxial side of the lamina; (**E**) habit of a group of flowering plants; (**F**) habit of a single plant with developing flower buds, note the large leaf-free gaps along the stem due to detached leaves; (**G**,**H**) flowers in bright sunlight; (**I**) flower in diffuse light; and (**J**) stamens and styles. (**A**,**E**,**H**) from near Kentdale, Western Australia, 26 August 2022. (**B**,**D**) from near Denmark, Western Australia, 15 September 2021. (**C**,**I**,**J**) from Mount Frankland North National Park, Western Australia, 13 September 2021. (**F**) from near Walpole, Western Australia, 25 August 2022. (**G**) from near Denmark, Western Australia, 4 September 2014. Images: T. Krueger.

**Description:** Tuberous perennial herb, (7–)11–32(–51) cm tall above ground including inflorescence. **Tuber** subglobose, ca. 5 mm in diameter, enclosed in black papery sheaths from previous seasons’ growth. **Stem** (subterranean part) 2.2–8.0 cm long, 1.5–2.2 mm in diameter, enclosed in brown, fibrous tunic formed from previous seasons’ stems and roots. **Roots** few, fibrous, emerging laterally from along subterranean part of stem, mostly immediately above tuber. **Stem** (epigeous part) erect, self-supporting, simple, terete, straight or slightly to strongly fractiflex, glabrous, (6–)10–24(–43) cm tall, 0.4–0.9 mm in diameter near soil surface, 0.3–0.9 mm in diameter at internodes, red to yellowish green, always red near soil surface. **Cataphylls** 3–7 on lower part of stem, subulate, 1.2–2.5 mm long, ca. 0.5 mm wide, red. **Leaves** solitary on each node, alternate, (0–)4–23 present in flowering individuals, caducous, leaves frequently detach randomly from stem before and during anthesis (i.e., leaving large gaps along stem); internodes 2–23 mm. **Petioles** terete, semi-erect, arcuated abaxially (downwards) along whole length or arching increasing gradually towards the lamina, glabrous, 4–12 mm long, 0.3–0.7 mm wide at base, tapering to 0.1–0.3 mm towards the lamina, red or sometimes yellowish green. **Lamina** peltate, orbiculate or sometimes orbiculate with very slightly flattened adaxial lateral margin, shallowly concave, adaxial surface facing outwards or slightly downwards, (1.5–)2.0–3.5 mm long, (1.5–)2.0–3.8 mm wide; lamina adaxial surface covered with stalked, carnivorous, secretive capitate glands (tentacles); tentacles 1.5–4.5 mm long at lamina margin, decreasing in size towards centre of lamina, stalk red or sometimes red in lower half with yellowish green upper half; lamina abaxial surface glabrous but microscopically punctate. **Inflorescence** a 1–7-flowered scorpioid cyme, terminal, simple, single-sided, (1.0–)2.5–6.0(–7.5) cm long. **Peduncle** terete, (0.5–)1.2–4.5 cm long, (0.2–)0.3–0.7 mm in diameter, glabrous, red, reddish orange, or yellowish green. **Pedicels** terete, erect or semi-erect in fruit, (5–)7–12(–15) mm long in fruit, 0.2–0.7 mm in diameter, spaced by 3–8 mm along rhachis, glabrous, red, reddish orange, or yellowish green. **Bracts** spathulate, narrowly obovate, elliptic, lanceolate, subulate or oblong, often slightly concave and arcuated adaxially (upwards), apex entire or rarely irregularly crenulate, 1.5–2.8 mm long, 0.4–1.1 mm wide, glabrous. **Sepals** 5, narrowly elliptic to narrowly obovate, arcuated adaxially (upwards), slightly concave, often reflexed during anthesis, apex entire or crenulate, 5–9 mm long, 2.2–3.1(–3.7) mm wide, abaxial surface microscopically glandular (appearing glabrous), red to yellowish brown, often with 3–5 red veins. **Corolla** 8–15 mm in diameter. **Petals** 5, reddish orange with deep red base, narrowly obovate to spathulate, deeply concave and slightly arcuated adaxially (upwards), apex rounded and entire, 4.5–7.3 mm long, 2.4–4.1 mm wide. **Stamens** 5, 2.5–3.5 mm long. **Filaments** slightly dilated towards apex, 0.2–0.3 mm wide at base, 0.3–0.6 mm wide near apex, deep red in lower half, reddish purple in upper half. **Anthers** bithecate, retrorse, 0.6–1.0 mm wide, thecae yellow to orangey yellow. **Pollen** yellow to orangey yellow. **Ovary** obovoid, 3-carpellate, fused, 1.2–1.7 mm in diameter, deep red. **Styles** 3, divided into a few filiform segments just above the base, style segments again divided into many terete style segments, forming a crowded tuft, extending laterally beyond the filaments, 1.6–2.5 mm long, deep red, reddish purple near stigmas. **Stigmas** simple or shortly branched, at tips of style segments, 0.3–0.7 mm long, reddish purple. **Seeds** narrowly obtrullate to narrowly obovate, straight or slightly curved, outline rectangular with slight ellipsoid swelling in the proximal and distal half, funicular (upper) end truncate, basal (chalazal) end pointed with obtuse tip, 1.6–1.8(–2.0) mm long, 0.40–0.65 mm wide, testa pale brown, only the middle part blackish brown; testa more or less isodiametrically reticulate, with anticlines thin and only shallowly raised.

**Etymology:** The specific epithet is derived from the Greek *micros* (=small) and *phyllon* (=leaf), referring to the small leaves of this species.

**Taxonomic notes:** *Drosera microphylla* is morphologically similar to *D. koikyennuruff* and *D. reflexa*, from which it is distinguished by (contrasting characters in parentheses): (1) its reddish orange petal colour (petals dark red in *D. koikyennuruff* or purplish pink with deep red base in *D. reflexa*); (2) its tendency to detach leaves during and prior to anthesis, often leaving large leaf-free gaps along stem, except in populations from the Stirling Range and Mt. Lindesay (leaves do not detach before or even after anthesis); (3) its late flowering time from August to October (from June to early September); and (4) its reddish purple stigma colour (stigmas deep red or dark red). It is further distinguished from *D. koikyennuruff* by: (1) its tentacle stalk colour, which is red or red in lower half with upper half yellowish green (tentacle stalks yellowish green), and (2) its preference for mossy wet areas on and near granite outcrops, seasonally wet swamps, or rocky mountain slopes (relatively dry sandy habitats in open Mallee woodlands).

*Drosera microphylla* has historically often been confused with *D. calycina*, from which it can be distinguished by (contrasting characters in parentheses): (1) its lamina shape, which is orbiculate or sometimes orbiculate with very slightly flattened upper margin (lamina reniform or orbiculate with flattened, often truncated upper margin); (2) its comparatively small flowers with a corolla diameter of 8–15 mm (corolla diameter 14–20 mm); (3) its petal colour, which is reddish orange with deep red base (petals deep red in inner half transitioning to purplish red in outer half); (4) its stamen length, which reaches only 2.5–3.5 mm (stamens 4.0–6.5 mm long); and (5) its styles, which extend laterally beyond the filaments and have reddish purple stigmas (styles not laterally extending beyond the filaments, with deep red stigmas).

The leaves of *D. microphylla* frequently detach from near the petiole bases, mostly shortly before and during anthesis, resulting in large, leaf-free “gaps” along the stem that are distinctive for this species (Figure 14B,F). In some cases, mature flowering individuals have been observed with only two or three widely separated leaves still attached to the stem, the remainder having been shed. This is also very apparent in the majority of herbarium specimens of *D. microphylla*. For example, *Cranfield & Ward 25110* (PERTH 08507929!) has at least ten leaf nodes but only three with leaves still attached to them. However, in populations from Stirling Range National Park and Mt. Lindesay, the leaves do not appear to detach at all (T. Krueger pers. obs.).

Leaf detachment in most *D. microphylla* populations appears to serve a role in clonal propagation. In at least four populations near Denmark and Walpole, a red, prostrate, adventitious dropper shoot was observed to emerge from near the centre of the adaxial (tentacle-bearing) lamina surface, directly opposite the point of petiole attachment on the abaxial side (so-called epiphyllous budding; Figure 14D). This is congruent with observations made of naturally occurring asexual regrowth from both basal rosette and stem leaves in the tuberous *Drosera auriculata* Backh. ex Planch. and *D. peltata* Thunb., both likewise from *D.* section *Ergaleium* (depicted and described in detail by Vickery [42]) and from artificially detached leaf cuttings reported for other erect tuberous *Drosera* in cultivation [43] (A. Fleischmann and G. Bourke pers. obs.). Since the detached leaves of *D. microphylla* fall on soils that are still wet at that time of year (from July to September), these adventitious droppers can form a new tuber from their tips once they penetrate the soil, prior to the onset of the dry summer conditions. Leaf detachment thus appears to be a strategy for *D. microphylla* to quickly colonise the bare mossy or sandy soils of its preferred open habitats and may have arisen in connection with the usually very wet seepage soils this species grows in.

The ability to propagate clonally by epiphyllous budding from the leaves mirrors that of perennial *Drosera* in South Africa and Latin America, where only those species growing in rather wet habitats have the capacity to readily multiply asexually via leaf cuttings, something closely related species from drier habitats are incapable of [44]. Gibson [43] observed that, in cultivation, some erect tuberous *Drosera* do multiply through adventitious epiphyllous budding from artificially detached leaves, while others do not; again, there seems to be some connection with whether the species’ natural habitats are wet or not. Although clonal propagation by budding from leaves still attached to the basal rosette and stem has been reported to occur naturally in *D. auriculata* and *D. peltata* [42], *D. microphylla* thus far is the only tuberous *Drosera* known to employ its stem leaves for vegetative propagation after shedding them from the mother plant and, generally, it seems to be the only tuberous *Drosera* species with caducous leaves.

*Drosera microphylla* is a very variable species and at least two collections exist that potentially represent additional undescribed taxa, the precise status of which could not be determined during this work. *Coffey 103 A* (PERTH 08468338!) comprises a collection from near Hopetoun, about 200 km east of the confirmed distribution area of *D. microphylla* (this collection is marked with “3” in Figure 3). While the specimen appears to match *D. microphylla* in morphological detail and could indeed represent an outlying population of this species, additional in situ observations are required to confirm its identity. Despite multiple attempts between 2019 and 2022, the authors of this work failed to re-locate this population.

*Newbey 4204* (PERTH 00666459!; marked with “2” in Figure 3) represents a collection from near Boxwood Hill, collected in June 1974. While the flowering time would match that of *D. koikyennuruff*, the petal colour appears to be orange, while the overall habit more closely resembles that of *D. microphylla*. Further in situ observations of this population are required to determine whether it represents *D. microphylla*, *D. koikyennuruff*, or an undescribed taxon.

In addition, red-flowered plants with a different seed shape are known to co-occur with *D. microphylla* at sites along the south coast [15] (G. Bourke pers. obs.). While these have been included within *D. microphylla* in the present work, additional studies are recommended to evaluate whether these represent intraspecific variation of *D. microphylla* or a distinct taxon.

Previously published illustrations labelled as “*D. microphylla*” depict *D. calycina* [22] (p. 40, illustration 2) and a mixture of *D. calycina* with either *D. hortiorum*, *D. macropetala*, or *D. rubricalyx* [12] (p. 65) (see Taxonomic notes under *D. calycina*). Lowrie [1] (p. 609) correctly depicts *D. microphylla*, although the large leaf-free gaps evident on the cited specimen *A. Lowrie 3044* (PERTH 08692637!) are not illustrated. The illustration by Diels [8] (p. 120, illustration E, F) appears to be based on *Diels 3009* (B 100755996!), the precise identity of which could not be determined during this work (see Taxonomic notes under *D. koikyennuruff*).

**Distribution and habitat:** *Drosera microphylla* occurs from Walpole to Cheynes Beach and throughout the Stirling Range (Figure 3). It grows in mossy wet areas on and near granite outcrops, seasonally wet seepage slopes near large swamp systems, rocky mountain ridges, and slopes. Soil usually sandy clay or peat, sometimes with laterite gravel or rocks.

**Phenology:** Flowering has been recorded from August to early October.

**Conservation status:** Not eligible for Conservation Code listing and Least Concern (LC) according to the IUCN Red List Criteria, following Cross [45]. Not threatened, locally common, and widespread, occurring across many reserves managed by the Western Australian Department of Biodiversity, Conservation and Attractions (DBCA).

**Notes on the type collection:** The type collection of *D. microphylla*, *Hügel s.n.* (W 0009732!), comprises a single plant collected from near Albany (King George Sound). The relatively long, straight (i.e., not fractiflex) stem and narrow petals indicate that it does indeed belong to the orange-flowered plants from the area and not to those newly described here as *D. koikyennuruff* and *D. reflexa*. While the type specimen appears to feature leaf-free gaps along the stem, it is unknown whether this is a result of the distinctive leaf detachment habit or whether they detached later in the brittle herbarium material. The large, separated leaf lying next to the lower part of the stem is a leaf of *D. macrantha*, which often co-occurs with *D. microphylla* in its preferred granite outcrop habitat around Albany where this collection was made.

**Additional specimens examined:** AUSTRALIA. Western Australia: King George’s Sound, August 1898, *Goadby s.n.* (PERTH 666939!); 3 miles W of Denmark on Nornalup Road, large sloping granite outcrop, 19 July 1965, *N.G Marchant 6570* (PERTH 666408!); 3 miles W of Denmark on Nornalup Road, large sloping granite outcrop, 19 July 1965, *N.G Marchant 6570* (PERTH 666408!); Tick Flat, halfway between Mount Gardner and the Reserve Office on the lower slopes of Mount Gardner, Two Peoples Bay Nature Reserve, 29 August 1973, *G.T. Smith & L.A. Moore s.n.* (PERTH 5294738!); on granitic dome ca. 5 miles W of Denmark along rd. to Walpole, 17 October 1974, *L. Debuhr 4145* (RSA0229908 photo!); Lower slopes of Mount Manypeaks, on granite rocks, 25 August 1980, *D. Davidson 30 A* (PERTH 04546628 photo!); W of Waychinicup, moss over granite, 27 August 1980, *D. Davidson s.n.* (PERTH 04546601photo!); corner Narrikup Road and Albany Highway, grey sandy loam soil, in association with a Eucalyptus sp. and Banksia sp. grove, 27 August 1984, *E.J. Croxford 3425* (PERTH 04546636 photo!); 50 m up walk-track from Denmark River, Denmark, Mount Lindesay, granitic sand, on and around granitic slabs on steep slope, 3 September 1990, *B.G. Hammersley* (PERTH 1188100!); Thompson Road, open flat, grey peaty sands, 5 August 1994, *R.W. Hearn ARA 4384* (PERTH 4127528!); Foot of Bluff Knoll (mountain), 500 m from the ‘peak walk’ carpark, Stirling Range, plants growing on ridge with water running shallow soils, brown humous sandy loam, 25 September 1994, *W. Bopp 119* (PERTH 4284968!); Mount Lindesay summit (207), Denmark, grey sand on granite, 11 October 1994, *S. Barrett 176* (PERTH 4213327!); Mount Lindesay, Q207, hillside—summit, grey clayey sand over granite, 31 August 1995, *S. Barrett 616* (PERTH 4273699!); Centre Road from South Western Highway, hillside, lateritic pebbles, granite rocks, brown dry soils, 27 September 2003, *S.C. Coffey 19* (PERTH 7998236!); Near junction South Coast Highway and Lapko Road, Shadforth, near Denmark, grows in skeletal, gritty, black silt soils covered with moss on the aprons of granite outcrops, 3 September 2004, *A. Lowrie 3044* (PERTH 8692637!); Stirling Range National Park, below Bluff Knoll, valley and rocky, brown with green layer (possibly algae) with clayey loam soil, 9 September 2009, *S.C. Coffey 101* (PERTH 8468362!); Ca. 330 m W along Little Lindesay walk trail from Stan Road to first granite outcrop, N of trail, granite outcrop, reserve, dry yellow sand/loam, 1 September 2010, *J. Liddelow JAL 141* (PERTH 8951691!); Surprise forest block, 1.8 km ENE of Western Road along Mountain Road, hill slope, bare to littered dry yellow to brown clayey sand soil over granite, 22 September 2010, *R.J. Cranfield & B.G. Ward 25110* (PERTH 8507929!). King George’s Sound, without date, *A. Collie s.n.* (BM014605114 photo!); De Swan River au Cap[e] Riche [most duplicates only mention as locality “Sw. R.”=Swan River; only the specimen from de Candolle’s herbarium at G provides the full information on the collection locality, the one from Boissier’s herbarium does not], without date [collected in 1847 or 1848 [30]], *J. Drummond* [*coll. V*] *n. 282* (BM014605112 photo!, G-6988-420! [inventory number, not a barcode], G s.n.!, P04963085 photo!, K000215091!, W0131692!).

### 3.8. **Drosera reflexa**
*G.Bourke & A.S.Rob.*, **sp. nov.** (*Figure 3, Figure 4, Figure 5, Figure 15 and Figure 16*)

**Type:** AUSTRALIA. Western Australia: Kentdale [precise locality withheld for conservation purposes], shallow peaty soil on and near the margins of a granite outcrop, 31 August 2018, *G.J. Bourke 458* (holotype PERTH!; isotypes MEL2500514A!; MEL 2500512A! [spirit collection]).

**Figure 15 biology-12-00141-f015:**
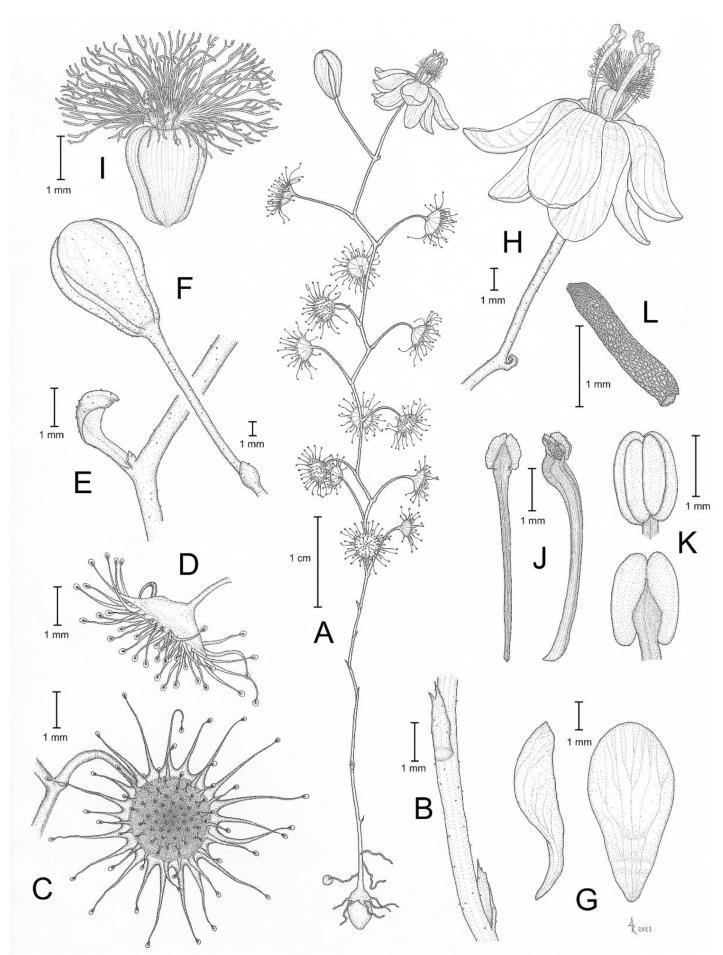
*Drosera reflexa* G.Bourke & A.S.Rob. (**A**) habit (tunic not depicted); (**B**) stem base with two cataphylls; (**C**) lamina, adaxial view; (**D**) lamina, lateral view; (**E**) peduncle and bract; (**F**) pedicel and flower in bud; (**G**) petals, left side view, right adaxial view; (**H**) flower, lateral view; (**I**) gynoecium; (**J**) stamens, left dorsal view, right lateral view; (**K**) thecae, top ventral view, bottom dorsal view; and (**L**) seed. (**A**–**K**) from the type (*G.J. Bourke 458*, spirit material), (**L**) from SEM images. Drawing: A. Robinson.

**Figure 16 biology-12-00141-f016:**
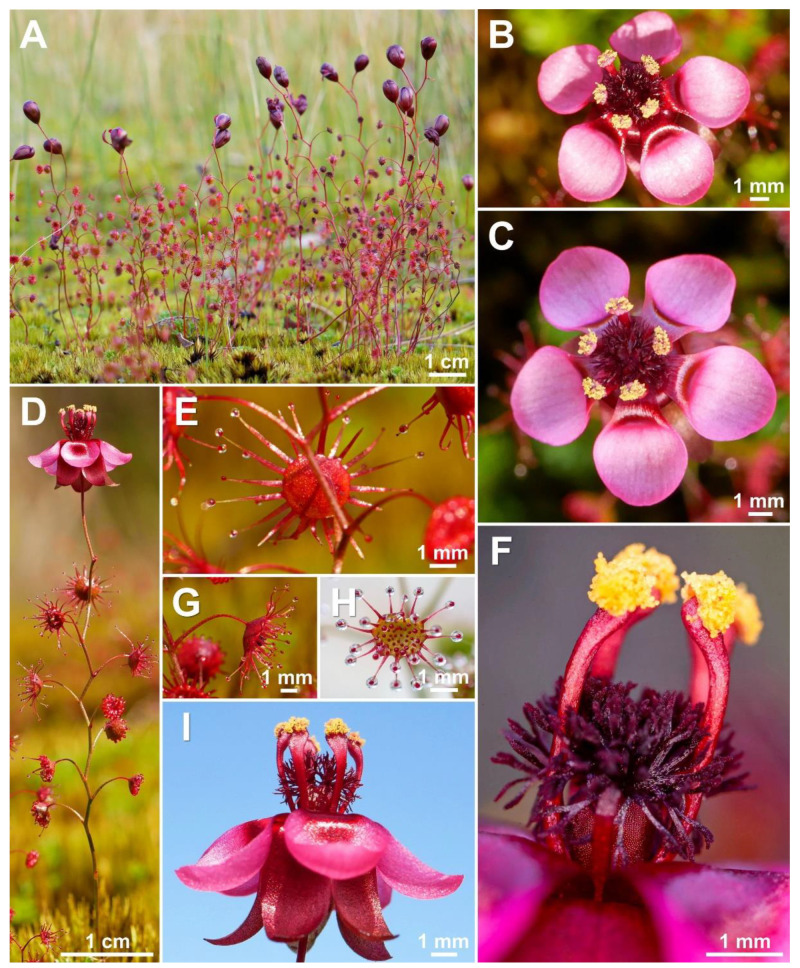
*Drosera reflexa* G.Bourke & A.S.Rob. (**A**) group of plants; (**B**) flower in bright sunlight; (**C**) flower in diffuse light; (**D**) habit; (**E**) lamina, abaxial view; (**F**) stamens and styles; (**G**) petiole and lamina, lateral view; (**H**) juvenile lamina, adaxial view; and (**I**) flower, lateral view. (**A**–**E**,**G**,**I**) from near Kentdale, Western Australia, 26 August 2022; images by T. Krueger. (**F**,**H**) from cultivated material originating from near Kentdale, Western Australia; images by G. Bourke.

**Diagnosis:***Drosera reflexa* is morphologically most similar to *D. esperensis* Lowrie, *D. microphylla* Endl. and *D. koikyennuruff* T.Krueger & A.S.Rob. It differs from *D. esperensis* by (contrasting characters in parentheses): (1) its filament shape, which is strongly dilated towards apex, 0.5–0.9 mm wide near apex (filaments ± linear, 0.3–0.6 mm wide near apex); (2) its petal colour, which is purplish pink with a deep red base (petals white with pale purplish red base); and (3) its flowering time from June to early September (flowering late August to October, sometimes until December). It differs from *D. microphylla* by (contrasting characters in parentheses): (1) its leaves, which remain attached to the stem even post-anthesis (leaves often detaching during and prior to anthesis, leaving large leaf-free gaps along stem, except in populations from the Stirling Range and Mt. Lindesay); (2) its stamen length of 3.8–4.4 mm (stamens 2.5–3.5 mm long); (3) its petal colour, which is purplish pink with deep red base (petals reddish orange with deep red base); (4) its stigma colour, which is dark red (stigmas reddish purple); and (5) its flowering time from June to early September (flowering from August to October). It differs from *D. koikyennuruff* by (contrasting characters in parentheses): (1) its tendency to form dense populations via adventitious stolons (plants in sparse populations, not colony-forming); (2) its petal shape, which is obovate to very broadly obovate (petals narrowly obovate to broadly spathulate); (3) its petal colour, which is purplish pink with deep red base (petals dark red); (4) its red to purplish red sepals (sepals yellowish brown to greenish yellow); and (5) its habitat preference of shallow moss on granite outcrops between Walpole and Denmark (plants occurring in sandy soils in open Mallee woodlands in and around the Stirling Range).

**Description:** Tuberous perennial herb, 5–15(–25) cm tall above ground including inflorescence. **Tuber** subglobose, 2–4 mm in diameter, enclosed in black papery sheaths from previous seasons’ growth, pink to red. **Stem** (subterranean) 1.8–5.0 cm long, 0.8–1.2 mm in diameter, enclosed in brown, fibrous tunic formed from previous seasons’ stems and roots. **Roots** few, fibrous, emerging laterally from along subterranean vertical stem. **Stolons** few, laterally produced on subterranean stem producing tubers as plants enter dormancy. **Stem** (epigeous part) erect, self-supporting, simple, or occasionally branching from the base, slightly to strongly fractiflex, glabrous throughout, 4–15 cm tall, 0.3–0.8 mm in diameter, yellowish green fading to red towards the end of the season, often red near soil surface. **Cataphylls** 2–8 on lower part of stem, subulate, 1.2–2.3 mm long, ca. 0.5 mm wide, red. **Leaves** solitary on each node, irregularly alternate, 7–18 present in flowering individuals; internodes 2–8 mm. **Petioles** terete, semi-erect, arcuated abaxially (downwards) along whole length, occasionally arching increasing gradually towards the lamina, glabrous to microscopically punctate, 3–8 mm long, 0.3–0.5 mm wide above slightly thickened base, tapering to 0.1–0.2 mm in diameter towards the lamina, yellowish green to red, often darker near lamina. **Lamina** peltate, orbiculate, occasionally with very slightly flattened adaxial lateral margin, shallowly concave, adaxial surface facing outwards or slightly downwards, 1.5–3.5 mm long, 1.8–3.5 mm wide; lamina adaxial surface covered with stalked, carnivorous, secretive capitate glands (tentacles); tentacles 1.7–4.0 mm long at margin, decreasing in size towards centre of lamina, stalk red throughout or red in lower half with greenish yellow upper half; lamina abaxial surface glabrous but microscopically, sparsely punctate. **Inflorescence** a 1–3(–5)-flowered scorpioid cyme, terminal, simple, single-sided, 1.0–3.2(–4.9) cm long. **Peduncles** terete, 0.3–1.9(–4.0) cm long, 0.2–0.5 mm in diameter, microscopically glandular (appearing glabrous), reddish orange to red or rarely yellowish green. **Pedicels** terete, semi-erect or erect in fruit, 6–18 mm long in fruit, 0.2–0.4 mm in diameter, spaced by 2–9 mm along rhachis, microscopically glandular (appearing glabrous), usually more reddish than peduncle. **Bracts** spathulate or narrowly obovate, concave, arcuated adaxially (upwards), apex entire or crenulate, 1.7–3.0 mm long, 0.6–1.0 mm wide, abaxial surface minutely glandular (appearing glabrous). **Sepals** 5, narrowly obovate to narrowly elliptic, arcuated adaxially (upwards), slightly concave, often reflexed during anthesis, apex entire or crenulate, 5.2–8.3 mm long, 2.2–3.3 mm wide, abaxial surface minutely glandular (appearing glabrous), red to purplish red. **Corolla** 9–15 mm in diameter. **Petals** 5, purplish pink with large, deep red blotch towards the base, obovate to very broadly obovate, rarely broadly spathulate, deeply concave and slightly arcuated adaxially (upwards), margins entire, 4.2–7.0 mm long, 2.5–4.3 mm wide. **Stamens** 5, 3.8–4.4 mm long. **Filaments** dilated towards apex, 0.2–0.3 mm wide at base, 0.5–0.9 mm wide near apex, deep red. **Anthers** bithecate, retrorse, 0.9–1.1 mm wide, thecae pale yellow to orangey yellow. **Pollen** yellow to orangey yellow. **Ovary** obovoid, 3-carpellate, fused, 1.5–1.9 mm in diameter, deep red. **Styles** 3, divided into a few filiform segments just above the base, style segments again divided into many terete to distally flattened segments, forming a crowded tuft, extending laterally just beyond filaments, 1.5–1.8 mm long, dark red. **Stigmas** simple or shortly branched, at tips of style segments, papillose, ca. 0.2 mm long, dark red. **Seeds** narrowly elliptical to narrowly obovate, outline more or less narrowly rectangular, rarely curved, funicular (upper) end truncate or with shallow funicular disc, basal (chalazal) end acute to obtuse, 1.3–1.8 mm long, 0.3–0.4 mm wide, dark brown, appearing black, tips pale brown; testa more or less longitudinally reticulate, with anticlines thin and only shallowly raised.

**Etymology:** The specific epithet is derived from the Latin *reflexus* (=turned back or away) and refers to the often strongly reflexed (by up to ca. 140–170° with respect to the floral axis) sepals and petals.

**Taxonomic notes:** The overall habit as well as petiole, lamina, and style shape of *D. reflexa* indicate that it is morphologically most similar to *D*. *esperensis*, *D. koikyennuruff*, and *D. microphylla*. In living specimens, the petal colour can be easily used to differentiate the species within this group (purplish pink with deep red base in *D. reflexa*, white with pale purplish red base in *D. esperensis*, dark red in *D. koikyennuruff*, and reddish orange with red base in *D. microphylla*). *Drosera reflexa* is found in close proximity to *D. microphylla* (ca. 200 m) but the two taxa do not co-occur (no syntopic occurrence) despite their very similar habitat preferences. No hybrids or intermediates between the two species have been observed. This taxon was first mentioned by Lowrie et al. [15] (p. 266) under *D. microphylla* as “a diminutive form with bi-coloured red and pink flowers”.

**Distribution and habitat:** Kentdale (between Walpole and Denmark near the south coast of Western Australia; Figure 3). Occurs in shallow decomposed granitic soils over granite lenses in mosses.

**Phenology:** Flowering has been recorded from June to early September.

**Conservation status:** Listed as Priority Two (poorly known species) under Conservation Codes for Western Australian Flora and Fauna (Western Australian Herbarium 1998–; https://florabase.dpaw.wa.gov.au/ (accessed on 6 December 2022)), under the phrase-name “*Drosera* sp. Kentdale (G.J. Bourke 458)”. It is Critically Endangered (CR) under IUCN Red List criteria B1ab(iii,v)+2ab(iii,v) following IUCN [29]. *Drosera reflexa* is only known from a single population that is partially located on land managed by the Western Australian Department of Biodiversity, Conservation and Attractions (DBCA). Targeted surveys in the region in 2002, 2017, 2019, 2020, 2021, and 2022 were unable to identify any additional populations despite considerable areas of apparently suitable habitat being surveyed (G. Bourke and T. Krueger pers. obs.). Suitable habitat on nearby private land was not surveyed and may yield additional remnant populations. The total population size is estimated at ca. 1000 mature individuals. Damage to the habitat by recreational vehicles has been observed (G. Bourke pers. obs.) and significant invasive weed infestation is apparent in parts of the habitat (T. Krueger pers. obs.). Further surveys are recommended to gain a better understanding of this taxon’s distribution, number and size of populations, and to identify additional potential threats.

### 3.9. **Drosera rubricalyx**
*T.Krueger & A.Fleischm.*, **nom. nov.** (*Figure 3, Figure 4, Figure 5, Figure 17 and Figure 18*)

**Type:** AUSTRALIA. W. [Western] Australia. Between Moore & Murchison Rivers, without date [“1853” written on the herbarium label is the accession date at K, not the actual collection date], the year of collection is either 1850 or 1851 [30]], *J. Drummond* [*coll. VI n.*] *110* (holotype K000215038!; isotypes: BM014605113 photo!; FI011165 photo!; G00410323!; LD1971779 photo! (mixed collection, two individuals belong to *J. Drummond coll. VI n. 110*, the remainder being *J. Drummond coll. VI n. 109*, *D. macropetala*); MEL 97061!; OXF00140704!; P00713914 photo!; P00749101 photo!).

≡ *Drosera calycina* var. *minor* Benth., Fl. Austral. 2: 469 (1864).

**Lectotype (designated here):** AUSTRALIA. W. [Western] Australia. Between Moore & Murchison Rivers, without date [actual year of collection is either 1850 or 1851 [30]], *J. Drummond* [*coll. VI n.*] *110* (K000215038!; isolectotypes: BM014605113 photo!; FI011165 photo!; G00410323!; LD1971779 photo! (mixed collection, two individuals belong to *J. Drummond coll. VI n. 110*, the remainder being *J. Drummond coll. VI n. 109*, *D. macropetala*); MEL 97061!; OXF00140704!; P00713914 photo!; P00749101 photo!).

**Figure 17 biology-12-00141-f017:**
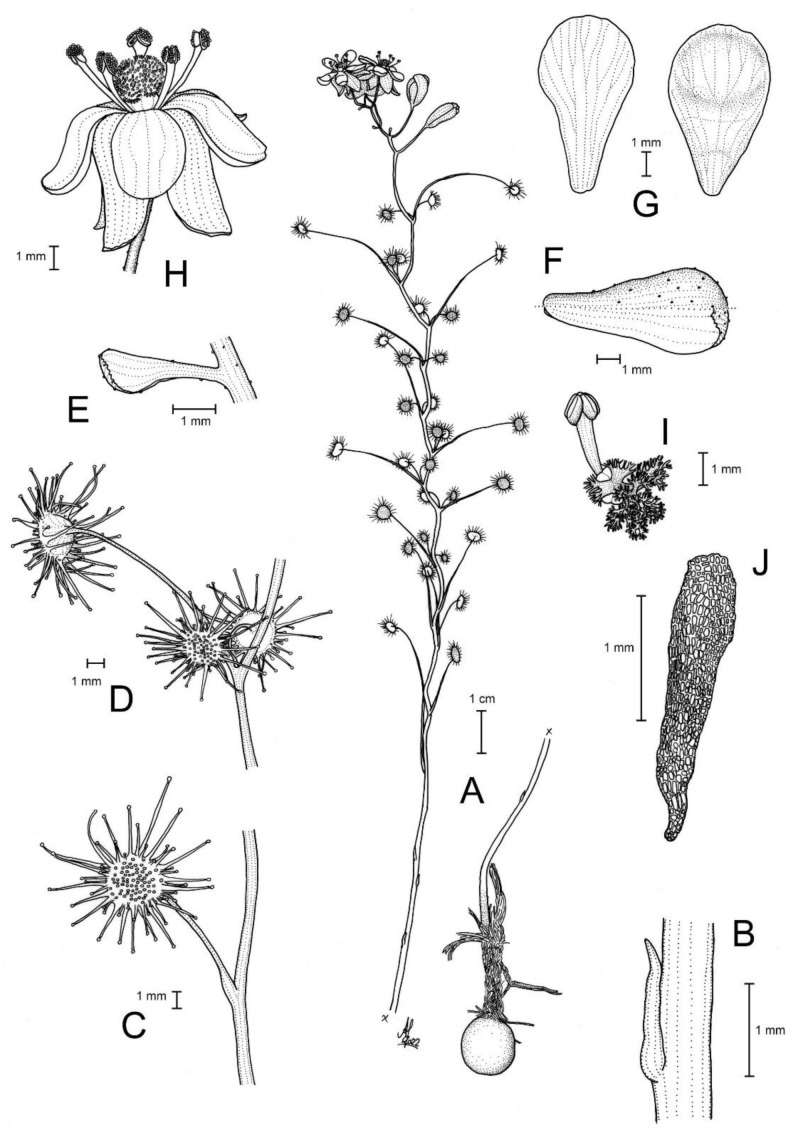
*Drosera rubricalyx* T.Krueger & A.Fleischm. (**A**) habit; (**B**) cataphyll from stem base; (**C**) cauline leaf from lower part of the stem; (**D**) group of leaves from upper node of the stem, consisting of one cauline leaf and two axillary leaves; (**E**) bract; (**F**) sepal, top abaxial view, bottom adaxial view; (**G**) petals, left pressed, right in living condition; (**H**) flower, lateral view; (**I**) gynoecium, top view, with two styles only partially drawn, one stamen from the androecium additionally depicted; and (**J**) seed. (**A**,**B**,**G**-left) from the type (*J. Drummond coll. VI n. 110*), (**C**–**G**-right,**H**–**J**) from photographs taken near Jurien Bay, Western Australia. Drawing: A. Fleischmann.

**Figure 18 biology-12-00141-f018:**
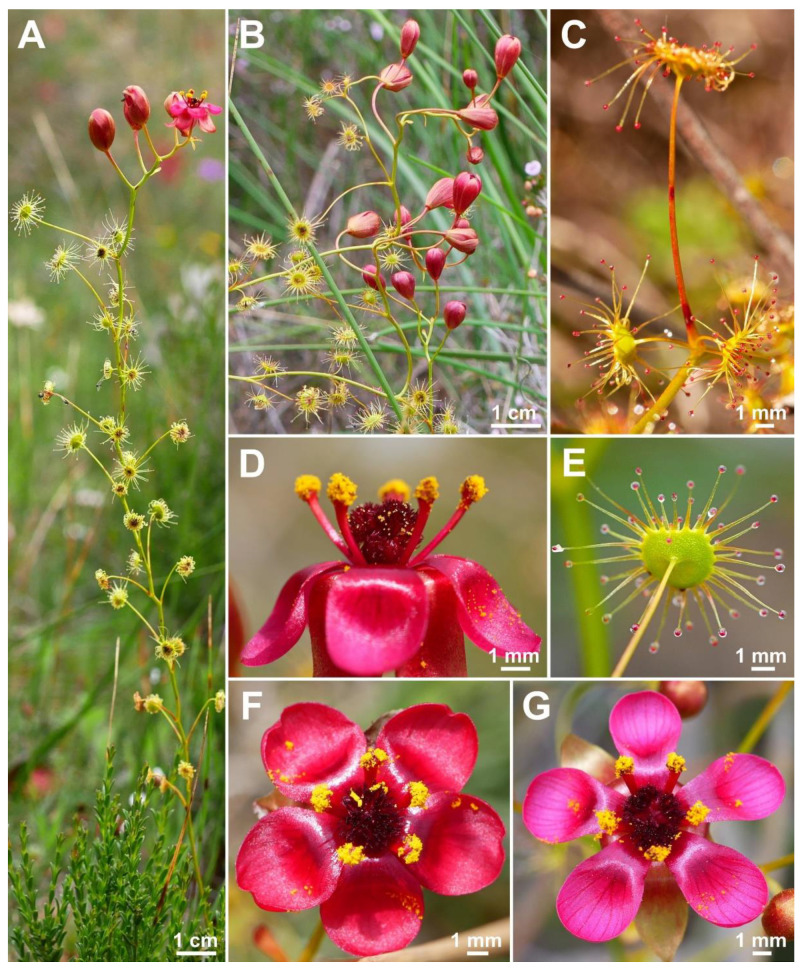
*Drosera rubricalyx* T.Krueger & A.Fleischm. (**A**) habit; (**B**) sepals showing distinctive red colouration; (**C**) leaf in upper part of the stem with a pair of smaller axillary leaves emerging from the axil; (**D**) flower in bright sunlight, lateral view; (**E**) lamina, most leaves are either orbiculate or orbiculate with flattened upper margin (as shown here); (**F**) flower in bright sunlight; and (**G**) flower in diffuse light. Images: T. Krueger from near Jurien Bay, Western Australia, 28 August 2021.

**Diagnosis:***Drosera rubricalyx* is morphologically most similar to *D. hortiorum* T.Krueger & G.Bourke and *D. macropetala* (Diels) T.Krueger & A.Fleischm. from which it differs by (contrasting characters in parentheses): (1) its petal colour, which is deep red in inner half transitioning to deep pink in outer half (petals deep red in inner half transitioning to dark purplish red in outer half in *D. hortiorum* or white with deep red base in *D. macropetala*); (2) its short peduncles, which are 0.8–2.2 cm long (peduncles 1.2–4.2 mm long); and (3) its red sepals, which contrast strongly with the yellowish green stem (sepals yellowish green, yellowish brown, or red, not contrasting strongly with the yellowish green or red stem). It is further distinguished from *D. macropetala* by (contrasting characters in parentheses): (1) its narrower petals, which are 3.3–4.8 mm wide (petals 4.5–7.5 mm wide); (2) its filament shape, which is only slightly dilated towards apex, 0.4–0.6 mm wide near apex (filaments strongly dilated towards apex, 0.5–0.9 mm wide near apex); and (3) its yellowish green tentacle stalk colour (tentacle stalks red in lower half, yellowish green in upper half or red throughout). *Drosera rubricalyx* further shares morphological similarities with *D. calycina* Planch. (of which it was initially described as an infrataxon in 1864 by Bentham), from which it is distinguished by (contrasting characters in parentheses): (1) the presence of 2 smaller axillary leaves in the axils of the upper 2–10 cauline leaves (all cauline leaves solitary); (2) its lamina shape, which is orbiculate or orbiculate with slightly flattened upper margin (lamina reniform or orbiculate with flattened, often truncated upper margin); and (3) its filament shape, which is only slightly dilated towards apex, 0.4–0.6 mm wide near apex (filaments strongly dilated towards apex, 0.5–1.1 mm wide near apex).

**Description:** Tuberous perennial herb, 14–35(–45) cm tall above ground including inflorescence. **Tuber** subglobose, ca. 12–14 mm in diameter, enclosed in black papery sheaths from previous seasons’ growth. **Stem** (subterranean part) 3.5–6.5 cm long, 1.6–4.0 mm in diameter, enclosed in brown, fibrous tunic formed from previous seasons’ stems and roots. **Roots** few, fibrous, emerging laterally from along subterranean part of stem, mostly immediately above tuber. **Stem** (epigeous part) erect, self-supporting, simple, terete, slightly fractiflex, glabrous, (11–)14–30(–41) cm tall, 0.8–1.2 mm in diameter near soil surface, 0.5–1.0 mm in diameter at internodes, yellowish green, red near soil level; sometimes 2–4 stems emerging from the same tuber. **Cataphylls** 5–9 on lower part of stem, subulate, 1.5–3.5 mm long, ca. 0.5 mm wide, red to orangey yellow. **Leaves** solitary in lower part of stem but upper (20–)30–60% of leaves in groups of three per node, due to two much shorter axillary leaves emerging from the axils; internodes 3–20 mm and 12–20 nodes bearing leaves (foliose nodes) present in flowering individuals. **Petioles** terete, semi-erect, straight or slightly arcuated abaxially (downwards), strongly arcuated abaxially near tip, glabrous, 9–25(–29) mm long, 0.5–0.9 mm wide at base, tapering to 0.1–0.3 mm towards the lamina, yellowish green, sometimes blotched with red. **Lamina** peltate, orbiculate or orbiculate with slightly flattened adaxial lateral margin, shallowly concave, adaxial surface facing outwards or slightly downwards, 2.2–3.5 mm long, 2.4–3.9 mm wide; lamina adaxial surface covered with stalked, carnivorous, secretive capitate glands (tentacles); tentacles 2–6 mm long at lamina margin, decreasing in size towards centre of lamina, with greenish yellow stalk (sometimes slightly at base); lamina abaxial surface glabrous. **Petioles of axillary leaves** terete, semi-erect, arcuated downwards along whole length, glabrous, 3–7 mm long, 0.3–0.4 mm wide at base, tapering to 0.1–0.2 mm towards lamina, yellowish green. **Lamina of axillary leaves** of same shape as the lamina described above, (1.7–)1.9–2.7 mm long, (1.8–)2.0–3.0 mm wide. **Inflorescence** a 1–8-flowered scorpioid cyme, terminal, simple, single-sided, 2.0–5.5 cm long. **Peduncle** terete, 0.8–2.2 cm long, 0.5–0.8 mm in diameter, microscopically glandular (appearing glabrous), yellowish green, sometimes blotched with red. **Pedicels** terete, erect in fruit, 5–17 mm long in fruit, 0.4–0.7 mm in diameter, spaced by 2–7 mm along rhachis, microscopically glandular (appearing glabrous), yellowish green, usually transitioning to red in upper half. **Bracts** spathulate, narrowly obovate, or subulate, arcuated adaxially (upwards), often concave, apex entire or irregularly crenulate, 1.9–4.0(–5.0) mm long, 0.6–1.1 mm wide, glabrous. **Sepals** 5, narrowly obovate to narrowly elliptic, arcuated adaxially (upwards), slightly concave, often reflexed during anthesis, apex entire or crenulate, 6–12 mm long, (2.0–)2.5–4.2 mm wide, abaxial surface microscopically glandular, red, sometimes with yellowish brown in upper half, minute black spots sometimes apparent. **Corolla** (10–)11–16(–18) mm in diameter**. Petals** 5, deep red in inner half transitioning to deep pink in outer half, obovate or narrowly obovate, deeply concave and slightly arcuated adaxially (upwards), apex rounded and entire, (5.0–)5.5–8.0 mm long, 3.3–4.8 mm wide. **Stamens** 5, 3.4–5.0 mm long. **Filaments** very slightly dilated towards apex, 0.4–0.5 mm wide at base, 0.4–0.6 mm wide near apex, deep red. **Anthers** bithecate, retrorse, 0.8–1.3 mm wide. **Pollen** yellow. **Ovary** obovoid, 3-carpellate, fused, 1.3–1.9 mm in diameter, deep red. **Styles** 3, divided into a few filiform segments just above the base, style segments again divided into many terete style segments, forming a crowded tuft, not extending laterally beyond filaments, 1.2–1.7 mm long, very dark red. **Stigmas** simple or shortly branched, at tips of style segments, papillose, ca. 0.2 mm long, very dark red. **Seeds** narrowly obtrullate to narrowly obovate, outline narrowly conical with slight ellipsoid swelling in the upper (funicular) half, funicular (upper) end truncate (=pin- or nail-shaped seeds), basal (chalazal) end pointed with short conical to fusiform and often slightly curved appendage, 1.8–2.4 mm long, 0.3–0.5 mm wide, testa black-brown, chalazal and funicular ends pale brown; testa longitudinally reticulate, with anticlines thin and only shallowly raised.

**Etymology:** The specific epithet is derived from the Latin *ruber* (=red) and *calyx* in reference to the red sepal colour of this species that provides a distinct colour contrast to the yellowish green stem and peduncle.

**Taxonomic notes:** The distinctive characters distinguishing *D. rubricalyx* from the morphologically most similar taxa (*D. hortiorum* and *D. macropetala*) are detailed under those respective subheadings. The unique petal colour combination of deep red and deep pink usually quickly allows identification of *D. rubricalyx* in the field (Figure 4). Only *D. calycina* may occasionally produce a similar petal colour, although that species always has solitary cauline leaves, while *D. rubricalyx* consistently produces two smaller axillary leaves in the axils of the upper 2–10 cauline leaves. Additionally, *D. calycina* has a different lamina shape, which is reniform to orbiculate with flattened, often truncated, upper margin vs. lamina orbiculate or orbiculate with slightly flattened upper margin in *D. rubricalyx*.

Bentham [21] distinguished *D. calycina* var. *minor* (=*D. rubricalyx*) from *D. calycina* based on its smaller leaves and flowers. Indeed, the petals of *D. rubricalyx* are usually much smaller, especially in width, compared with those of *D. calycina* and *D. macropetala* (the latter species was included in Bentham’s description of *D. calycina* as he cites *J. Drummond coll. VI n. 109*, the type of *D. macropetala*). Additionally, Bentham [21] already noted that *D. rubricalyx* has “rather less dilated” filaments when compared to these two species, which is indeed a reliable distinguishing floral feature. However, it is notable that Bentham was able to detect this feature in the dried herbarium specimens he studied, as the filaments in dried specimens of *D. microphylla* complex species often considerably shrink in width (T. Krueger pers. obs.).

The infrataxon epithet of *D. calycina* var. *minor*, published by Bentham in 1864 [21], cannot be elevated to species rank as an older homonym with nomenclatural priority exists (*Drosera minor* Schumach. & Thonn. in Schumach., published in 1827 and generally treated as a synonym of *D. indica* L.). Therefore, a new name, *D. rubricalyx*, had to be coined for *D. calycina* var. *minor* at the rank of species.

*Drosera rubricalyx* was possibly illustrated by Erickson [22] (p. 40, illustration 1, as “*D. microphylla*”), who had examined specimens of this species from Mt. Lesueur (*C.A. Gardner 9350*).

**Distribution and habitat:** *Drosera rubricalyx* is known only from Lesueur National Park and a population in the same general vicinity (Figure 3). Both sites are located near the coastal town of Jurien Bay, ca. 200 km north of Perth. It occurs in low heath in poorly drained, seasonally moist flats, depressions, and hillslopes with *Calothamnus quadrifidus* R.Br. (Myrtaceae) and *Drosera gigantea* Lindl.

**Phenology:** Flowering has been recorded in August and early September.

**Conservation status:** Listed as Priority Two (poorly known species) under Conservation Codes for Western Australian Flora and Fauna (Western Australian Herbarium 1998–; https://florabase.dpaw.wa.gov.au/ (accessed on 6 December 2022)), under the phrase-name “*Drosera* sp. Lesueur National Park (C.A. Gardner 9350)”. Vulnerable (VU) under the IUCN Red List criteria D1+2 following IUCN [29]. *Drosera rubricalyx* has only recently been observed from a single roadside population in a reserve managed by the Western Australian Department of Biodiversity, Conservation and Attractions (DBCA) near Jurien Bay where it was photographed and shared online by local wildflower photographer Daniel Anderson. The images of these plants were posted on Facebook where the authors identified them as a possible new species. A survey of this population was subsequently conducted on 28 August 2021 and ca. 250 plants were counted growing scattered in an area of ca. 3000 m² (T. Krueger pers. obs.). During this survey (and also during two subsequent surveys in 2022), grasshoppers were observed eating large numbers of flower buds. It is not known whether these grasshoppers are native or whether they pose a significant long-term threat.

The other populations historically reported from Lesueur National Park (*C.A. Gardner 9350* and *E.A. Griffin 4207*) could not be re-located during recent surveys and it is not known if any other populations exist. *Drosera rubricalyx* is potentially vulnerable to unlicenced collection. Further surveys are recommended to gain a better understanding of this taxon’s distribution, number and size of populations, and to identify additional potential threats.

**Notes on the lectotypification:** Very little information is provided by Bentham’s description of *D. calycina* var. *minor* and none of the known syntypes of *J. Drummond coll. VI n. 110* appear to be annotated by him. However, since he was based at Kew, selecting K000215038 as the lectotype is a reasonable course of action. This specimen was already mentioned as the “holotype” for *D. calycina* var. *minor* by Lowrie [1], but this does not represent a valid lectotypification as it lacked the phrase “designated here” in accordance with ICN Art. 7.11. [26]. Further support for the choice of the K material as the lectotype comes from Moore [46] (pp. 29–30), who explains that Bentham only consulted the Kew collections for his Flora Australiensis [21] but not any other herbaria and specifically not the (often more accurately labelled) duplicates of the Drummond collections at BM [46].

**Notes on Drummond’s type collection:** The syntypes of *J. Drummond coll. VI n. 110* only provide the rough locality information of “Western Australia, between Moore River and Murchison Rivers”. However, more precise information on the locus classicus comes from the collector Drummond himself [18,19]. There, he describes a species that he refers to as “Another new *Drosera*, with bright scarlet flowers and glabrous sepals larger than the petals”, a description which matches *D. rubricalyx*. Drummond [18,19] mentions that this species “[...] is found near the Yandyait Spring, to the east of the Hill river”. This locality could not be precisely pinpointed, but it is likely within 15–20 km of the known population near Jurien Bay. As *J. Drummond coll. VI n. 110* is the only collection of *D. rubricalyx* by Drummond, it is safe to conclude that this is the collection locality. Additional support for this comes from Barker [41], who evidenced that Drummond [18] is referring to Drummond’s VI collection series, i.e., the series comprising the type collection of Bentham’s *D. calycina* var. *minor* and hence also *D. rubricalyx*. This means that the year of collection (not provided on any of the syntype specimens) is 1850 or 1851, as for all specimens comprising the VI collection [30].

While the Drummond specimen at MPU (MPU1254140 photo!) has been erroneously annotated as the “holotype” of *D. calycina* by Marchant 1985 (in sched.) (which did not constitute a valid lectotypification, see Notes on the lectotypification under that species), the plants clearly exhibit axillary leaves in the upper parts and a much more orbiculate lamina shape than that typically found in *D. calycina*. Together with the apparently dark red or purplish red petal colour and the overall habit, this indicates that MPU1254140 is most likely *D. rubricalyx* and thus another syntype of *J. Drummond coll. VI n. 110*, although the MPU specimen lacks a collector’s number (therefore, it is not included as a syntype here).

It should be noted that Drummond did not number his collections sequentially [33,46], i.e., the type collection of *D. rubricalyx* (*J. Drummond coll. VI n. 110*) was likely not made immediately after that of *D. macropetala* (*J. Drummond coll. VI n. 109*). Generally, it is often difficult to georeference and trace back Drummond’s collections, as complained about by Diels [33] (pp. 49–50, literally translated), who pointed out that Drummond’s “enormous collections are not labelled. Their numbering is unreliable, and the individual sets do not always correspond to each other regarding their numerics”. Diels [33] (p. 50) further wrote: “During sorting and distributing of the exsiccates, various mistakes and confusions arose […]”, which Diels blames on the long transport times and difficult communication between “Western Australia and the outside world”. This might explain why some of the collections bear false collection numbers and/or comprise mixed collections from two different gatherings made by Drummond (e.g., LD1971779, labelled “110”, consists of two individuals of *J. Drummond coll. VI n. 110*, *D. rubricalyx*, and four individuals of *J. Drummond coll. VI n. 109*, *D. macropetala*)—however, these mistakes could also have been made at the respective herbaria later during mounting or (re)labelling of the specimens (which historically was often done inaccurately, e.g., at K, according to Moore [46]).

**Additional specimens examined (paratypes):** AUSTRALIA. Western Australia: Mount Lesueur, 20 August 1949, *C.A. Gardner 9350* (PERTH 00666955!; PERTH 00661805!; PERTH 00661791!); Proposed Lesueur National Park [precise locality withheld for conservation purposes], upper slope, poorly drained, sandy clay, 5 September 1985, *E.A. Griffin 4207* (PERTH 01613472!).

**Additional localities examined:** Jurien Bay [precise locality withheld for conservation purposes], poorly drained, seasonally moist flat, 28 August 2021, T. Krueger pers. obs.

### 3.10. Identification Key to the Species of the Drosera microphylla Complex (See Table 2 for Multiple Access to the Morphological Characters)

Axillary leaves absent in adult flowering plants, all cauline leaves solitary...........................................................................................................................................**2**
-Axillary leaves present at least on uppermost 1–9 nodes (cauline leaves in groups of 3 [2,3,4,5] per node with one larger main leaf and usually two smaller axillary leaves) (Note: white-flowered plants from Cape Arid with axillary leaves belong to unusual *D. esperensis*, see under that species which normally has solitary cauline leaves)..........................................................................................**6**Lamina reniform or orbiculate with strongly flattened, often truncated upper margin; petioles straight or only slightly arched; corolla diameter 14–20 mm; petals deep red in inner half transitioning to purplish red in outer half; stamens 4.0–6.5 mm long; plants occurring in Jarrah forest in the western Darling Range.....................................................................................................................***D. calycina***-Lamina orbiculate or orbiculate with slightly flattened (but not truncated) upper margin; petioles arched along whole length or arching gradually increasing towards tip; corolla diameter 8–15 mm; petals reddish orange with deep red base, white with pale purplish red base, dark red throughout, or purplish pink with deep red base; stamens 2.5–4.4 mm long; plants occurring in moss or low heath on granite outcrops, swamps, Mallee sandplains, or mountain slopes near the WA south coast or around the Stirling Range...........................................**3**
Petals white with pale purplish red base; plants frequently forming dense, mat-like colonies; stem, petiole, lamina and inflorescence red; filaments ± linear (0.3–0.6 mm wide throughout); filaments and styles white with red or purplish red base; stigmas white; plants occurring on coastal granite outcrops east of Esperance.....................................................................................................................***D. esperensis***-Petals reddish orange with deep red base, dark red throughout, or purplish pink with deep red base; plants not colony-forming or only forming relatively sparse (not mat-like) colonies; stem, petiole, lamina and inflorescence yellowish green or red; filaments dilated towards apex (from 0.2–0.3 mm near base to 0.3–0.9 mm near apex); filaments and styles deep red or purplish red; stigmas reddish purple, deep red, or dark red; plants occurring on granite outcrops, swamps, Mallee sandplains, or mountain slopes along the south coast between Walpole and Cheynes Beach or around the Stirling Range....................................**4**Petals reddish orange with deep red bases; leaves frequently detach before anthesis, leaving large leaf-free gaps along stem (except in populations from the Stirling Range and Mt. Lindesay); stigmas reddish purple; flowering from August to October....................................................................................................................***D. microphylla***-Petals dark red throughout or purplish pink with deep red base; leaves do not detach before or after anthesis; stigmas deep red or dark red; flowering from June to early September...............................................................................................**5**Petals dark red; plants not colony-forming, occurring in very sparse populations; stem slightly fractiflex; sepals yellowish brown to greenish yellow; petal shape narrowly obovate to broadly spathulate; plants occurring in sandy soils in open Mallee woodlands in and around the Stirling Range.......................................***D. koikyennuruff***-Petals purplish pink with deep red base; plants colony-forming via adventitious stolons, occurring in dense populations; stem usually strongly fractiflex; sepals red to purplish red; petal shape obovate to very broadly obovate; plants occurring in shallow moss on granite outcrops between Walpole and Denmark.................................................................................................................***D. reflexa***Axillary leaves present on all nodes, often even on upper cataphylls, all leaves in groups of 2–5 per node; inflorescence with 5–23 flowers; corolla diameter 6–9 mm; petals very dark red to blackish red; styles branched mostly above base, style segments then only sparsely branched; style segments extending laterally beyond the filaments; seeds falcate to allantoid and 2.3–2.6 mm long................................***D. atrata***-Axillary leaves present only on uppermost 1–9 nodes, never on cataphylls, lower 4–15 leaves solitary; inflorescence with 1–8 flowers; corolla diameter 8–22 mm; petals deep red, dark purplish red, white, or deep pink; styles branched above base and style segments themselves then further branched, forming a crowded tuft; style segments not laterally extending beyond filaments; seeds very narrowly obconic, narrowly clavate to acerose, 1.7–2.4 mm long..................................................................................................................................**7**Petals deep red in inner half transitioning to dark purplish red in outer half; corolla diameter 8–11 mm; plants occurring east and south-east of Perth...........***D. hortiorum***-Petals white with deep red base or deep red in inner half transitioning to deep pink in outer half; corolla diameter 11–22 mm; plants occurring on and just west of the Dandaragan Plateau north of Perth.......................................................**8**Petals white with deep red base; plants yellowish green with yellowish brown sepals or red with reddish brown sepals; tentacle stalk colour red or red in lower half with greenish yellow upper half; peduncle length 1.3–4.2 cm; petal width 4.5–7.5 mm; filaments strongly dilated towards apex (from 0.3–0.5 mm to 0.5–0.9 mm); plants occur in laterite gravel near tops of hills on the Dandaragan Plateau, between Mogumber and Dandaragan......................................................................***D. macropetala***-Petals deep red in inner half transitioning to deep pink in outer half; plants yellowish green with contrasting red sepals; tentacle stalk colour greenish yellow (only rarely with reddish base); peduncle length 0.8–2.2 cm; petal width 3.3–4.8 mm; filaments only slightly dilated towards apex (from 0.4–0.5 mm to 0.4–0.6 mm), plants occur in sandy clay in seasonally moist flats, depressions, and hillslopes between Badgingarra and Lesueur National Park............***D. rubricalyx***

**Table 2 biology-12-00141-t002:** Morphological comparison of all nine species of the Drosera microphylla complex. Abbreviations: *D. calyc.*=*D. calycina*; *D. esper.*=*D. esperensis*; *D. hort.*=*D. hortiorum*; *D. koiky.*=*D. koikyennuruff*; *D. macro.*=*D. macropetala*; *D. micro.*=*D. microphylla*; and *D. rubri.*=*D. rubricalyx*.

Character	*D. atrata*	*D. calyc.*	*D. esper.*	*D. hort.*	*D. koiky.*	*D. macro.*	*D. micro.*	*D. reflexa*	*D. rubri.*
Colony forming, clonal growth	no	no	yes	no	no	no	yes (sometimes no)	yes	no
Stem shape	± straight	slightly to strongly fractiflex	strongly fractiflex	slightly fractiflex	slightly fractiflex	slightly fractiflex	slightly to strongly fractiflex	strongly fractiflex	slightly fractiflex
Axillary leaves	present (all nodes)	absent	absent (except at Cape Arid)	present (upper 1–7 nodes)	absent	present (upper 1–9 nodes)	absent	absent	present (upper 2–10 nodes)
Petiole shape	arching increasing towards tip	±straight	arched along whole length	±straight or slightly arched	arched along whole length	slightly arched along whole length	arched along whole length	arched along whole length	±straight or slightly arched
Lamina shape	orbiculate with flattened margin	reniform/orbiculate with flattened margin	orbiculate	orbiculate/ orbiculate with slightly flattened margin	orbiculate	orbiculate/ orbiculate with slightly flattened margin	orbiculate	orbiculate	orbiculate/ orbiculate with slightly flattened margin
Flowers per inflorescence	5–23	1–9	1–5	2–6	1–2	2–8	1–7	1–3	1–8
Sepal colour	yellowish brown	yellowish brown	red	yellowish brown	yellowish brown	yellowish brown to red	red to yellowish brown	red to purplish red	red
Corolla diameter (mm)	6–9	14–20	10–14	8–11	8–12	13–22	8–15	9–15	11–16
Petal colour (simplified)	blackish red	deep red	white	dark purplish red	dark red	white + deep red	reddish orange	purplish pink + deep red	deep pink + deep red
Filament shape	not dilated	dilated	not dilated	very slightly dilated	dilated	dilated	dilated	dilated	very slightly dilated
Stamen length (mm)	2.0–3.0	4.0–6.5	2.8–3.4	3.0–3.5	2.9–3.5	3.3–4.9	2.5–3.5	3.8–4.4	3.4–5.0
Relative style length	extending laterally beyond filaments	not extending laterally beyond filaments	extending laterally ± to filaments	not extending laterally beyond filaments	extending laterally beyond filaments	not extending laterally beyond filaments	extending laterally beyond filaments	extending laterally just beyond filaments	not extending laterally beyond filaments
Stigma colour	dark red	dark red	white	deep red	deep red	blackish red	reddish purple	dark red	blackish red
Habitat	lateritic hills, kwongan heath	lateritic hills, Jarrah forest	granite hills, mossy seepages	granite outcrops, moist sandplains	sandplains, Mallee scrub	lateritic hills, kwongan heath	granite outcrops, swamps, mountain slopes	granite outcrops, moss	moist flats or hillslopes, kwongan heath
Flowering time	May to August	August to September	August to October	June to September	June to July	August	August to October	June to September	August to September

## 4. Discussion

### 4.1. Endemism and Species Conservation in the Drosera microphylla Complex

The species of the *Drosera microphylla* complex are among the most narrowly endemic and most threatened species of *D.* section *Ergaleium* (tuberous sundews). Six of the nine species of the complex have known distribution areas with a maximum diameter of less than 100 km (*D. atrata*, *D. calycina*, *D. koikyennuruff*, *D. macropetala*, *D. reflexa*, and *D. rubricalyx*; Figure 3). By contrast, only 3 of the 68 remaining species of *D.* section *Ergaleium* occur across such small areas of distribution (these are *D. graniticola* N.G.Marchant, *D. orbiculata* N.G.Marchant & Lowrie, and *D. prostratoscaposa* Lowrie & Carlquist; [1]; https://florabase.dpaw.wa.gov.au/ (accessed on 22 December 2022)). Additionally, most members of the *D. microphylla* complex are uncommon even within their distribution areas, being highly localised and often present in very small populations of fewer than 100 individuals (or even fewer than ten individual plants, as is often the case for *D. atrata*). These very small distribution areas and population sizes, in combination with the threats of habitat loss (including presumed reductions in gene flow as a result of habitat fragmentation) and illegal collection—identified as threats for seven of the nine species—indicate a strong necessity for targeted conservation efforts to ensure their long-term survival in nature.

*Drosera macropetala* and *D. reflexa* are here assessed as the most threatened members of the *D. microphylla* complex, with a recommended Western Australian Conservation Code status of Priority One and Priority Two, respectively (Western Australian Herbarium 1998–; https://florabase.dpaw.wa.gov.au/ (accessed on 22 December 2022)); and an IUCN category of Endangered (EN) and Critically Endangered (CR), respectively [29]. Despite considerable survey efforts, both species are currently each only known from a single roadside location (with *D. macropetala* being historically also recorded from a second location further south; Figure 3). Such roadside habitats are particularly vulnerable to threats from road maintenance and construction, altered hydrology, and weed infestation [16].

*Drosera atrata* and *D. rubricalyx* are both assessed as Vulnerable (VU) [29] given the available data, with a Western Australian Conservation Code status of Priority Three and Priority Two, respectively. While both are known from multiple locations, their distribution areas are very small (ca. 20–50 km; Figure 3).

*Drosera hortiorum* is unusual within the *D. microphylla* complex as it has a relatively large distribution area spanning at least 160 km but is only known from four locations (Figure 3), each with a population size of fewer than 50 individuals. It is recommended for a Western Australian Conservation Code status of Priority Two, but insufficient survey efforts for this species means it cannot yet be assessed under IUCN criteria (=Data Deficient, DD) [29]. Similarly, *D. koikyennuruff* (which is only known from two locations near the Stirling Range; Figure 3) could not be assessed under IUCN criteria given the lack of available survey data, although it is assessed as Priority Two under the Western Australian Conservation Code.

The only species of the *D. microphylla* complex that are not currently assessed as potentially threatened are *D. calycina*, *D. esperensis*, and *D. microphylla*. All three species are known from numerous sites and are generally quite common within their preferred habitat (i.e., they tend to form relatively large populations). In addition, *D. esperensis* and *D. microphylla* have relatively large distribution areas extending over >100 km (Figure 3).

Knowledge of the distributions and range extensions of the species of the *D. microphylla* complex for this study were not only gained through field studies and herbarium research, but also from photographs from citizen science and social media (see Section 4.3). This highlights the importance of citizen scientist contributions for nature conservation (see Section 4.3 and [47,48,49]).

### 4.2. Flower Biology of the Drosera microphylla Complex

The members of the *D. microphylla* complex rank among the comparatively few sundew species that have non-ephemeral flowers, i.e., flowers that last for longer than one day. Within that group, they represent a smaller group still of species whose flowers close every night until they finally fade after about 3–5 days (A. Fleischmann pers. obs.). The daily opening and closing of the flowers is achieved by one-sided petal growth (common in many other plants with flowers that cyclically open and close), as evidenced by the fact that the petals of all members of the *D. microphylla* complex increase in size during anthesis as they slightly enlarge with each new opening event. The non-ephemeral nature of the flowers of species from this affinity, as well as their nocturnal closure, was first reported by James Drummond [18,19]. The initial opening of the flower from bud takes some time, as both sepals and petals must spread open on the first occasion. Once a flower has fully opened for the first time, the concave petals close during the late afternoon (see Figure 1I), the sepals also subsequently close at night or during unfavourable weather such as on cold and/or rainy days, covering the reproductive organs. The re-opening of an individual flower is light-dependant, but also strongly temperature-dependant, and covering individual, closed flowers with plastic bags on cold sunny days will often induce them to open within just a few minutes (F. Hort, J. Hort, and T. Krueger pers. obs. for *D. hortiorum*). The flowers of members of the *D. microphylla* complex are non-fragrant (A. Fleischmann pers. obs.; Gibson [14] for *D. esperensis*), which is an exception among members of *D.* section *Ergaleium*, which usually have strongly fragrant flowers [50] (A. Fleischmann pers. obs.). The combination of non-fragrant, non-ephemeral flowers that close each night is unique among Australian *Drosera* and restricted to the *D. microphylla* complex—it represents an ecological apomorphy of that complex. The strongly concave petals, which are shorter than the large sepals, are likewise an apomorphy for this complex. Within the genus, both characters are only paralleled in the very distantly related neotropical *Drosera biflora* Willd. from *D.* section *Drosera* [51].

It is interesting to note that the three species with exceptionally dark flower colours (*D. atrata*, *D. hortiorum*, and *D. koikyennuruff*) have unusually early flowering times, much earlier than most other species of the complex (only *D. reflexa* has also been observed to flower as early as June, G. Bourke pers. obs.) and also much earlier than most other erect tuberous *Drosera* species. This might be an adaptation to certain pollinators that are active during this period, both as temporal reproductive isolation from sympatric taxa and possibly a favoured or easy-to-spot colour among pollinators active at this time of year.

### 4.3. The Role of Citizen Science and Social Media in Taxonomy and Species Discovery—An Example from Drosera

The present work serves as another excellent example of citizen science, social media, and online photo platforms facilitating or even driving improvements in taxonomic and ecological knowledge. These networks have in several cases alerted botanists to new discoveries, resulted in the relocation of ‘lost’ or poorly known taxa, or simply extended the ranges of known species beyond those distributions established through herbarium and museum records alone [48,49,52,53,54,55]. *Drosera macropetala*, *D. rubricalyx*, *D. hortiorum*, and *D. koikyennuruff* were first (re-)discovered via online photographs or significant contributions to their range and distribution gained from images shared on the iNaturalist website and app and other social networking platforms. Carnivorous plants such as *Drosera* are usually well-represented on naturalist photographic databases and platforms as these peculiar plants are frequently photographed [56]. For example, iNaturalist (Research-Grade observations only) hosts georeferenced photographs of 236 (89.7%) out of the ca. 260 *Drosera* species known to science to date, representing 62,344 individual observations (most of which consist of several photographs) made by 17,477 observers (https://www.inaturalist.org/observations?place_id=any&taxon_id=51935 (accessed 21 December 2022)). As of December 2022, the Global Biodiversity Information Facility (GBIF; www.gbif.org (accessed on 14 January 2023)) provides 453,239 occurrence records of 263 species of *Drosera* (=100% taxon coverage); 79.4% of these records are based on citizen science observations (“human observation”), as opposed to 18.5% that come from georeferenced, databased herbarium specimens (“preserved specimen”) (GBIF.org (accessed on 9 January 2023) GBIF Occurrence Download https://doi.org/10.15468/dl.h4fwxq). This coverage is not restricted only to widespread or common species (the most commonly observed carnivorous plant on GBIF is *Drosera rotundifolia* L. [56], which is as of 9 January 2023 represented in that database by 200,188 occurrences, 89.8% of which are citizen science observations (GBIF.org (accessed on 9 January 2023) GBIF Occurrence Download https://doi.org/10.15468/dl.9w2ded). For some formerly rarely encountered *Drosera* species, more “photo vouchers” exist as geographic records than are available as herbarium specimens. For example, the South African *Drosera xerophila* was known only from three herbarium specimens and seven records on iNaturalist at the time of its description (all of them cited in Fleischmann [44]); the number of herbarium specimens has not changed since the publication of the species in April 2018, but the number of observations for that species on iNaturalist had risen to 307 by January 2023, made by 131 different observers (iNaturalist community. Observations of *Drosera xerophila*. Exported from https://www.inaturalist.org (accessed on 9 January 2023)).

This illustrates very well the scientific value of these image repositories for taxonomic and biogeographic work and for nature conservation [47,54,57]. In other organismic groups that are frequently photographed and uploaded to citizen science social networks, such as iNaturalist, by enthusiasts, these records represent by far the largest contribution to our knowledge of the distribution of these taxa. This is particularly true for “charismatic animals”, such as mammals and birds, for which 70% (mammals) and 87% (birds) of the total records on GBIF in 2016 comprised online citizen science records [48]. As the global citizen science naturalist networks continuously and quickly increase their amount of data (e.g., in 2019 alone, the total number of observations on iNaturalist doubled from 25 million to 50 million [55]), the rich dataset of occurrence records provided by citizen science by far outweighs those gained annually from herbarium specimens and literature revisions, though of course the latter usually provide persistent, high-quality taxonomic data along with physical reference specimens that are additionally suitable for DNA extraction and genetic analyses, microscopic examinations, and digitised associated metadata.

Regarding species discoveries and range extensions through social media and citizen science, the present taxonomic revision provides an excellent example. Four out of the six species newly described or newly classified as species here were initially (re-)discovered on social media, with only *D. atrata* and *D. reflexa* (co-)discovered in situ by the authors of the present work. Citizen science and social media networks have also provided the first known photographs of *D. koikyennuruff*, *D. macropetala*, and *D. rubricalyx*, which were previously only known from decades-old herbarium collections. Another relatively well-publicised example of a *Drosera* species discovered on social media is *D. magnifica*, which was first spotted and identified as a species new to science from photographs posted on Facebook in 2014 [27]. An example for significant range extensions in *Drosera* provided by citizen science is *D. biflora*, the first record of which from Colombia was made by photographs posted on iNaturalist; the rediscovery of that species in Brazil was also facilitated by photographs posted online (republished in Gonella et al. [51]). These, at the same time, also represented the rediscovery and first known photographs of this species, which was previously only known from 200-year-old herbarium collections made in Venezuela. The citizen scientist photographs of living *D. biflora* specimens also provided additional unique biological and morphological details for this species (curiously, including the unique character of patent to reflexed sepals, a character this species shares with the unrelated species from the *D. microphylla* complex treated here), which could not be discerned from the historic herbarium material and helped to increase the knowledge about the taxonomy and relationships of this species [51].

Even organismic interactions can be revealed or documented for the first time through social media photographs and citizen science [58], such as plant-pollinator relationships. There are numerous examples where floral visitors and pollinators of *Drosera* species have been first documented by citizen scientists via photographs shared online. Carnivorous plant-prey relationships can also be documented by citizen science, because the different interacting organisms can often be identified from photographs (*Drosera* prey, for example, can often be identified from photographs [59]). These platforms effectively connect taxonomic experts from different fields, such as entomologists and botanists. An example involving *Drosera* is an iNaturalist photograph that was used as a voucher in a citizen science approach for mosquito species monitoring in Australia (published as Figure 4B in Braz Sousa et al. [60]). It shows the mosquito *Aedes camptorhynchus* captured on a leaf of *Drosera planchonii* Hook.f. ex Planch. in South Australia (based on an iNaturalist photograph by observer “frank_prinz”: https://www.inaturalist.org/observations/54139335 (accessed on 21 December 2022)). While the *Drosera* expert or trained botanist would have been able to correctly identify the targeted plant taxon in the field and the mosquito expert will name the insect, the advantage of the social media platforms is that both experts, and additionally the observer, are linked via a photograph showing different target taxa of interest. Many of the citizen scientist photographs do not just represent single-species observations but in fact are (often unnoticed) documents of species interactions [58].

An increasing number of applications are being developed to use citizen science and social media data for biodiversity exploration and flora monitoring, linking them with taxonomy (e.g., [56,61]). Machine learning approaches are frequently improved to automate species recognition and thus enhance data mining for images suitable for taxonomy (e.g., [62,63]).

Potential negative effects of providing locality data, especially of rare flora, online in social networks arise in plant groups that are of horticultural interest, such as cacti, orchids, or carnivorous plants [64,65]. For some carnivorous plants, pitcher plants in particular, populations in the wild are mainly threatened by overcollection and poaching for the illegal trade to meet the horticultural demand [16]. However, *Drosera* species, in particular tuberous *Drosera* from Western Australia, have also been and continue to be heavily poached (including from protected areas such as nature reserves) to be sold illegally on social media or internet marketplaces to carnivorous plant enthusiasts and growers worldwide [16,65]. This is a major potential threat, particularly to rare, micro-endemic taxa such as the majority of species from the *D. microphylla* complex treated in this paper. iNaturalist automatically obscures the geographic information of observations of threatened taxa, which greatly helps with mitigating this threat (https://www.inaturalist.org/pages/help#geoprivacy (accessed on 23 December 2022)). For this reason, exact localities have been withheld for conservation purposes for the species of the *D. microphylla* complex recommended to be listed as Priority under the Western Australian Conservation Codes and the authors of the present work generally do not share locality information for threatened flora.

## 5. Conclusions

This study highlights the importance of both citizen science and careful herbarium examination for taxonomic research and conservation efforts. Of the six species newly recognised here, four were (re-)discovered on social media and all but *D. reflexa* had already been represented in herbaria for many decades. Crucially, these six species are rare, narrowly endemic, and potentially threatened, thus the accurate taxonomic classification provided here is expected to contribute to their conservation.

## Figures and Tables

**Table 1 biology-12-00141-t001:** Taxonomic history of the *Drosera microphylla* complex. N/A=taxon not yet described by the date of the publication.

Taxon	Endlicher 1837 [20]	Planchon 1848 [17]	Bentham 1864 [21]	Diels 1906 [8]	Marchant et al. 1982 [23]	Lowrie 2014 [1]	Present Work
*D. microphylla*	Described	Accepted	Under synonymy of *D. filicaulis* ^1^	Accepted	Accepted	Accepted	Accepted
*D. calycina*	N/A	Described	Accepted	Under synonymy of *D. microphylla*	Under synonymy of *D. microphylla*	Accepted	Accepted
*D. calycina* var. *minor*	N/A	N/A	Described	Under synonymy of *D. microphylla*	Under synonymy of *D. microphylla*	Under synonymy of *D. calycina*	Accepted at species rank ^2^
*D. microphylla* var. *macropetala*	N/A	N/A	N/A	Described	Under synonymy of *D. microphylla*	Under synonymy of *D. calycina*	Accepted at species rank
*D. esperensis*	N/A	N/A	N/A	N/A	N/A	Described	Accepted

^1^ This taxon is today considered conspecific with *D. menziesii*. ^2^ Under the replacement name *D. rubricalyx*.

## Data Availability

Not applicable.

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
