# Peer review of "Small Leaves, Big Diversity: Citizen Science and Taxonomic Revision Triples Species Number in the Carnivorous Drosera microphylla Complex (D. Section Ergaleium, Droseraceae)"

_biology, 2023, doi:10.3390/biology12010141_

Round 1

Reviewer 1 Report

The authors presented a very detailed observation and taxonomic treatment on the Drosera microphylla complex. It is exciting the hidden species diversity of these tiny plants was uncovered in this study. It is a little pity that the species delimitations were not accompanied with the molecular evidence, or the statistical evidence using multiple morphological characters. It is difficult to judge whether or how much the new treatments proposed in this study are scientifically reasonable. I hope the authors test their morphological classification using new evidence in the future.

Besides, I do not agree with the authors' view on the lectotypification of D. calycina and D. macropetala.

For D. calycina, there is no direct or solid evidence showing one of the two individuals was from LINN. There are too many possibilities that the LINN specimen was not found from LINN/BM/K. Missing, destroyed, one day appeared again…

For D. macropetala, from the note given by the author, the B specimen is obviously the holotype.

Some other suggestions and minor revisions needed:

In the "Materials and Methods" portion, the authors should clearly declare which species concept they are adopting in this study.

Line 158-161: SEM observations on which organs? Also I did not see any SEM photos from the manuscript. Please add some SEM photos.

Line 370 & Line 598 & 1182, 1185: The authors and original publications should be cited for these names.

Line 974: This name not validly published should not be cited as a synonym. The author can mention this name in the Note portion.

Line 1183: "Type" should be definitely given, holotype?

Line 1508: Not "sp. nov.", but "nom. nov."

Line 1509-1515: This portion is nearly identical to line 1517-1522. It should be deleted.

Line 1612-1616: The sentence explaining the new name should be moved into the next portion.

Table 1. In the first line, the abbreviated specific name should be given with full names in the table legend.

Figure 8: It is more like a light sketch, not a scientific drawing. It should be improved or deleted.

Reviewer 2 Report

The manuscript titled “Small leaves, big diversity: citizen science and taxonomic revision triples species number in the carnivorous Drosera microphylla complex” (No.2150172) has been reviewed. The paper addressed carnivorous Drosera microphylla complex with 9 species from southwest Western Australia. Overall, the description is clear, the pictures and drawing are of good quality, the results are appropriate be important to conservation interest which in the Regional Red Lists.  However, some revisions are necessary.

1. In present study, besides morphological characters of D. calycina var. minor and D. microphylla var. macropetala, are DNA data or chloroplast genome data should be added to elevate their species rank within the genus, together with morphological features?

2. How many fresh flowers (or plants) and specimens investigated respectivly in the study? How can morphological characteristic differences of individual belonged to different habitat be excluded?

For figs: the serial number in the figure is in capital letter, while it is in lowercase letters in legends, keeping the same style.

3. And scale bars for fig.4-5.

4. Figs 3-5 can be moved to the front of the figs.1-2, to introduce the sampling sites and general characteristics of the 9 species of the study.

5. Line 206. The picture title is “Comparison of the seed of all nine species of the Drosera microphylla complex”, but only 8 species are listed? In addition to seed size, color and shape, what ornamentation present on surface (micromorphology) under SEM? Are they the same or different within the genus?

6. Line370, 1508, add previous name remarks (e.g. D. microphylla var. macropetala and D. calycina var. minor).

Reviewer 3 Report

The work presented is a very well prepared taxonomic study. With high standards and an impressive amount of data. The work was a pleasure to read. There are a few elements that are worth improving in the work.

The introduction provides an interesting explanation for the reader in the complex hisotrical taxonomy. It is quite easy to get lost in the changes that took place, so I suggest the authors prepare a table as a summary, with the authors and taxa, and an indication of which author accepted which of them. Including the most recent work, of course.

Line 98-103: I do not fully agree with this interpretation. The authors forgot about ICN article 9.10, which says that the indication of one gathering among, i.e. syntypes by a later author, while using words referring to the type, should be treated as an error that should be corrected as a lectotype. The rule applies until January 1, 2001, so it also applies to Marchant et al. 1982. Unfortunately, I do not have access to Flora Australia to confirm this possibility. 

This should be taken into account and checked to see if it does not apply in any of the lectotype selections.

Line 151: IH should be cited in references, has specific way of citing on website.

Figure 3: The map is not very readable. The problem is mainly with the markers, which are not very clear and are lost in the background. The biggest problem is in the numbers indicating the locations of potential new taxa and in the marker for D.koikyennuruff. I suggest avoiding bright colors.

Figure 5: The authors mentioned that the photos are of nine species, but eight can be found. Although this is standard, however, I also suggest adding "1 mm grid paper" in the description for clarity.

In the taxonomic section, all species should be cited in full, that is, with the place of their publication and possibly, if they have one, with a list of synonyms. Finally, the work presented will be the most comprehensive treatment of the complex.

Figure 8: Should be improved. The scan is of poor quality. The scale is certainly wrong if it is supposed to apply to all elements, and there is no information that this is not the case. I suggest preparing a line-drawing based on the one in the paper.

There is no scalebar in some of the pictures in the plates. Since they appear on most of them, they should consistently be on all of them, i.e. Fig.11G

For consideration, the authors indicate the conservation status, you may want to use a tool like GeoCAT to estimate the distribution aspect of the IUCN criteria

Line 974: The end is missing, "nomen nudum"?

In the discussion, authors use links to databases, which sometimes have specific citation rules. This certainly applies to GBIF, where the appropriate DOI number is usually used. This should be corrected, all the rules are on the website.

Reviewer 4 Report

General:
The term "complex" is nowadays used quite loosely to denote taxonomic groups whose relationships may be too complex for the respective authors to be properly understood. Anyway, the term should be avoided whatsoever, as it does not provide any scientifically significant information.

Lines 1881-1882:
"will often quickly induce them to open (F. Hort, J. Hort & T. Krueger pers. obs.). Bagged flowers typically open within just a few minutes."
Seems to repeat the same observation and should be contracted to:
"will often induce them to open within just a few minutes (F. Hort, J. Hort & T. Krueger pers. obs.)." It should also be mentioned which species was/were investigated.

Line 2028:
Complete end of line and paragraph.

Line 2044:
Replace "RBGE" by "E".

Reviewer 5 Report

When referring to "the authors" it would be better to use first person pronouns to make clear that the authors of the manucript are being referred to and not the authors of one or more the cited papers.

It would be better to treat Drosera as a noun instead of an adjective 'species of Drosera' rather than 'Drosera species,' or else change the Latin ending of 'Drosera' to possessive. 
